# Molecular mechanisms and topological consequences of drastic chromosomal rearrangements of muntjac deer

Yuan Yin [1,20], Huizhong Fan[2,3,20], Botong Zhou [1,20], Yibo Hu[2,4,20], Guangyi Fan[5,6,7,20], Jinhuan Wang[8,20], Fan Zhou[9,20], Wenhui Nie[8,20], Chenzhou Zhang[1], Lin Liu[9], Zhenyu Zhong[10], Wenbo Zhu[1], Guichun Liu[1], Zeshan Lin [1], Chang Liu[1], Jiong Zhou[1], Guangping Huang[2], Zihe Li[1], Jianping Yu[11], Yaolei Zhang[5,12], Yue Yang[1], Bingzhao Zhuo[1], Baowei Zhang[13], Jiang Chang [14], Haiyuan Qian[11], Yingmei Peng[1], Xianqing Chen[1], Lei Chen[1], Zhipeng Li[15,16], Qi Zhou [17,18,19✉], Wen Wang [1,4✉] & Fuwen Wei [2,3,4✉]

Muntjac deer have experienced drastic karyotype changes during their speciation, making it an ideal model for studying mechanisms and functional consequences of mammalian chromosome evolution. Here we generated chromosome-level genomes for *Hydropotes inermis* (2n = 70), *Muntiacus reevesi* (2n = 46), female and male *M. crinifrons* (2n = 8/9) and a contig-level genome for *M. gongshanensis* (2n = 8/9). These high-quality genomes combined with Hi-C data allowed us to reveal the evolution of 3D chromatin architectures during mammalian chromosome evolution. We find that the chromosome fusion events of muntjac species did not alter the A/B compartment structure and topologically associated domains near the fusion sites, but new chromatin interactions were gradually established across the fusion sites. The recently borne neo-Y chromosome of *M. crinifrons*, which underwent male-specific inversions, has dramatically restructured chromatin compartments, recapitulating the early evolution of canonical mammalian Y chromosomes. We also reveal that a complex structure containing unique centromeric satellite, truncated telomeric and palindrome repeats might have mediated muntjacs' recurrent chromosome fusions. These results provide insights into the recurrent chromosome tandem fusion in muntjacs, early evolution of mammalian sex chromosomes, and reveal how chromosome rearrangements can reshape the 3D chromatin regulatory conformations during species evolution.

A full list of author affiliations appears at the end of the paper.

n mammals, the chromosome number ranges from $2n = 6$ in the female Indian muntjac (*M. muntjak vaginalis*) to $2n = 102$ in the viscacha rat (*Tympanoctomys barrerae*)[1,2]. Even in related species, such as within rodents[3], gibbons[4] and muntjacs[5], the chromosome number can be dramatically different. Previous cytogenetic studies have also revealed that the male and female of some species harbored different karyotypes that only formed very recently[6]. The molecular cause and consequence of karyotype changes in mammals has long been an unsolved genetic mystery[7,8]. Muntjac deer (*Muntiacus*, Muntiacinae, Cervidae) were quoted in Barara McClintock's Nobel lecture as a spectacular example with stunning changes of chromosome organization[9,10]. They were proposed to have an ancestral karyotype ($2n = 70$) similar to that of *Hydropotes inermis*[11], and recurrent chromosome fusions have led to the karyotypes of extant species varying from $2n = 46$ of *M. reevesi*[12] to $2n = 8/9$ of *M. crinifrons* or *M. gongshanensis*[13,14], and to $2n = 6/7$ of *M. muntjak vaginalis*[15]. Previous studies on the sequence composition of some fusion sites suggested that the chromosome fusions may be produced by sequence-specific recognition and illegitimate recombination between homologous DNA elements (or other specific motifs) on nonhomologous ancestral chromosomes[16–18]. However, the concrete molecular and evolutionary mechanisms underlying such recurrent and massive chromosome fusions still remain to be elucidated.

As one of the muntjacs with a very low chromosome number, *M. crinifrons* additionally possesses one sex chromosome system that does not exist in other muntjac species including its closest relative, *M. gongshanensis*. In *M. crinifrons*, the original eutherian X chromosome had experienced a centric fusion to one copy of chromosome 4, forming the "X + 4" chromosome, and the short arm of chromosome 1 had undergone a male-specific translocation to another copy of chromosome 4, creating the "1p + 4" chromosome[19–22] (Supplementary Fig. 1). Interestingly, two inversions involving a large part of the "1p + 4" were identified in male *M. crinifrons*, making the influenced regions (the 'neo-Y' regions) to evolve like a canonical mammalian Y chromosome, and its homologous counterparts on the chromosome X + 4 and chromosome 1p to be neo-X regions. Most mammalian Y chromosomes originated over 180 million years ago and bear few functional genes or traces of their evolution but only massive transposable elements, because of long-term suppression of homologous recombination[23–25]. The young neo-Y chromosome of *M. crinifrons* provides us a rare opportunity to explore the mammalian Y chromosome evolution at the very early stage. Previous study on a few *M. crinifrons* neo-Y-linked genes showed that they had indeed accumulated deleterious mutations that either disrupted the open reading frames (ORF), or downregulated the expression of some genes[22]. However, the neo-Y degeneration process at the whole chromosome level, which bears important implications for the early stage mammalian Y chromosome evolution, remain unknown.

The dramatic difference in karyotypes among the closely related muntjac species also provides an ideal model to study the evolution of three-dimensional (3D) chromatin architectures during speciation. Besides facilitating genome assemblies[26], the Hi-C data are widely used to reveal the 3D chromatin architectures in terms of active (A) or inactive (B) chromatin compartments, topologically associated domains (TADs), and significant chromatin interactions[27]. Previous studies revealed that TADs seem to be generally conserved between species[28–30], while artificially fused chromosomes of yeast exhibit remarkable changes in the global 3D genome structure and significant chromatin interactions[31]. This raised the question that how chromosome rearrangements would reshape different 3D chromatin architectures in vivo during species evolution.

Here, we produced high-quality chromosome-level genomes and a large quantity of Hi-C data for multiple muntjacs and *H.*

*inermis* representing ancestral karyotype (Fig. 1a and Supplementary Fig. 2), which enable us to reconstruct the detailed process of chromosome fusions from the muntjac ancestor to the extant species, investigate the molecular basis of dramatic chromosome fusion events during muntjac species evolution and explore the impact of chromosome fusions on 3D chromatin architectures. Our study provides insight to the species evolutionary history, the possible molecular basis of chromosome fusion and topological consequences of drastic chromosomal rearrangements of muntjac deer.

## Results

**Genome assemblies, annotation, and phylogenetic analysis.** Using high-coverage long reads and short-reads sequencing data combined with large-scale Hi-C data (Supplementary Fig. 2 and Supplementary Tables 1–4), we produced chromosome-level genomes of *H. inermis*, *M. reevesi*, male and female *M. crinifrons* and a contig-level genome of *M. gongshanensis*. In detail, we sequenced 75~122× long reads for female *M. crinifrons*, *M. reevesi* and *M. gongshanensis* using Oxford Nanopore Technology and used these data to produce draft genome assemblies. Then we used the Illumina short reads to polish these draft genomes and generated contig-level genome assemblies with contig N50 length ranging from 24.47 Mb to 37.86 Mb (Supplementary Table 5). For male *M. crinifrons*, we reassembled its genome with the PacBio data that we previously generated[32] and improved the contig N50 length from 1.46 Mb[32] to 3.79 Mb. Using Hi-C data, we further anchored 98.82%, 91.49%, 98.62% and 97.57% of the contigs from female and male *M. crinifrons*, *M. reevesi* and *H. inermis*[33] into 4, 5, 23, and 35 haploid chromosomes, respectively (Supplementary Fig. 3 and Supplementary Table 5), which are consistent with their reported karyotypes[20,34,35]. The high quality and continuity of these genome assemblies are reflected by their large contig N50 lengths, high Benchmarking Universal Single-Copy Orthologs (BUSCO) values (92%~95%) (Supplementary Table 5) and conserved synteny with the *Bos taurus* genome (Supplementary Fig. 4). We annotated 40%~44% of these genome sequences as repetitive sequences (Supplementary Fig. 5a) and about 21,000 protein-coding genes per genome (Supplementary Fig. 5b). The repeat and gene contents are similar to those of other ruminants[32].

To provide a phylogenetic framework for subsequent evolutionary analysis, by including the published *M. muntjak vaginalis*'s draft genome sequences[32], we reconstructed the maximum likelihood (ML) tree for muntjac deer based on the fourfold degenerate sites (4dTV) and mitochondrial genomes (Fig. 1a and Supplementary Fig. 6a). We estimated the divergence time among *M. muntjak vaginalis*, *M. gongshanensis* and *M. crinifrons* to be about 1~2 million years ago (Mya). Particularly the 1.44 Mya divergence time between *M. crinifrons* and *M. gongshanensis* (Fig. 1a and Supplementary Fig. 6b) sets the upper limit for the age of neo-sex chromosomes of *M. crinifrons*.

**Demographic history of muntjac deer.** We reconstructed the temporal changes of the historical population size of muntjac deer and related species. The results showed that the *M. muntjak vaginalis*, *M. gongshanensis*, and *M. crinifrons* with reduced chromosome number underwent a striking decline of population size about 1~2 Mya when they diverged with each other (Fig. 1b). However, such a population decline during this period was not detected in *M. reevesi* and two nonmuntjac deer species (Supplementary Fig. 7 and Supplementary Table 6), as well as in other previously investigated ruminants[32]. This time window overlaps with the Xixiabangma glaciation (XG) occurred 0.8-1.17 Mya in the area of the Hengduan mountain and its surroundings[36],

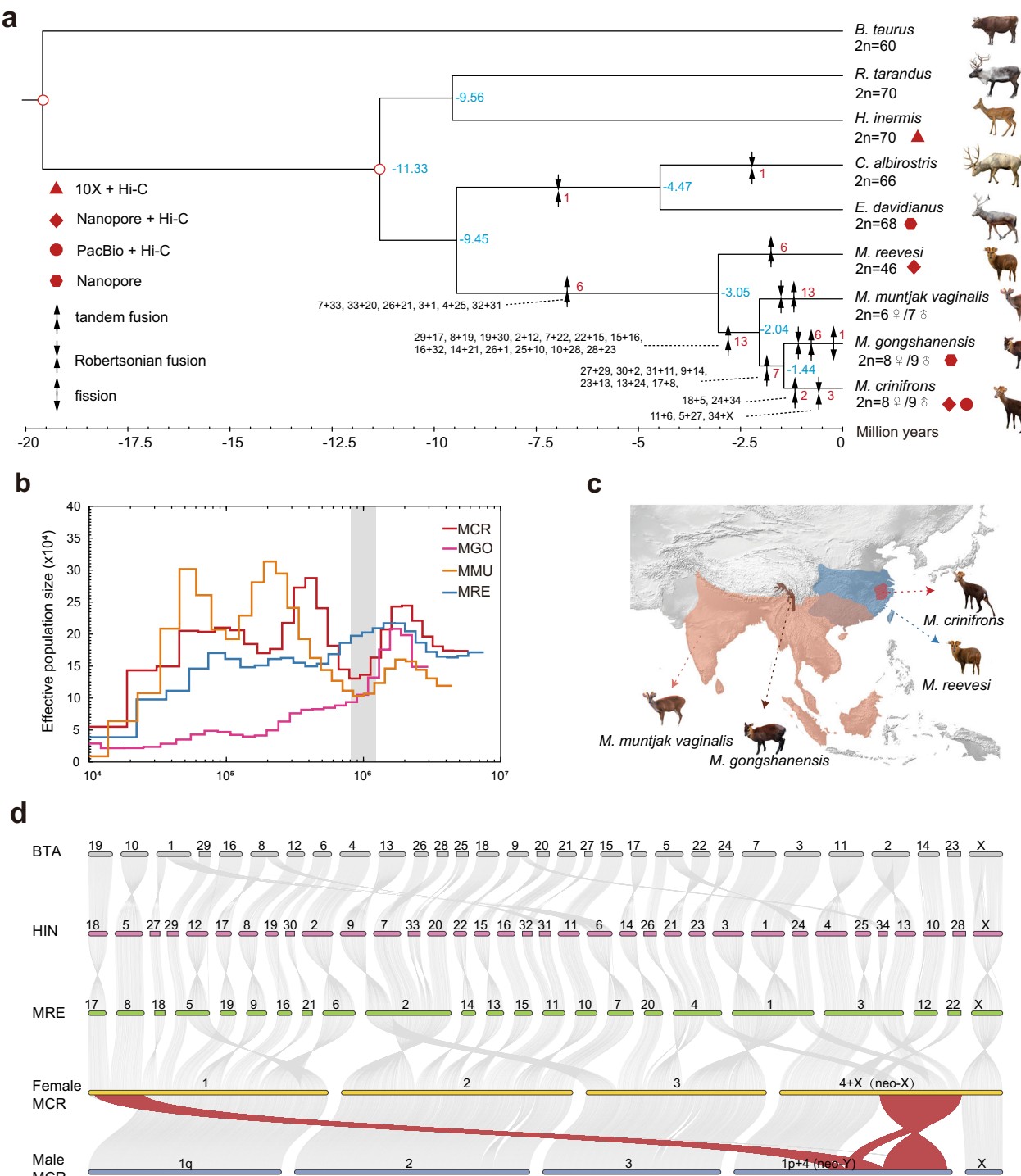

**Fig. 1 Phylogeny, demographic histories, and distribution and chromosome synteny of muntjac species. a** Maximum likelihood tree of muntjac and outgroup species with the respective sequencing technologies (red geometries), the divergence time (blue numbers) and number of chromosome fusion or fission events (red numbers) shown. Different combinations of black arrows represent different types of chromosome fusion and fission. The 31 fusion events leading to *M. crinifrons* are displayed in detail with the chromosome code (black numbers) of *H. inermis*, which are connected with the arrow mark on the phylogenic tree with dotted lines. Red hollow circles mark the nodes whose divergence times were used as calibration for estimating the divergence time among other species. **b** The demographic histories of *M. reevesi* (MRE), *M. muntjak vaginalis* (MMU), *M. gongshanensis* (MGO), and *M. crinifrons* (MCR) estimated by PSMC[37]. The gray box marks the time range of the Xixiabangma Glaciation (XG, 0.8-1.17 million years ago). **c** Topographic map on current geographic distribution of the four muntjac species. The colors of dashed line are consistent with the colors of distribution areas of a particular species. **d** The chromosome synteny between *B. taurus*, *H. inermis*, *M. reevesi*, female and male *M. crinifrons* with chromosome names shown above. 1p and 1q represent short arm and long arm of chromosome 1, respectively. The red line indicates the synteny blocks of female and male *M. crinifrons* in neo-Y inverted regions.

where *M. reevesi*, *M. muntjak vaginalis* and *M. gongshanensis* are now distributed (Fig. 1c). We proposed a hypothesis that genetic drift associated with the geographic isolation facilitated by XG might have facilitated the fixation of extensive chromosome fusions in *M. muntjak vaginalis*, *M. crinifrons*, and *M. gongshanensis* during or after their divergence from *M. reevesi*. It is noteworthy that this hypothesis still needs more data, such as more population data in the future, due to the reduced reliability of demographic inference of PSMC method at 1 Mya[37].

**Drastic chromosome rearrangements in muntjac deer**. We analyzed the chromosome rearrangement events occurred in different muntjac species through the whole-genome alignments. Our results confirmed that chromosome fusions are the dominant type of rearrangements that account for the drastic interspecific variations of muntjac chromosomal numbers (Fig. 1d). We further identified the putative orientation of centromeres' positions in the acrocentric chromosome preserved in *H. inermis* and *M. reevesi* (Supplementary Fig. 8). A fusion event was defined as tandem fusion if it connected the apical centromeres of one ancestral chromosomes and the distal telomere of another ancestral chromosome, and a fusion event was defined as Robertsonian fusion if it connected the apical centromeres of two ancestral chromosomes. Based on the centromeres' position and chromosomal synteny, in the female *M. crinifrons* we identified 28 tandem fusion events and 3 Robertsonian fusion events derived from the 2n = 70 ancestral karyotype (Fig. 1a and Supplementary Table 7). Combined with published cytological results[21,38,39], we determined occurrence order of the 31 fusion events along the phylogenetic tree from the common ancestor node of five muntjacs to the *M. crinifrons* (Supplementary Fig. 9), and estimated the rate of chromosome fusions as 8.20~8.52 events per million years among the *M. muntjak vaginalis*, *M. gongshanensis* and *M. crinifrons* after they diverged from *M. reevesi* (Fig. 1a and Supplementary Fig. 9). We compared the genome-wide substitution rates between species (*M. reevesi* (2n = 46), *H. inermis* (2n = 70) vs female *M. crinifrons* (2n = 8)) with dramatic karyotype changes and Bovidae species (*B. taurus* (2n = 60), *Ovis aries* (2n = 54) vs *Capra hircus* (2n = 60)) with similar karyotypes. The results show that at least 90% of female *M. crinifrons* genomic sequences could be mapped to *M. reevesi* and *H. inermis* with average sequence identity more than 90% (Supplementary Fig. 10 and Supplementary Table 8), similar to that of the Bovidae species (Supplementary Table 8). In addition, the substitution rates and mutation rate between different muntjac genomes are also similar to those between Bovidae species (Supplementary Tables 6 and 8). These results demonstrated that the rapid karyotype evolution among muntjacs is not accompanied by rapid evolution of genomic sequences.

**Impact of chromosome fusions on 3D chromatin architectures**. To illuminate the impact of drastic chromosome fusions on 3D chromatin architectures in these mammals, we generated Hi-C data using blood samples of *H. inermis*, *M. reevesi*, female, and male *M. crinifrons* (Supplementary Table 4). By dividing the genome to bins with the same length and counting the Hi-C read-pairs aligned to each bin or bin-pairs, we constructed the interaction matrix at the different resolution that are represented by size of the bin. Through the homologous mapping of bins between different genomes (Supplementary Fig. 11a, b), we compared their 3D chromatin architecture at different hierarchical levels, including compartment A/B, TADs, and significant interactions.

At a 100 kb resolution, we identified the A or B (active or inactive) compartments of different muntjac species following the

previous practice[40,41]. As expected, we found significantly higher gene density and GC content, higher gene expression level in the euchromatic A compartments than in the heterochromatic B compartments (Supplementary Fig. 12a, b)[40]. We also found that over 90% of the genomic regions of female *M. crinifrons* (Fig. 2a and Supplementary Fig. 13), as well the flanking regions of fusion sites, (Supplementary Fig. 14a, b) have the same compartment type with their homologous regions in *H. inermis*, *M. reevesi*, and male *M. crinifrons*, indicating that the fusions have little impact on the chromatin compartment type.

At a finer resolution (40 kb), we compared the TADs of male *M. crinifrons*, *M. reevesi* and *H. inermis* with female *M. crinifrons* based on their reciprocally mapped bins. Following the criteria of a previous study[41], we defined the TADs as conserved TADs which have over 70% overlapped intervals between genomes or species. The results showed that ~73.3% of TADs are conserved in female and male *M. crinifrons* (Fig. 2b), which is consistent with the TAD stability observed between different human cell lines[41]. The proportions of interspecific conserved TADs between female *M. crinifrons* and *M. reevesi*, between female *M. crinifrons* and *H. inermis* are 63.7%~65.4% and 43.3%~49.2%, respectively (Fig. 2b), indicating that the conservation of TADs between different species decrease with the increase of divergence time. In addition, we further obtained a total of 316 *M. crinifrons*-specific TADs (Methods) that were absent in all the other examined species. After examining the distance between bins in these TADs and nearest fusion sites, we found that these bins are almost evenly distributed along chromosomes, but are not enriched near the fusion sites (Fig. 2b), indicating that the recurrent chromosome fusions also have little impact on TADs.

We then compared significant interactions of different muntjac species at a 20 kb resolution (Supplementary Fig. 15) and found that *M. crinifrons* has established more long-range significant interactions (Supplementary Fig. 16a). In detail, 73.12% (15148/56345) of these long-range significant interactions (>5 Mb) are established within ancestral chromosome segments, indicating that these long-range significant interactions in *M. crinifrons* are not due to the calculation error caused by its super long chromosomes. Furthermore, most of these long-range significant interactions (88.38%) have no homologous significant interactions in *M. reevesi* (Supplementary Fig. 16b). These results suggested that these long-range significant interactions may be related to the more compacted chromosomes observed in the reconstructed 3D genome structure of *M. crinifrons* (Supplementary Fig. 17). In addition, the majority of long-range significant interactions connect the same compartment type (Supplementary Fig. 16c, d), indicating that during the compression of giant chromosomes, the chromatin regions with the same compartment type tend to be closer in space as reported before[40]. However, the reconstructed 3D genome structure revealed that ancestral chromosome segments fused in *M. crinifrons* are not physically closer in the reconstructed 3D genome structures of *M. reevesi* and *H. inermis* (Supplementary Fig. 17), indicating that the fusion events were not directly caused by spatial proximity of two ancestral chromosomes.

Finally, we investigated the significant interactions across fusion sites in female *M. crinifrons*. The result showed that almost all of the significant interactions across fusion sites (95.37%) were newly established in *M. crinifrons* compared to *M. reevesi* (Fig. 2c). This is in contrast to the rest of the significant interactions that do not span the fusion sites, in which only 44.43% of them are unique in *M. crinifrons* (Fig. 2c). These results provide evidence that chromosome fusion may be a major driver of these novel significant interactions across fusion sites in *M. crinifrons*. In addition, most of these significant interactions only span one fusion site and anchor their two ends on two adjacently fused

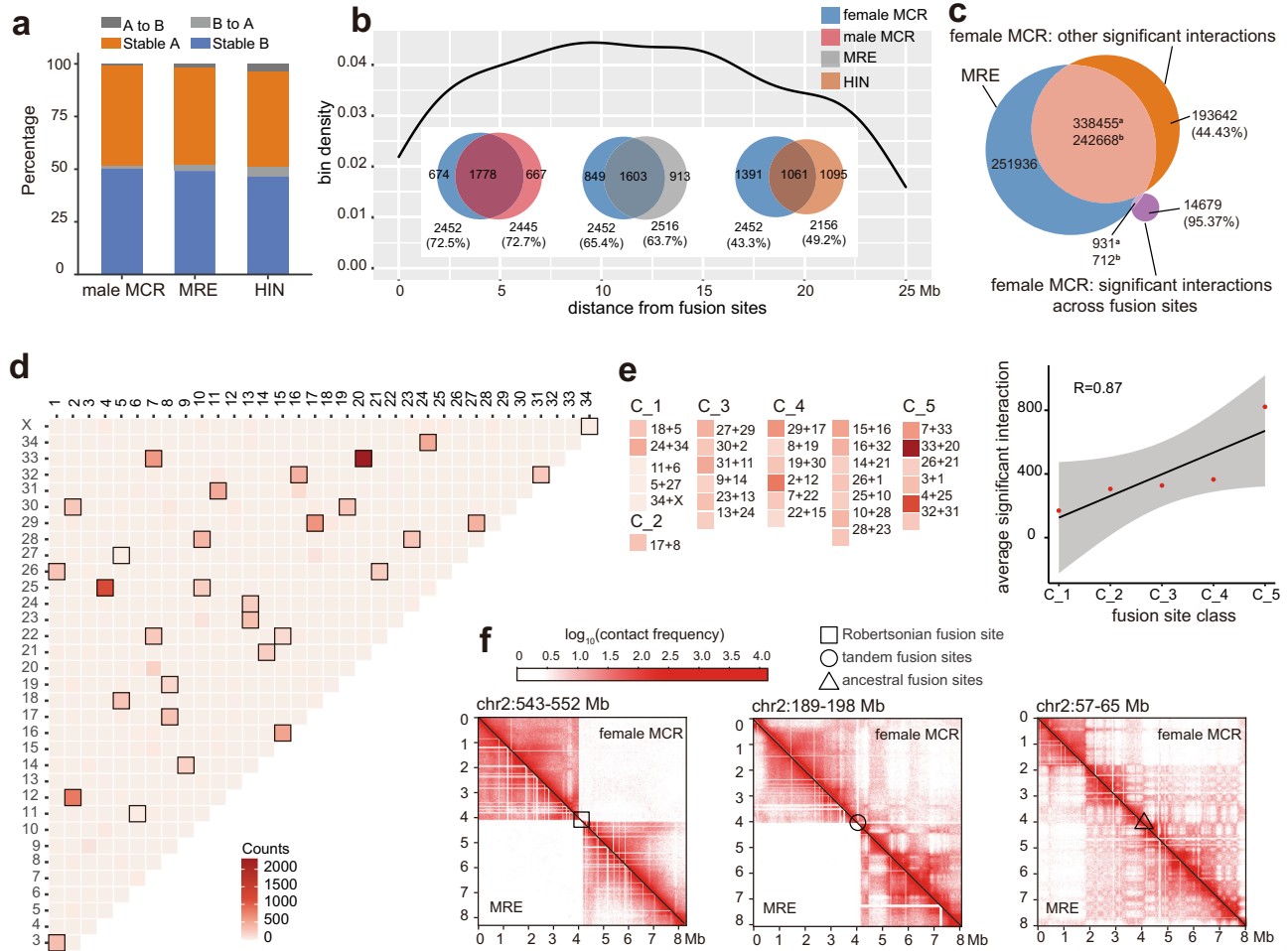

**Fig. 2 Evolution of 3D chromatin architectures along with chromosome fusion in muntjacs. a** Comparison of compartment A/B between female *M. crinifrons* and other three genomes. "A to B" means the homologous bins are compartment A in female *M. crinifrons* but compartment B in other genome. "B to A" is the opposite. "stable A" or "stable B" indicates the homologous bins in all genomes have the same compartment type. "Other" represents bins without exact compartment type. MCR, *M. crinifrons*; MRE, *M. reevesi*; HIN, *H. inermis*. **b** Comparison of TADs. Venn diagrams show the number and proportion of conserved or specific TADs in female *M. crinifrons* compared with other three genomes. Density map shows density distribution of *M. crinifrons*-specific TAD bins along fusion sites. **c** Number of shared and specific significant interactions between *M. reevesi* and female *M. crinifrons*. Superscript "a" indicates significant interactions of *M. reevesi* shared by female *M. crinifrons* and superscript "b" indicates significant interactions of female *M. crinifrons* shared by *M. reevesi*. In parentheses are the percentage of significant interaction specific to female *M. crinifrons*. **d** Heatmap of number of significant interaction across fusion sites in female *M. crinifrons*. The horizontal and vertical coordinates are ancestral chromosomes represented by chromosome coeds of *H. inermis*, on which the significant interaction anchored their two ends. The ancestral chromosome pairs adjacently fused in female *M. crinifrons* are framed by black boxes. **e** Number of significant interactions just across one fusion site extracted from Fig. 2d. and displayed according to ages of fusion site (C_1-C_5). The average number of significant interactions were fitted linearly with the age stage of fusion site using Pearson method. **f** Combined heatmaps of contact matrix around the fusion sites of female *M. crinifrons* (upper right) and their homologous regions in *M. reevesi* (lower left) at 20 kb resolution. Hollow shapes mark the locations of fusion sites. Different geometries represent different fusion site types. The "ancestral fusion sites" are the oldest tandem fusion sites. The "tandem fusion sites" represent the rest tandem fusion sites. The "Robertsonian fusion sites" refers to the youngest fusion sites raised by Robertsonian fusion.

ancestral chromosomes, while only a few (16.82%) span two or more fusion sites (Fig. 2d). Interestingly, the abundance of significant interactions across fusion sites is positively associated with the ages of fusion sites (Fig. 2e and Supplementary Fig. 9). Similar patterns can also be directly seen in those interaction matrix that we constructed using Hi-C read pair for all 3D chromatin structure analyses (Fig. 2f and Supplementary Fig. 18). Particularly, the frequency of Hi-C read pair interactions spanning the oldest fusion sites has reached the same level as those in other genomic regions that have not undergone fusions (Fig. 2f, Supplementary Fig. 18). Such a gradient of interactions exhibited by the different age of fusion sites indicates that novel *cis*-interactions can be generated by chromosome fusions. Some new

significant interactions across fusion sites may be of biological significance during the species divergence. This can be implicated by the genes involved in the significant interactions spanning the youngest two classes of fusion sites formed in the lineages of *M. crinifrons* and the common ancestor of *M. crinifrons* and *M. gongshanensis*, respectively (Supplementary Data 1). These genes are enriched in GOs or pathways related to lipid metabolism, bone morphogenesis and melanogenesis (Supplementary Data 2). Interestingly, *M. crinifrons* and *M. gongshanensis* are very similar to each other except for the fur color, body size, and habitat climate[42]. Further comprehensive studies on these lineage-specific interacting genes may provide more insights into the genetic basis of different traits in different muntjac species.

In summary, the chromosome fusion events have little impact on the compartment type and TADs, even near fusion sites, but they can lead to more novel significant interactions connecting distant genomic loci or across the fusion sites, which may have both cellular and morphological significances during the evolution and adaptation of muntjac deer.

**Possible molecular basis of chromosome fusions in muntjac species.** To dissect the possible molecular cause of chromosome fusions, we tested two mutually nonexclusive hypotheses: 1) certain muntjac-specific repetitive sequences have mediated tandem fusions through nonallelic homologous recombination; 2) mutations in genes related to regulating genome stability may contribute to the rapid evolution of muntjac chromosomes. The first hypothesis predicts that a certain combination of centromeric and telomeric sequences to be enriched at the fusion sites, as previously reported at *M. muntjak vaginalis* fusion sites[16]. We firstly checked the fusion sites in the genome of female *M. crinifrons*, the results showed that only one ancient fusion site shared by all muntjacs was completely assembled without gaps in female *M. crinifrons* (Supplementary Notes, Supplementary Fig. 19 and Supplementary Table 9), possibly because of the difficulty of assembling the long and complex repeat architectures at the fusion sites. We therefore searched for evidence and sequence features of chromosome fusions directly in the raw Nanopore reads (Supplementary Table 10). In addition, in order to have a better resolution in comparing these reads from the fusion sites, we generated the Nanopore reads of a more closely related Cervinae species than *H. inermis* (belonging to Hydropotinae) to muntjac deer, *Elaphurus davidianus*, with a 2n = 68 karyotype.

Previous cytogenetic studies revealed that apical telomere, satellite II (satII), then IV (satIV) and I (satI) repetitive sequences are sequentially arranged at the centromeric region of *M. reevesi* acrocentric chromosomes[43] and the satI and telomeric repeats are immediately adjacent in a few fusion sites of *M. muntjak vaginalis*[16]. Therefore, we first focused on the nanopore reads containing both satI and telomeric repeats in the four investigated species (*M. crinifrons*, *M. gongshanensis*, *M. reevesi*, and *E. davidianus*). For most (58%~75%) of these reads in *M. crinifrons* and *M. gongshanensis*, the satI and telomeric repeats were immediately juxtaposed (satI-telomere: <500 bp) to each other, but far away from each other (satI-telomere: >=500 bp) in most (92.16%~93.75%) of such reads in *M. reevesi* and *E. davidianus* (Fig. 3a and Supplementary Table 11). Considering that both *M. crinifrons* and *M. gongshanensis* have 31 fusion sites, while *M. reevesi* and *E. davidianus* have only 12 and 1 fusion sites, respectively (Fig. 1a), these results suggest that the satI-telomeric sequence juxtaposed reads may possibly be derived from fusion sites and the reads with satI and telomeric sequence far away from each other may be from the centromeric regions of ancestral acrocentric chromosomes largely retained in *M. reevesi* and *E. davidianus*. The length distribution of telomeric sequence in these satI and telomeric repeats-containing reads can be used for testing this hypothesis. The length of the telomeric sequence in the satI-telomere juxtaposed reads is scattered (Fig. 3b), corresponding to its origin from random breakage of ancestral distal telomere during chromosome fusion. The length distribution of telomeric sequences in the reads with satI and telomeric sequence far away from each other is concentrated at about 38 bp in *M. reevesi* (Fig. 3b), which is far shorter than the average length (667 bp) of telomeric sequences observed on all reads containing telomeric sequences. Similar truncated telomeric sequences (~25 bp) were also found in reads containing satellite IV and telomeric (satIV-telomere) or satellite I, satellite IV, and telomeric sequences (satI-satIV-telomere) (Fig. 3b). Further examination revealed that the truncated telomeric sequence is primarily

located in the middle, not at the ends of the Nanopore reads (Supplementary Fig. 20), which also indicated that these truncated telomeric sequences are not caused by pre-termination of the Nanopore reads, but are more likely derived from the acrocentric regions of ancestral chromosomes and have originally been located in the region between satI and satIV (Supplementary Fig. 21). Intriguingly, we found that the truncated telomeric repeats in about 90% of these reads containing satI or satIV sequence are flanked by palindromic sequences (Supplementary Fig. 21 and Supplementary Table 11) which were previously reported to be genomic fragile sites where DNA double-strand breaks can frequently happen[44–46]. In addition, the three types of reads with truncated telomeric sequence are only abundant in *M. reevesi* (Fig. 3c and Supplementary Table 12), indicating that such truncated telomeric sequence and its flanked palindromic sequence and centromeric satI and satIV might have been amplified in the apical-centromere regions in the muntjac ancestor's chromosomes. After frequent DNA double-strand breaks nearby the palindromes, the truncated telomeric repeats might have mediated nonallelic homologous recombination (NAHR) between different ancestral chromosomes and resulted in massive tandem fusions, and then extensive loss of such structures in the post-fusion species like *M. crinifrons* and *M. gongshanensis* (Fig. 3d).

To test the second hypothesis, we identified the rapidly evolving genes (REGs) and positively selected genes (PSGs) in the *M. crinifrons*, *M. gongshanensis*, and *M. muntjak vaginalis* with large fused chromosomes, as well in their common ancestor node (Supplementary Data 3). The results showed that the PSGs and REGs in these lineages are enriched in GOs and pathways related to the maintenance of genomic stability (e.g., cell cycle, DNA damage response, or telomere maintenance)[47–49] (Fig. 3e and Supplementary Data 4). Among these lineages, orthologs of a total of 72 REGs and PSGs have been functionally tested in mice, and they seem to play important roles in processes related to cell cycle, DNA repair, or chromosome stability (Supplementary Data 5). However, since the occurrence/frequency of genomic rearrangements (>10 kb) of *M. crinifrons* and *M. gongshanensis* (3.06~3.89 events/Mb) are not significantly higher than those in *M. reevesi*, *E. davidianus*, and *C. albirostris* (3.11~4.56 events/Mb) (Supplementary Table 13), it is possible that at least some of these REGs and PSGs related to cell cycle, DNA damage response or telomere maintenance are selected to stabilize the giant fused chromosomes during these cell processes, rather the cause of chromosome fusions. Future experiments with artificial chromosomes can be the solidest evidence to this conclusion.

**Evolution of neo-sex chromosomes in *M. crinifrons*.** To systematically study the rare mammalian neo-sex system originated in less than 1.5 Mya in *M. crinifrons* (Fig. 1a), we first identified 375 Mb inverted neo-Y region in male *M. crinifrons* and 378 Mb homologous neo-X regions in female *M. crinifrons* (Fig. 4a, Supplementary Table 14 and methods). The alignment results showed that 97% of the neo-Y regions sequence can be aligned to the neo-X with an average 99.5% sequence identity, and the repeat content of the neo-Y and neo-X regions are almost the same (41.14% vs 41.29%). The neo-X regions also exhibited a similar male and female coverage from Illumina reads (Fig. 4a, track C). These results indicated that the degeneration of mammalian neo-Y chromosomes is much slower than that of *Drosophila miranda*, whose neo-Y is of a similar age, but has already lost 40% of the functional genes[50].

Although limited sequence degeneration was identified in the neo-Y regions, we found higher densities of SNPs and insertions/

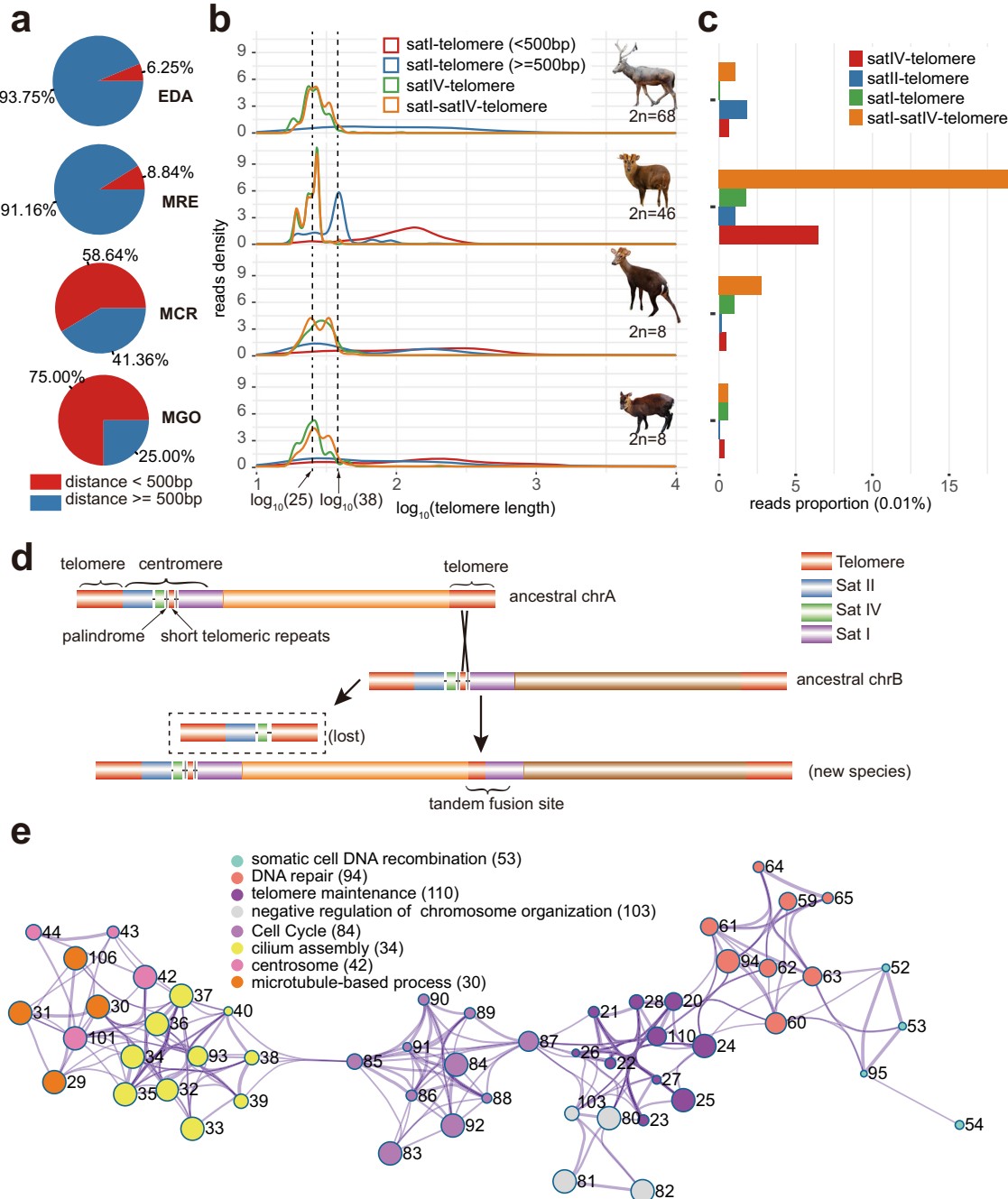

**Fig. 3 Molecular basis of tandem chromosome fusion of muntjac species. a** Proportion of reads with distance between satellite I and telomeric repeats smaller (red) or longer (blue) than 500 bp. Only reads containing telomeric sequences and satellite I but not satellite II and IV are considered in this statistic. EDA, *E. davidianus*; MRE, *M. reevesi*; MCR, *M. crinifrons*; MGO, *M. gongshanensis*. **b** Kernel density estimate of reads with different length of telomeric sequence. Reads containing different combinations of satellite and telomeric sequence are displayed separately with colored lines. satI, satellite I; satIV, satellite IV; telomere, telomeric sequence. The top-to-bottom of species order is consistent with that of Fig. 3a. **c** Proportion of reads containing different combinations of satellite or telomeric repeats in all investigated reads of *E. davidianus*, *M. reevesi*, *M. crinifrons*, and *M. gongshanensis*. **d** Schematics of a conjectural mechanism of tandem fusion. DNA double-strand breaks occurred at palindromes nearby the short telomeric repeats, which mediated nonallelic homologous recombination between different ancestral muntjac chromosomes and led to the tandem fusions. **e** Network plot of GO terms enriched by REGs and PSGs of *M. gongshanensis*. REGs and PSGs are pooled in the GO enrichment analysis and each GO term is numbered. GO terms with a similarity >0.3 are connected by edges. Circles with the same color represent that GO terms belong to the same cluster. The size of each circle reflects the *p* value, where terms containing more genes tend to have a more significant *p* value. For each cluster, we display the name and number of a representative GO term. Only clusters related to cell cycle, nuclear division, DNA damage response or telomere maintenance are displayed here.

deletions (indels) of male *M. crinifrons* individuals than those of female in the homologous neo-sex regions but not in the rest of the genome (Fig. 4a, track D), indicating the early divergence between neo-Y and neo-X. In detail, we found that most (66.46%)

of candidate male-specific mutations are in the neo-Y regions (Fig. 4a, b, and Supplementary Fig. 22a, b), in sharp contrast to the sequence proportion of neo-Y region in the whole-genome (15.37%). Based on the distribution of candidate male-specific

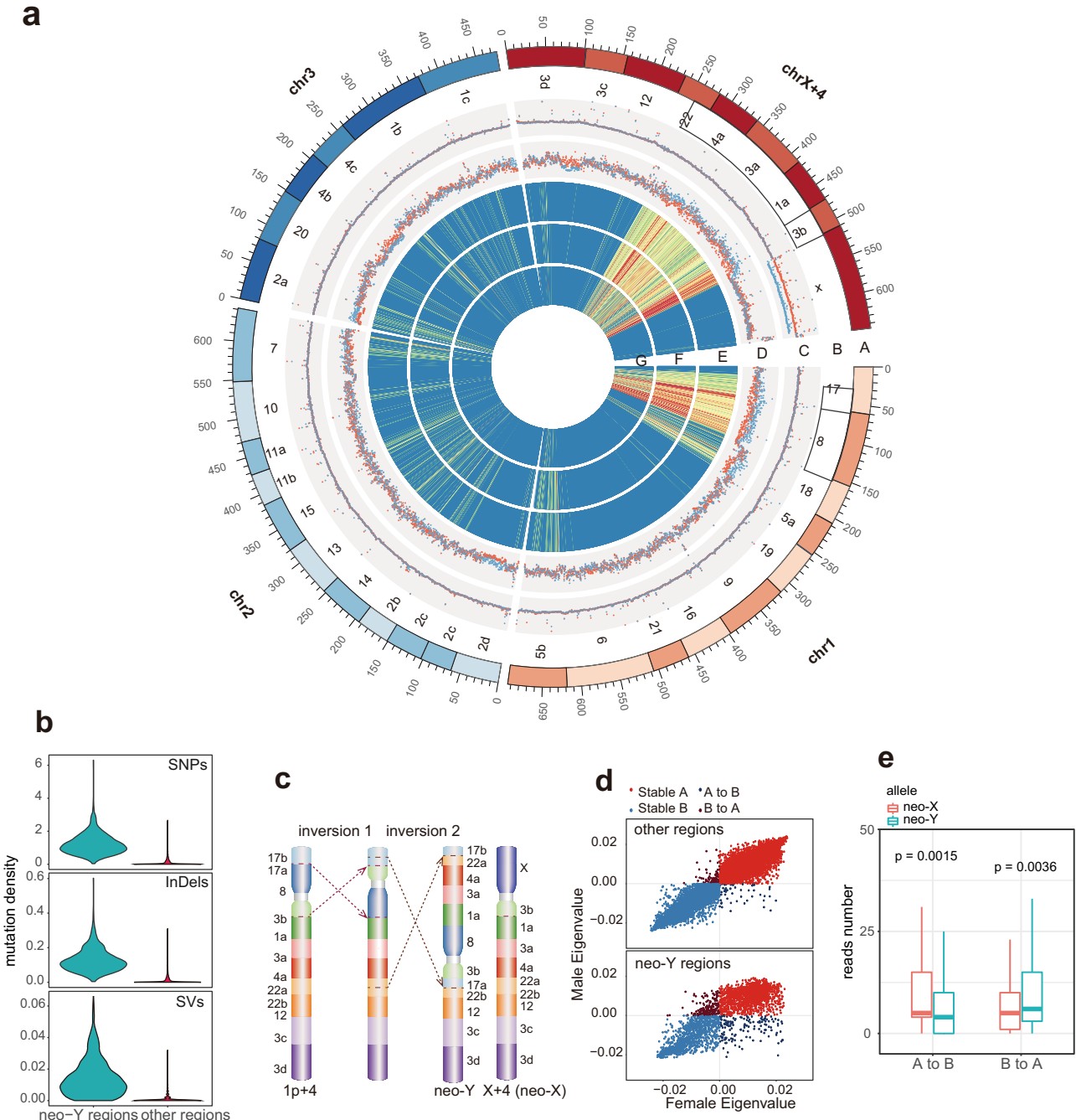

**Fig. 4 Evolution of neo-sex chromosomes in *M. crinifrons*. a** Circos plot of female *M. crinifrons* genome. Track A: chromosomes of female *M. crinifrons* showed by 35 colored ancestral chromosomes. Track B: labels of ancestral chromosome under previous nomenclature[21]. Black frames enclose neo-X regions. Track C: Mapping coverage of female (red) and male (blue) *M. crinifrons* Illumina reads. Track D: SNPs and indels density of female (red) and male (blue) *M. crinifrons*. Track E: Heatmap showing the density of candidate male-specific SNPs. Density from low to high correspond the color group "spectral-7-div-rev" in circos. Track F: Heatmap showing the density of candidate male-specific indels with the same colors coding as track E. Track G: Heatmap showing the density of candidate male-specific SVs with the same colors coding as track E. **b** Violin plot showing the density of candidate male-specific mutations in neo-Y regions and other regions. **c** Two inversion events happened on the neo-Y chromosome of male *M. crinifrons*. Block 17a, 17b, 22a, and 22b are ancestral chromosomal segments disrupted by inversions and each of the remaining colored blocks represent the completed ancestral chromosome. **d** The eigen values of homologous bins in female and male *M. crinifrons*. Eigen value greater and less than zero means compartment A and B, respectively. The "other regions" indicate genomic regions excluding neo-Y regions and mammalian X chromosome region. **e** Boxplot of number of RNA-seq reads supporting neo-X or neo-Y allele in male *M. crinifrons* sample. "A to B" means compartment switch from A in neo-X regions of female *M. crinifrons* to B in homologous neo-Y regions of male *M. crinifrons*. "B to A" is opposite. There are 160 pairs of neo-X and neo-Y alleles in "A to B" regions and 366 pairs in "B to A" regions. The lower and upper hinges correspond to the first and third quartiles. The line inside box is the median. The whisker extends from the hinge to at most 1.5 * IQR (where IQR is distance between the first and third quartiles). Outliers are hidden. Difference in reads number supporting neo-X and neo-Y alleles was tested by two-sided Wilcoxon test (*p* value <0.05).

mutations, we corrected the order of the two inversions (Fig. 4a, c) previously proposed based on the cytogenetic data[21]. In addition, we further analyzed the candidate male-specific mutations and found that these mutations have led to high-density of nonsynonymous amino acid substitutions in neo-Y regions (0.278/kb CDS vs 0.024/kb CDS in other regions) (Supplementary Table 15) and disruption of open reading frames (ORF) in 107 genes in neo-Y regions. In particular, ten of these ORF-disrupted genes express equally in male and female samples but differently in neo-X and neo-Y alleles in male sample (Supplementary Data 6), which may reflect the initiation of dosage compensation, or the affected alleles have suffered mutations in the regulatory regions.

We finally examined how the 3D chromatin architectures have changed in response to the large neo-Y inversion. In contrast to the highly similar A/B compartment pattern between female and male *M. crinifrons* in general (Fig. 2a), the neo-Y region has accumulated significantly more A/B compartment switches (8.9% of total 3,310 bins have undergone compartment switch, $\chi^2$ test, $p < 0.01$) than other genomic regions (only 0.7% of 17,640 bins have undergone compartment switch) (Fig. 4d and Supplementary Fig. 23a, b). There are 131 genes in the neo-Y regions with switched compartment (Supplementary Data 7). The 57 genes in regions with compartment switched from A in female *M. crinifrons* to B in male *M. crinifrons* are enriched in GOs or pathways related to metabolism and biosynthesis (Supplementary Data 8), and the 74 genes in regions with an opposite direction of compartment switch are enriched in GOs or pathways related to transmembrane transport and development (Supplementary Data 8). This result indicates that these function-related genes located in neo-Y region are firstly affected by compartment switch. Moreover, we also observed that the compartment switch on the neo-Y region may impact the expression patterns of enclosed genes. Our results show that 526 sites on 87 genes that can be distinguished by candidate male-specific mutation generally have significantly higher expression level in compartment A than compartment B in the neo-Y region with compartment switch (Fig. 4e). Moreover, the compartment switch from A in neo-X regions to B in neo-Y regions is accompanied by the significant downregulation of gene expression of neo-Y alleles compared with neo-X alleles (Fig. 4e), while the opposite type of compartment switch is accompanied by the significant upregulation expression of neo-Y alleles (Fig. 4e). In contrast to compartment A/B, we did not find significant differences of TADs between female and male *M. crinifrons* genomes in the homologous neo-sex regions relative to other regions (Supplementary Table 16). These results indicate that, although few degenerations happened at the DNA sequence level, degeneration of gene expression regulation in mammalian young neo-Y chromosome might have happened at the level of the large-scale chromatin architecture (compartment switch).

## Discussion

Muntjac deer provide an excellent model for studying molecular basis and functional consequences of mammalian chromosome rearrangements[5,38,39,51]. How and why their chromosomes experienced recurrent tandem fusions remains unknown. Demographic analysis suggested population contractions during the XG period one Mya, may have provided unusual premises for rapid chromosome speciation by fixing some fused chromosomes in muntjac ancestral populations. By further scrutinizing the sequence characteristics around fusion sites, we revealed that a complex repeat structure very likely mediated the illegitimate recombination of nonhomologous chromosomes and thereby resulted in recurrent chromosome fusions. Our exploration on the molecular basis of chromosome fusion in muntjac species provide new insight into the mechanism of recurrent chromosome fusion.

The giant chromosomes resulted from recurrent fusions of ancestral smaller acrocentric chromosomes further provide a unique model to tackle the question of how chromosome rearrangements can reshape 3D chromatin architectures. Our results show that new long-range significant interactions and significant interactions across fusion site have been extensively established along with the chromosome speciation in muntjacs, but compartment type are conservative even between such distantly related species as *M. crinifrons* and *H. inermis*, and the conservation of TADs is more likely related to a genetic distance of species, but not affected by chromosome fusions. These discoveries shed lights on the chromatin regulatory roles of chromosome rearrangement events during species evolution. It is noteworthy that a recent study comparing only *M. reevesi* and *M. muntjak vaginalis* without other species as control, concluded that the compartments are not well conserved between *M. reevesi* and *M. muntjak vaginalis*[52], which is in contrast to our results. We downloaded their data and conducted careful comparison and found their analyses and conclusions had problems. Firstly, the quality of their genomes and the amount of Hi-C data are limited. The contig N50 lengths of muntjac genomes we assembled (3.79~37.86 Mb) are 15~150 folds higher than that of *M. reevesi* and *M. muntjak vaginalis* genome (~200 kb) assembled by Mudd et al.[52]. We used about 100 folds more Hi-C data (264~328×) than Mudd et al. (~30×) to analysis the 3D chromatin architecture, making the resolution reach 20 kb, which is sufficient to analyze 3D chromatin architectures at different hierarchical levels. The amount of Hi-C data sequenced by Mudd et al. only guarantee 1 Mb resolution which is not suitable for analyzing local compartment[53–57]. Secondly, the method Mudd et al. used to detect and compare the compartment of *M. reevesi* and *M. muntjak vaginalis* is even more problematic. They mapped the Hi-C reads from *M. reevesi* to *M. muntjak vaginalis* genome diverging ~3 Mya, which caused the reduction of half of the mapping rate (26.63%) compared with aligning *M. reevesi's* Hi-C reads to its own genome (59.373%) (Supplementary Table 17). We mapped the Hi-C reads of *M. muntjak vaginalis* and *M. reevesi* from Mudd et al. to their own reference genomes and used all of the interaction information on the whole chromosome to identify the global compartments as a regular practice in Hi-C compartment analysis[40]. A similar result to our comparison results between *M. crinifrons* and *H. inermis* or *M. reevesi* was obtained, namely, the compartment structure is conserved between *M. muntjak vaginalis* and *M. reevesi* (Supplementary Fig. 24). In the future, more studies on the 3D chromatin architecture of other organismal lineages with chromosome rearrangements can further testify our conclusions.

The study on sex chromosome evolution in mammals is difficult because after hundreds of millions of years of degeneration, few traces are retained on the Y chromosome[25]. Neo-sex chromosomes which are usually formed by the fusion of autosomes and sex chromosomes with subsequent suppression of recombination also undergo degeneration similar to sex chromosomes. The neo-sex chromosomes of *M. crinifrons* are the youngest known mammalian neo-sex chromosomes[22]. Due to technical limitations, only a few of sequences and genes on neo-Y chromosome of *M. crinifrons* were analyzed before[22]. In the current study, we not only assembled the almost complete neo-X and neo-Y chromosomes of *M. crinifrons*, but also dissected the neo-Y chromosome in many aspects based on various data types, including the degeneration level of neo-Y, gene death, dosage alternation of gene expression, and changes of 3D chromatin architectures. Particularly, our results show that Y chromosome may firstly become inert by extensively switched chromatin

compartment types and then followed by gradual sequence and gene expression degeneration.

Interestingly, our results show that chromosome fusions have little impact on both the A/B compartment and TADs, but chromosome inversions in the neo-sex chromosomes seem to drive more dramatic compartment changes. This may be caused by the formation mechanism of TADs, the biological effect of the compartment, and the different changes caused by chromosome fusion and recombination inhibition. TAD is formed by loop extrusion coordinated by boundary proteins, including CTCF and architectural proteins cohesion[58]. CTCF can identify specific DNA sequence motifs and limit the boundary of TAD[58]. Therefore, as long as CFCF and cohesion proteins, as well as the sequence motif, are not affected by chromosome fusion or inversion, TADs do not seem to be affected. Large-scale chromosomal rearrangement, such as fusion or inversion can barely affect CTCF and cohesion proteins. The TAD boundary sequence motif is short and thus unlikely being damaged biasedly in recombination suppression regions or regions near fusion sites. The compartment type is associated with chromatin type, where compartment A corresponds to euchromatin and compartment B corresponds to heterochromatin[40]. Genes in the A compartment tend to be actively expressed, while those in the B compartment tend to be silent[41]. Compartment switch would thus be accompanied by changes of gene expression[41]. Sequence degradation and chaotic gene expression caused by recombination inhibition may lead to compartment switch happened first in neo-Y regions while homologous sequences between genomes of different species are more likely to maintain the stability of compartments.

Overall, we have generated high-quality genome assemblies for multiple muntjacs and outgroup deer species. Combined with Nanopore long reads and large-scale Hi-C data, we have explored the possible molecular cause of chromosome fusion and its effect on the 3D chromatin architectures. Our results sheds lights on the molecular basis of recurrent chromosome fusions, the evolution of regulatory 3D chromatin evolution caused by chromosome rearrangements during mammalian species evolution, and early evolution of mammalian sex chromosomes. In the future, more studies on the 3D chromatin architecture of other organismal lineages with chromosome rearrangements can further testify our conclusions.

## Methods

**Genome sequencing and assembly.** The blood, frozen tissue, and fibroblast cell lines samples from different muntjac species and related deer species were used for Nanopore, Illumina, Hi-C, and RNA sequencing analyzes. The contig-level genomes of different muntjac species were firstly assembled and polished using different software. Then these assembled genomes were anchored to chromosome using high-quality Hi-C data. Particularly, we manually adjusted chromosome 1p + 4 (neo-Y chromosome) of male *M. crinifrons* using candidate male-specific mutations. The completeness of muntjac genomes was assessed using Benchmarking Universal Single-Copy Orthologs (mammalia_odb9)[59] and synteny analysis with *B. taurus* genome. The detailed information for samples collection, genome sequencing, and assembly is available in the Supplementary Notes. It is worth noted that the analysis of compartment A/B, TADs, and neo-sex chromosome include female and male *M. crinifrons* genomes, while in all other analysis, just female *M. crinifrons* is used to represent *M. crinifrons*, due to that the female *M. crinifrons* has higher assembly quality and simpler chromosome composition.

Animal care and experiments were conducted to the guidelines established by the Regulations for the Administration of Affairs Concerning Experimental Animals (Ministry of Science and Technology, China, 2013) and were approved by the Committee for Animal Experiments of the Institute of Zoology, Chinese Academy of Science, China (IOZ-IACUC-2021-145).

**Gene annotation.** To comprehensively annotated the protein-coding genes, we firstly detected and masked the repeats for genomes we assembled. Then, we used the combined pipeline of de novo and homology-based method to predict genes. Finally, we annotated the function of protein-coding genes by aligning their protein sequence to the SwissProt and KEGG database using BLAST software (version 2.2.26)[60]. The detailed steps for repeat annotation, gene model prediction, and gene function annotation can be found in Supplementary Notes.

**Phylogenetic tree construction and divergence time estimation.** The phylogenetic relationship of muntjac deer and outgroup species were reconstructed using the pipeline from the Ruminant Project[32]. First, genome of *B. taurus* (ARS-UCD1.2) was selected as reference, because it has a very high-quality chromosome-level genome and gene annotation results as one of the most important domestic species. The genomes of *Rangifer tarandus*[61], *E. davidianus*[62], *Cervus albirostris* and *M. muntjak vaginalis*[32], *H. inermis*, *M. reevesi*, *M. gongshanensis*, and female *M. crinifrons* were separately aligned to the reference genome using LAST software (version 885)[63]. Then the MULTIZ software (version 11.2)[64] was used to merge the pair-wised alignments into multiple genome alignments using *B. taurus* genome as the reference. The consensus coding sequences of each species were extracted based on the *B. taurus* genome annotation file using in-house Perl scripts. For each genome, a '-' symbol would be inserted if the site is not aligned to the reference genome. The orthologous genes with any species containing more than 20% '-' would be filtered out. Then we utilized an in-house Perl script to extract the 4dTV sites in the remaining orthologues for reconstructing the Maximum Likelihood (ML) phylogenetic tree using RAxML (GTRGAMMAI model) software (version 8.2.9)[65] with 200 bootstrap replicates.

In addition, we also constructed a phylogenetic tree using the mitochondrial genomes of the muntjac and outgroup species. The mitochondrial genomes of *H. inermis* (NC_011821.1) and *B. taurus* (NC_006853.1) were obtained from NCBI and the mitochondrial genomes of *R. tarandus*, *C. albirostris*, *M. reevesi*, *M. muntjak vaginalis*, and *M. crinifrons* were obtained from the Ruminant Project[32]. The mitochondrial genome of *M. gongshanensis* was assembled using MITObim software (version 1.8)[66] with the *M. crinifrons* mitochondrial genome as the reference. All these collected mitochondrial genomes were aligned by MUSCLE (version 3.8.31)[67] and ambiguous regions were excluded. The ML phylogenetic analyses were also conducted using RAxML (GTR + CAT model)[65] with 1000 bootstrap replicates.

We used the MCMCTREE in the PAML package (version 4.8)[68] and two nodes with calibrated divergence time to estimate the divergence times of muntjac species based on 4dTV sites. The calibrated time range of the common ancestor of *B. taurus* and Cervidae species (17.2~21.6 Mya) obtained from Heckeberg et al.[69] and the common ancestor of Cervidae species (9.8~17.3 Mya) was obtained from TimeTree[70].

**Demographic history reconstruction.** The Pairwise Sequentially Markovian Coalescence (PSMC) model (version 0.6.5-r67)[37] was used to infer the demographic history of muntjac species. Briefly, the Illumina reads from sample MRE1, MGO2, female MCR2, and 10× linked-reads data of a *H. inermis* sample downloaded from NCBI (PRJNA438286) were firstly aligned to their reference genomes using Bowtie2 (version 2.2.9)[71] with default parameters, respectively. Then, the SAMtools (version 1.3.1)[71] was used to convert the aligned results to sorted bam files (samtools view) and then estimate the genotype likelihoods with adjusted mapping quality larger than 50 (samtools mpileup). The genotype likelihoods data was fed to bcftools (version 1.3.1)[72] for identifying SNPs with default parameters. The outputted variant call format (VCF) files were used to conduct the PSMC analysis. The PSMC file of *M. muntjak vaginalis* and *C. albirostris* were obtained from Chen et al.[32]. The generation time of the muntjac species were set as 2.5[73] and the generation time of the two deer species (*H. inermis* and *C. albirostris*) were set as 5[32]. The neutral mutation rates (μ) of these species were estimated by r8s (version 1.70)[74].

**Detection of chromosome fusion events.** To obtain the chromosome rearrangement events among *B. taurus*, *H. inermis*, *M. reevesi*, female, and male *M. crinifrons*, we firstly performed the pair-wised alignment of these chromosome-level genomes using LAST software (version 885)[63]. Then we filtered out the short syntenic fragments using different cutoff values of 8 kb, 14 kb, 35 kb, 55 kb for *B. taurus* - *H. inermis*, *H. inermis* - *M. reevesi*, *M. reevesi* -female *M. crinifrons* and female and male *M. crinifrons*, respectively. We visualized these syntenic blocks using MCScan software (https://github.com/tanghaibao/jcvi/wiki/MCscan (Python-version)) and detected the chromosome fusion events among these muntjac species from the syntenic blocks plot.

The types of fusion events happened in female *M. crinifrons* genomes were further defined based on the synteny block plot and the position information of centromeres on chromosomes of *B. taurus*, *M. reevesi*, and *H. inermis*. All the chromosomes of *H. inermis*, some chromosomes of *M. reevesi* and *B. taurus* could represent the ancestral chromosomes. All the autosomes of *B. taurus* are acrocentric and thus their centromeres have been orientated at the 5' end of chromosome DNA sequences[75,76]. The centromere positions of *M. reevesi* and *H. inermis* were obtained by mapping the centromeric satellite sequences of Cervidae to the their genomes using BLAST software[60]. For those chromosomes that we could not map cervid centromeric satellite sequences, the centromere positions were determined based on the sequence homology with *B. taurus*.

Based on the synteny results, the comparative BAC maps between *M. reevesi* and other four muntjac species, including *M. crinifrons*[21], *M. muntjak vaginalis*[38], *M. gongshanensis* and *M. feae*[39] and the phylogenetic relationship of these five muntjac species[5], the fusion events of female *M. crinifrons* were further divided into five classes (C_1, C_2, C_3, C_4, and C_5). Fusion events belonging to class C_5 are shared by all five muntjac species and occurred at the earliest. The class C_4 contain fusion events shared by the rest four muntjac species except *M. reevesi*, the class C_3 are fusion events shared by *M. feae*, *M. gongshanensis*, and *M. crinifrons*, and the

class C_2 represents fusion events happened in *M. gongshanensis* and *M. crinifrons*. The class C_1 means fusion event only happened in *M. crinifrons*.

**Construction of normalized contact matrix.** The HiC-Pro pipeline (version 2.11.4)[77] was used to construct the contact matrices of *H. inermis*, *M. reevesi*, female and male *M. crinifrons* based on the high-quality Hi-C data and chromosome-level genomes. This pipeline firstly mapped the Hi-C reads to reference genome using Bowtie2 software[78]. Then the unmapped, singletons and multihits reads were filtered out. The remaining reads were assigned to the restriction fragments (restriction enzyme for *H. inermis* is Mbo I and restriction enzyme for female and male *M. crinifrons* and *M. reevesi* is Dpn II) and the valid reads pairs were obtained by filtering out the dangling end, self-circle, ligation, dumped pairs and PCR artefacts. The valid reads pairs were used to construct the raw contact matrices at different resolutions (1 Mb, 100 kb, 40 kb, and 20 kb). The raw contact matrices were then normalized using the iterative correction and eigenvector decomposition (ICE) method[79].

**Construction of homologous bin map.** The homologous relationship of bins between female *M. crinifrons* and other three genomes (*H. inermis*, male *M. crinifrons*, and *M. reevesi*) were obtained following the below steps. The three genomes were firstly and respectively aligned to the female *M. crinifrons* genome using LAST software[63], then the pair-wised alignment results were converted into "psl" and "chain" formats using "mafToPsl" and "axtChain" tools in UCSC genome browser[80]. To ensure that more bins (over 95% at 40 kb resolution) in female *M. crinifrons* genome have homologous bins in other three genomes, the liftOver tool[80] with different parameters (female *M. crinifrons* vs male *M. crinifrons* and female *M. crinifrons* vs *M. reevesi*: -minMatch=0.85, female *M. crinifrons* vs. *H. inermis*: -minMatch=0.7) was used to obtain the homologous bin pairs (Supplementary Fig. 25).

**Comparison of compartment A/B.** To compare the compartment A/B of different genomes, we firstly identified the compartment type of each bin in the four genomes (female and male *M. crinifrons*, *M. reevesi*, and *H. inermis*). Principal component analysis was conducted using cwold-dekker software with default parameters (https://github.com/dekkerlab/cworld-dekker/releases, v1.01) based on the ICE-normalized contact matrix at 100 kb resolution. Following many published studies[40,81–83], the first principal component (PC1) was used to identify compartment A/B. Positive or negative values of the PC1 separate chromatin regions into two spatially segregated compartments and regions with higher gene density and GC content were assigned as compartment A, while the rest were compartment B. The X chromosome was excluded in the subsequent comparative analysis because it affected the chromosome's 3D architecture by dosage compensation effect[57]. If three or more consecutive bins on female *M. crinifrons* genome exhibited different compartment type with their homologous bins in another genome, these bins are defined as compartment switched regions, while the remained bins are defined as the stable compartment type. The ratio of bins with switched or stable compartment type on whole-genome level were then counted. In order to check the effect of chromosome fusion on compartment A/B, the ratio of bins with switched or stable compartment type in the "near fusion site region" and "other region" were also counted, respectively. The 'near fusion site regions' are the regions, which included the fusion site regions and their 5 Mb upstream and downstream regions, and the 'other regions' are the remaining part of the genome but X chromosome.

**Comparison of TADs.** To compare the TADs of different species, we firstly used the insulation score method to detect the TAD boundaries for each genome based on the normalized contact matrix at 40 kb resolution[84]. Then based on the mapped bins of male *M. crinifrons*, *H. inermis*, and *M. reevesi* with female *M. crinifrons*, we used the bedtools (v2.25.0)[85] to obtain the overlapped regions between TADs of different genomes. Following the previous practice[41], the conserved or shared TADs are defined as two TADs in different genomes whose overlapped regions cover more than 70% of their respective lengths. We excluded missing bins when calculating the proportion of overlap interval. Because our liftover parameters reduce the proportion of total missing bins at the 40 kb resolution to be less than 5% (Supplementary Fig. 25) and most TADs actually have only 0~2 missing bin (Supplementary Fig. 26), the TAD comparison between genomes would not be affected. The *M. crinifrons*-specific TADs are defined as the TADs of female *M. crinifrons* which are shared with male *M. crinifrons* but are not shared with *H. inermis* and *M. reevesi*. The distance of bins in *M. crinifrons*-specific TADs from their nearest fusion sites were calculated and then counted using the "geom_density" function in the ggplot2 package[86].

**Comparison of significant interaction.** To compare the significant interactions between different muntjac species, we first used the Fithic software (version 1.1.3)[87] to get the significant interactions (*p* value <0.01 and *q* value < 0.01) which are supported by more than nine valid read-pairs based on the normalized interaction matrix at 20 kb resolution. Then we used in-house scripts to examine the length distribution of significant interaction among genomes. We transformed the length of intrachromosome significant interactions by $\log_{10}$ and then calculated the total number of intrachromosome significant interactions in sliding windows with

length 0.1. We used the min-max normalization method to normalize the number of intrachromosome significant interactions within each window.

For the long-range significant interaction with length more than 5 Mb in female *M. crinifrons* genome, we divided them into four classes according the compartment type at their two ends. 'A-A' means that the both ends of a long-range significant interaction are compartment A, 'B-B' shows that the both ends are compartment B, 'A-B' indicates that one end is compartment A and another is compartment B; 'others' indicates that at least one end has no explicit compartment type. The number and proportion of long-range significant interaction belonging to different classes were also obtained using in-house scripts. In addition, we used an in-house script to obtain the newly established long-range significant interaction in female *M. crinifrons* following the below steps. All significant interactions of *M. reevesi* were mapped to the female *M. crinifrons* genome according the homologous bin map. If the start-end bins of one long-range significant interaction are homologous with the corresponding bins of *M. reevesi*'s significant interactions or adjacent one upstream or downstream bin, this long-range significant interaction of female *M. crinifrons* is considered as the shared with *M. reevesi*. The remaining long-range significant interaction that does not share with *M. reevesi* are considered as the newly established long-range significant interaction in female *M. crinifrons*. We also checked the number of long-range significant interaction across fusion sites.

Next, we focused on the significant interactions across fusion sites in the female *M. crinifrons* genome. We obtained the newly established significant interactions across fusion sites using the above strategies. In addition, for the significant interactions across fusion sites just spanning one fusion site, we checked the class of fusion site (C_1~C_5). The average values of significant interactions across each class fusion sites were calculated and used for correlation analysis with the ages of fusion sites using Pearson method. Furthermore, we collected the genes occupying more than half length of the bins at two ends of significant interactions across fusion sites spanning fusion sites belonging to C_1 or C_2 and conducted the GO and pathway enrichment analysis using metascape (version 3.5)[88].

Finally, we compared the contact pattern of *M. reevesi* and female *M. crinifrons* near the fusion site of female *M. crinifrons*. The contact matrix of *M. reevesi* at 20 kb resolution was first transformed onto the female *M. crinifrons* genome coordinate based on their bin's homologous relationship using in-house scripts. Then half of the transformed contact matrix of *M. reevesi* and half of the contact matrix of female *M. crinifrons* at the same resolution were combined for visualization using an in-house python script.

**Comparison of 3D genome structure.** To compare 3D genome structure of different species, we firstly used Pastis software (version 0.1.0)[89] to predict the 3D genome structure based on the normalized matrix at 1 Mb resolution. Then we used the PyMOL tool[90] to plot the 3D genome structure of different species based on the "pdb" files generated by Pastis. The homologous chromosomal segments among different genomes were showed in the same color.

**Content of satellite and telomeric sequences in Nanopore reads.** We explored potential clues in the Nanopore reads from fusion sites. The monomers of three classes satellite sequences (satI, satII and satIV) identified in the genome of female *M. crinifrons* (Supplementary Notes) and vertebrate telomeric sequences ((TTAGGG)n) were aligned to these nanopore reads of female *M. crinifrons*, *M. gongshanensis*, *M. reevesi* and *E. davidianus* using BLAST (-evalue 0.001)[60], respectively. Then we used in-house scripts to divide these reads into different patterns and counted their number following the below steps. We first calculated the length of satellite sequence and telomeric sequence contained in these nanopore reads. A Nanopore read is considered to contain certain class of satellite sequence if the total length of this satellite sequence in it is more than 30 bp. For the convenience of recording, we used four-bit binary number to represent different patterns. From left to right, each bit represents satellite I, satellite II, satellite IV, and telomeric sequence, respectively. When a read contains a certain type of satellite sequence or telomeric sequence, the corresponding position is set to 1, otherwise it is 0. For example, pattern 1011 indicates that this read synchronously contains satellite I, satellite IV, and telomeric sequence. Then we counted the number and proportion of Nanopore reads for each pattern.

In particular, we checked the distance between satellite I and telomeric sequence in reads just contain satellite I and telomeric sequences (pattern 1001) using an in-house python script. We divided these reads into two categories and counted their numbers, respectively. The distance between satellite I and telomeric sequence is greater than or equal to 500 bp in one kind of reads and less than 500 bp in another kind of reads. In addition, we detected palindromic sequence in reads containing satellite sequence and telomeric sequence by performing self-BLAST[60].

**Positively selected and rapidly evolving genes and smaller-scale genome rearrangements.** To identify the positively selected genes (PSGs) and rapidly evolving genes (REGs) in muntjac species, we firstly extracted the orthologous gene set from the multiple genome alignments of *B. taurus* and other ten species, including *Homo sapiens* (GRCh38.p12), *Equus caballus* (EquCab3.0), *Camelus ferus* (Ca_bactrianus_MBC_1.0), *R. tarandus*[61], *C. albirostris*, and *M. muntjak vaginalis*[32], *H. inermis*, *M. reevesi*, *M. gongshanensis*, and female *M. crinifrons*. We filtered the orthologous gene set following

the criteria used in the ruminant project[32]. To identify the PSGs, we firstly estimated the lineage-specific evolutionary rates for each ortholog using the Codeml module with a free-ratio model (model = 1) in the PAML package[68]. Then we used the branch-site model (model=2) to identify genes having a higher $\omega$ ($dN/dS$) than the rest of the lineage in the tree. A likelihood ratio test (LRT) was conducted to compare the above two models. The $p$-value (chi-square statistics) and false discovery rates (FDR) for each gene were calculated. There were only a few genes with FDR less than 0.05. Considering that the positive selection results would be robust when synonymous substitutions are far from saturation[91] and mean $dS$ (synonymous mutation rate) between muntjacs and other deer ranges from 0 to 0.3 (Supplementary Fig. 27), which is far from synonymous substitution saturation, we finally selected the genes with $p$ value less than 0.05 as PSGs to comprehensively explore the possible genetic mutations relevant to muntjac chromosome evolution (Supplementary Table 18). To identify the rapidly evolving genes (REGs), we tested the same orthologous gene set using branch model with null model (model = 0) assuming that all branches have evolved at the same rate and an alternative model which allows foreground branch evolves at a different rate (model = 2). Similarity, the LRT was used to compare the two models and $p$ value (chi-square statistics) and false discovery rates (FDR) of each gene were also obtained. Again, because there were too few genes with FDR less than 0.05 (Supplementary Table 18), we treated the genes with $p$ value less than 0.05 as REGs. We identified rapidly evolving genes (REGs) and positively selected genes (PSGs) in the $M.$ $crinifrons$, $M.$ $gongshanensis$, and $M.$ $muntjak$ $vaginalis$ with large fused chromosomes, as well as in their common ancestor node. The Metascape[88] was used to perform KEGG pathway, Reactome pathway, and Gene Ontology (GO) enrichment analysis.

In addition, we detected genomic rearrangement events (>10 kb) happened in $H.$ $inermis$, $E.$ $davidianus$, $M.$ $reevesi$, $M.$ $gongshanensis$, female, and male $M.$ $crinifrons$ compared with the $B.$ $taurus$ genome using the method from Chen et al.[32]. The rearrangement events just include inter-chromosome translocation, intrachromosome translocation, and inversion. Density of genomic rearrangement was calculated by dividing the genome size into genomic rearrangement number.

**Detection of neo-X and neo-Y regions**. To identify the neo-X regions in the female $M.$ $crinifrons$ genome and neo-Y regions in the male $M.$ $crinifrons$ genome, we firstly aligned the genome of female $M.$ $crinifrons$ to the male genome using the LAST software[63]. The inverted region on chromosome 1p + 4 of male $M.$ $crinifrons$ is defined as neo-Y regions[22], and their homologous regions in female genome as neo-X regions (Fig. 1d). The coordination of neo-Y and neo-X regions were figured out using in-house scripts.

**Mapping coverage**. To check the sequence degeneration degree of neo-Y regions, we compared the mapping coverage of Illumina reads from male and female $M.$ $crinifrons$ individuals on female $M.$ $crinifrons$ genome. We firstly aligned the Illumina reads from the individual female MCR2 and male MCR2 to the reference genome of female $M.$ $crinifrons$ using bwa software (version 0.7.12)[92], respectively. Then, the igvtools (version 2.3.75)[93] was used to calculate the mapping coverage with a 500 kb sliding window. The value of mapping coverage was normalized using the min-max normalization method. The variable is the mapping coverage value of 500 kb window, min is the minimum coverage value after transformed by $\log_2$ function and the max is the average sequencing depth on the whole-genome after transformed by $\log_2$ function.

**Identification and annotation of candidate male-specific mutations**. We separately detected the candidate male-specific SNPs and indels using the Illumina reads and structural variations (SVs) using the PacBio reads. Firstly, to detect the candidate male-specific SNPs and indels, we aligned the Illumina reads from four $M.$ $crinifrons$ individuals (female MCR2 and MCR3, male MCR2 and MCR3) and two $M.$ $gongshanensis$ individuals (MGO1 and MGO2) to the female $M.$ $crinifrons$ genome using bwa software[92]. Then the SNP and indels were called using GATK (version 4.1.2.0) HaplotypeCaller model[94] and filtered using the criterion of DP > = 15 and DP < = 100. An in-house script was used to identify the candidate male-specific SNP or indel site where it is heterozygous in the two male $M.$ $crinifrons$ samples, but is homozygous and the same with the reference allele in the two female $M.$ $crinifrons$ samples and the two $M.$ $gongshanensis$ samples. The density of these candidate male-specific SNPs and indels were calculated using vcftools (version 0.1.13)[95] with a 500 kb sliding window.

Secondly, to detect the candidate male-specific SVs, we mapped the PacBio reads from the male $M.$ $crinifrons$ sample (PRJNA438286) to the female $M.$ $crinifrons$ genome using bwa software[92]. Then we called the SVs using sniffles software (version 1.0.12)[96] and selected the heterozygous SVs as candidates. Finally, we used the flowing steps integrated in in-house scripts to filter the candidates. We first obtained the name of reads supporting heterozygous SVs from the outputted VCF file and used the picard software (FilterSamReads) to extract the alignments results of these reads. Then these extracted alignment results were used to call SNP using samtools[71] and bcftools (version 1.3.1)[72]. The SNPs which are the same to the above candidate male-specific SNPs, which identified by Illumina reads are regarded as the reliable SNPs. If there are at least one reliable SNP within 1 kb upstream and downstream of the candidate SVs, these SVs were regarded as conjectural male-specific SVs. The density of these candidate male-specific SVs was also calculated using vcftools[95] with a 500 kb sliding window.

After merged the candidate male-specific SNPs, indels and SVs, we used the SnpEff software (version 4.10)[97] to predict the impact of these male-specific mutations on the protein-coding genes. Four types of effects were recognized for the mutations. The mutations marked as "HIGH" disrupt the ORF of proteins. The mutations marked as "MODERATE" are amino acid substitution mutations. The mutation marked as "LOW" indicates that this mutation is synonymous. The mutation marked as "MODIFIER" represents variants in non-protein-coding regions or genes.

**Curation of male $M.$ $crinifrons$ neo-Y assembly using conjectural male-specific mutations**. Due to the insufficient differentiation between the neo-X and neo-Y chromosome, the neo-Y (1p + 4) chromosome assembled by PacBio reads and Hi-C data may be blended with sequence from neo-X chromosome. Therefore, to accurately get the neo-Y chromosome assembly, we aligned the primarily assembled neo-Y chromosome of male $M.$ $crinifrons$ to the homologue regions in female $M.$ $crinifrons$, including part of the chromosome neo-X (X + 4) and short arm of chromosome 1 (1p), using the LAST software[63]. Then we used in-house scripts to correct the neo-Y chromosome assembly according to the following method. For the candidate male-specific mutation sites identified above, if the base on neo-X chromosome is the same with that on neo-Y chromosome, the base on neo-Y were replaced to the alteration base of candidate male-specific mutations there. In this study, we used the curated neo-Y chromosome in all the analysis related to male $M.$ $crinifrons$ genome.

**Gene and allele expression levels in the neo-Y regions**. We used the hisat2 software (version 2.1.0)[98] to map the RNA-seq reads from a female sample (female MCR4) and a male sample (male MCR4) to the female $M.$ $crinifrons$ reference genome with the default parameters. The alignment result files were transformed to BAM format using the samtools[71]. Then we used the featureCount software[99] to count the reads number which mapped to the gene, and the DESeq2 software (version 1.20.0)[100] to identify the genes which differently expressed between female and male samples ($p$ value < 0.05).

Based on the above candidate male-specific SNPs and the alignment results of RNA-seq reads from male sample to the genome of female $M.$ $crinifrons$, we used an in-house perl script to distinguish the number of RNA-seq reads supporting neo-X and neo-Y alleles. If at least two candidate male-specific SNPs are located in a gene and the reads' number supporting neo-X and neo-Y alleles are significantly different (paired samples $t$ test, $p$ value <0.05), this gene was regarded as having differently expressed allele.

**Reporting Summary**. Further information on research design is available in the Nature Research Reporting Summary linked to this article.

## Data availability
The data that support this study are available from the corresponding authors upon reasonable request. The sequencing data generated in this study have been deposited in the NCBI Sequence Read Archive under accession number PRJNA640966. The PacBio data of male $M.$ $crinifrons$ used in this study are available in the NCBI Sequence Read Archive under accession number PRJNA438286. Source data are provided with this paper.

## Code availability
Customized codes used in this study have been deposited in GitHub (https://github.com/YinYuan-001/muntjac_code) and are also publicly available in the Zenodo repository (https://doi.org/10.5281/zenodo.5533097).

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

## Acknowledgements

We thank J. Liu and Y. Sun from the Beijing Institute of Genomics, Chinese Academy of Sciences, Beijing, China for their grateful help in 3D chromatin architecture analysis. This study was supported by the National Natural Science Foundation of China (31821001), the Strategic Priority Research Program of Chinese Academy of Sciences (XDB31000000), the Qianjiangyuan National Park, the Biodiversity Survey, Monitoring and Assessment Project of Ministry of Ecology and Environment of China (2019HB2096001006) to F.W.; the National Natural Science Foundation of China (32030016) and the Fundamental Research Funds for the Central Universities (3102019JC007) to W.W.; the National Natural Science Foundation of China (32170415, 32061130208), the Natural Science Foundation of Zhejiang Province (LD19C190001) and the European Research Council Starting Grant (grant agreement 677696) to Q. Z.

## Author contributions

F.W., W.W., and Q.Z. conceived the project and designed research. Y.Yin. H.F. and W.W. drafted the manuscripts. W.W., H.F., Q.Z., F.W. Y.H., G.H., J.Y., and L.C. revised the manuscript. H.F., G.F., J.W., W.N., C.Z., W.Z., G.L., B.W.Z., J.C., H.Q., Z.Y.Z., and Z.P.L. prepared the samples. Y.Y., B.T.Z., F.Z., L.L., Y.Z., C.Z., Z.S.L., C.L., J.Z., Z.H.L., Y. Y., B.Z.Z., Y.P., and X.C. performed the data analysis.

## Competing interests

The authors declare no competing interests.

## Additional information

¹School of Ecology and Environment, Northwestern Polytechnical University, Xi'an 710072, China. ²CAS Key Laboratory of Animal Ecology and Conservation Biology, Institute of Zoology, Chinese Academy of Sciences, Beijing 100101, China. ³Center of Evolution and Conservation Biology, Southern Marine Science and Engineering Guangdong Laboratory (Guangzhou), Guangdong 511458, China. ⁴Center for Excellence in Animal Evolution and Genetics, Chinese Academy of Sciences, Kunming 650223, China. ⁵BGI-Qingdao, BGI-Shenzhen, Qingdao 266555, China. ⁶BGI-Shenzhen, Shenzhen 518083, China. ⁷China National GeneBank, BGI-Shenzhen, Shenzhen 518120, China. ⁸Kunming Cell Bank, State Key Laboratory of Genetic Resources and Evolution, Kunming Institute of Zoology, Chinese Academy of Sciences, Kunming 650223, China. ⁹Frasergen Bioinformatics Co., Ltd, Wuhan 430074, China. ¹⁰Beijing Milu Ecological Research Center, Bejing 100076, China. ¹¹Qianjiangyuan National Park, Kaihua 324300, China. ¹²Department of Biotechnology and Biomedicine, Technical University of Denmark, 2800 Lyngby, Denmark. ¹³School of Life Sciences, Anhui University, Hefei 230039, China. ¹⁴State Key Laboratory of Environmental Criteria and Risk Assessment, Chinese Research Academy of Environmental Sciences, Beijing 100012, China. ¹⁵Colleague of Animal science and technology, Jilin Agricultural University, Changchun 130118, China. ¹⁶Institute of Special Animal and Plant Sciences, Chinese Academy of Agricultural Sciences, Changchun, China. ¹⁷MOE

Laboratory of Biosystems Homeostasis & Protection and Zhejiang Provincial Key Laboratory for Cancer Molecular Cell Biology, Life Sciences Institute, Zhejiang University, Hangzhou 310058, China. [18]Department of Neuroscience and Developmental Biology, University of Vienna, Vienna 1090, Austria. [19]Center for Reproductive Medicine, The 2nd Affiliated Hospital, School of Medicine, Zhejiang University, Hangzhou 310052, China. [20]These authors contributed equally: Yuan Yin, Huizhong Fan, Botong Zhou, Yibo Hu, Guangyi Fan, Jinhuan Wang, Fan Zhou, Wenhui Nie. ✉email: zhouqi1982@zju.edu.cn; wenwang@nwpu.edu.cn; weifw@ioz.ac.cn

