## [Peer Review File · Nature Communications]

REVIEWER COMMENTS

Reviewer #1 (Remarks to the Author):

The manuscript "Molecular mechanisms and topological consequences of drastic chromosomal rearrangements of muntjac deer" employs a variety of methods to examine the evolution of genomic architecture in the muntjac deer. The authors have been meticulous in the analyses from assembly to the identification of chromosome fusions. The balance between the sections in the paper is satisfactory. I have two major comments that I would like to see addressed and a few minor comments. Overall I believe this manuscript is very strong and contributes with significant knowledge on the evolution of mammalian genomes.

Major comments:

Reproducibility: Several of the analyses described in the methods are not reproducible. This makes it difficult to verify some analyses. In addition some in-house scripts have been used (e.g. line: 430). These should ideally be publicly available.

A/B compartment analyses: This paper contrasts the findings in Mudd et al. I would like this part to be extended in terms of analyses. In particular the 'Comparison of compartment A/B' section could be extended. E.g. On what basis did you choose to only look at PC1, What is the sensitivity and specificity of the designation of the compartments using gene density and GC content.

Minor:

Supp. Table 1: BMF4 (F stands for female?) is reported as a male?

Line 143: BMM (N50 contig length) is 3.7mb according to Supp. table 5.

Figure 1a: Placement of 6 tandem fusions?

Since milu deer and white-lipped deer have 68 and 66 some of the tandem fusions could as well have been placed before the split of the muntjacs and these two samples?

In addition I suggest adding all fission and fusions to the figure just like in Extended data fig2 e.

Line 185: Think it is an important result but circos plot, albeit pretty, does not convey the message very well to me

Line 188 and ext. fig. 3a: Why is CWD so rich on trans read pairs compared to the muntjacs?

Line 365: The comparison to the Mudd et al could be expanded (see major comment).

Line 457: What is the genotype likelihood data used for?

Line 463: What estimates of mutation rate did the authors obtain using r8s?

Line 511: Please provide information on how you decided on the 0.85 and 0.7 minimum match?

Line 515: Please specify the software and parameters used. This could generally be improved in the method section.

Line 515: Why is only a single PC for the A/B compartment?

Line 619: Given that the sample size for males is two, how often would you expect that designated male-specific SNPs happen by chance?

Line 698: Did the authors correct for multiple testing both in PSG and REG?

Reviewer #2 (Remarks to the Author):

In this paper, the authors investigate chromosomal rearrangements in Muntjac deer, a clade with extensive karyotype variation. The authors generate chromosome-level genome assemblies to characterize how chromosome fusions have contributed to genome evolution among species. They then investigate potential demographic and molecular drivers of these fusions, and their effects on chromatin architecture. The authors also leverage these data to investigate the neo-sex chromosomes of Muntjac deer, revealing how chromatin architecture is evolving in sex-linked genomic regions. Overall, I think the findings will be of interest to the community. However, some of the text and figure panels are difficult to parse. Below I give my comments and recommendations for how the manuscript could be strengthened.

Introduction: More context could be given for the study, both in terms of karyotype diversity across mammals, and in terms of molecular/evolutionary drivers of karyotype variation. For example, how much variation exists among mammals in chromosome number? Or at lines 99-101, what are some hypothetical mechanisms that could underlie fusions in this group?

There could also be more context presented for sex chromosome evolution in Black Muntjac, for example, what sex chromosome system does this species have and how does this compare to other taxa in the group?

L197-201/Fig 2b: Can you use RNAseq data to look at gene expression/co-expression around fusion sites before and after fusions have occurred?

L208-211/Fig 2c: It would be helpful to label the figure so it matches with the percentages described in the main text (95.27%/44.37%).

L212-214/Fig 2d: I found this difficult to follow. I'm not sure what is meant by the 'third or farther ancestral chromosomes'.

L217-219/Fig 2e: I also found this difficult to follow. Please edit this section and the figure to more clearly communicate the results.

Fig 3a: It's difficult to see the coverage, SNPs, etc in the circular plot. It would be helpful to see the sex-linked regions in their own panel.

Fig 3b: Are these inversions contributing to the formation of strata on the sex chromosomes?

Fig 3c/d: Is there any relationship between compartment switching and certain GOs or pathways?

Fig 3c/d: How many genes are included in these analyses?

L366-368: The same sentence is repeated at the end of the discussion (L393-394).

General: The final section of the results (molecular causes of fusions) would go better before the section on sex chromosome evolution. I suggest re-ordering the results in this way.

Discussion: The results show that chromosome fusions have a greater impact on TADs than on compartments, while recombination suppression among the sex chromosomes seems to drive more changes in compartment than in TADs. Could the authors comment further on these differences and

their implications?

There are a number of grammatical errors throughout the text that make some parts difficult to read. I recommend revising these for clarity; a few are highlighted below:

L209: should be 'almost all of them'

L210: should be 'this is in contrast to the rest of the significant interactions'

L223: should be 'may be of biological significance'

L292: should be 'we searched for evidence'

Reviewer #3 (Remarks to the Author):

Review of Yin et al.

Molecular mechanisms and topological consequences of drastic chromosomal rearrangements of muntjac deer

The authors have addressed one of the most interesting questions in mammalian chromosome evolution, i.e., how did the chromosomes of some but not all species of muntjacs become reduced to a small number of giant chromosomes in a relatively short time span of 1-2 million years. Although it has been known from FISH and other comparative studies that the giant chromosomes were formed largely by fusions and Robertsonian translocations, the mechanisms by which this occurs and the consequences of the rearrangements on the three dimensional structure in the nucleus have not been determined. The authors conducted comparative and evolutionary whole-genome analysis using de novo chromosome-scale assemblies and long read data to identify the possible mechanisms by which the fusions have occurred in the muntjac lineages that exhibit reduced diploid numbers. Deep coverage Hi-C data was used to explore how such rearrangements affected the definition of A/B chromatin compartments and TADs, as well as long range interactions between non-homologous chromosomes. Additionally, an interesting evolutionary analysis of the neo-X chromosome and the neo-Y was conducted.

General Comments

The authors are commended for preparing a well-written manuscript, although it needs thorough editing for grammar and spelling. Another issue is the excessive use of non-standard abbreviations that make the manuscript very difficult to read. Examples include species names, LRSI, and SIAF.

Genome assemblies and phylogenetic analysis are presented with an overlay of the demographic history of deer in the Far East. Based on genomic analysis the authors have concluded that Indian Muntjac, Black muntjac and Gongshan muntjac underwent a dramatic reduction in population size 1-2 Mya near the time of their divergence, and correlated these findings with the Xixiabangma glaciation event. The authors proposed that genetic drift and reproductive isolation might have contributed to the fixation of the dramatic chromosomal rearrangements found in these three species of muntjacs, but not Chinese muntjacs, which diverged from a common ancestor about 4.7 Mya. This conclusion seems reasonably well supported by the data.

One of the main conclusions of the paper is grossly overstated, e.g., "These results resolve the long-standing mystery underpinning the 79 recurrent chromosome fusions and reveal how chromosome rearrangements" The results do not "resolve" the mystery of precisely how the reduction in genome size occurred (see also line 281 section title "molecular cause of chromosome fusions in muntjac

species"). While the authors preferred this interpretation, cause and effect was not demonstrated (see further discussion below). The alternative hypothesis that other genomic mutations could have led to the chromosome reductions was not ruled out by the data, which also showed accelerated evolution and positive selection in genes that could be part of the compensatory mechanism for very large chromosome size. While it is recognized that cause and effect are difficult (but not impossible) to determine, this does not give the authors license to conclude that the mystery has been solved. For example, experimental systems with artificial chromosomes containing the complex repeat could be used to demonstrate chromosome instability (e.g., promotion of fusions) in vitro, as has been done for other similar systems.

The issue of compensatory mechanisms among the positively selected genes is an interesting one. Are there any data to support that the muntjac mutations are loss/gain-of-function mutations in any other species? Have knock-out, knock-ins or CRISPR mutations in these genes in mice been shown to affect chromosome stability during mitosis or meiosis?

The conclusion that truncated telomeric repeats are related to the fusion of muntjac chromosomes may not be supported by the data presented in the manuscript. Firstly, the genomes analyzed were not assembled across the fusion sites except for one fusion site in black muntjacs. To compensate for this, the authors had to use long ONT reads that "possibly" spanned the fusion sites (line 293). This approach is very different than directly analyzing a complete assembly where the length of the specific repeats can be accurately measured. Complete assembly across the fusion sites, admittedly being difficult, is really the only way to support the authors' conclusions about the length differences between species. The authors identified what they believe to be truncated telomeric sequences to support their claim. Can the authors be sure that these truncated reads are not due to problems with ONT reads? Long ONT reads are susceptible to termination due to secondary structures (hairpins, etc.), that are common when the single strands contain repetitive elements. Can the authors rule this out?

A second major conclusion of the paper is that A/B compartments are minimally affected by rearrangements, but that TADs are affected around the rearrangement sites. This is probably the most interesting and important finding of the paper. However, there appear to be methodological issues that require further explanation to support these conclusions.

As mentioned above, the genomes used were not fully assembled around the fusion sites. Given this issue, how can the authors be sure that the liftover coordinates on the black muntjac genome are not causing a problem with the definition of TAD boundaries? Are there any gaps or Ns added to these regions in any of the assemblies that might change the boundaries?

Relatedly, is the 70% of TAD length sufficient to call TAD boundaries around breakpoint and/or fusion sites? In the paper cited (Dixon et al., 2012) the comparison between human and mouse TADs is done using the exact human and mouse coordinates to determine the boundaries within syntenic regions. The authors did not use custom parameters for their analysis. The default parameter for the UCSC liftover tool for identity percentage is 0.95. That is the likely reason why the percentage of shared TADs between human and mouse was low. Using TADs overlap of 70% does not appear to be justified given the recent divergence time and would not appear to give precise coordinates for the comparison of fusion points in the different species.

Specific comments

L138 p7: "The assembled 4, 5, 23..." sentence not clear. Please rephrase.

L183 p9: Extended data Fig. 2f only show the percentage of the genomes that were aligned between the species. To support the claim that the rapid karyotype evolution in muntjacs is not accompanied by rapid sequence evolution, the authors should calculate the substitution rates for the species and

compare with other existing data or at least report the average sequence identity across the alignment.

L190 p10: What is the percentage of recovered bins for each species after the lift-over?

L199 p10. Related to the previous comment, did the authors underestimate the number of fusion related TADs in non-fused chromosomes because of possible missing bins in other species due to the lift-over?

L195 p10: It might be expected that a higher fraction of TADs would be shared between the muntjacs as the divergence times are 3 Mya maximum, when compared to human mouse that diverged 90 Mya. Can this difference be due to the methodological issues described above?

L202 p10: Can the length of LRSI observed be due to chromosome length? Were more LRSI in black muntjac observed because they have the longest chromosomes? Also, are inter-chromosomal LRSI in Chinese muntjac (and other species with higher chromosome number) more frequent between chromosomes that are fused in black muntjac? To help resolve these issues, it would be helpful to determine if the fused chromosomes are physically closer in the nucleus.

L209 p11: Do authors think that these newly established interactions (LRSI) are formed de novo and are completely absent in the non-fused chromosomes?

L211 p11: The authors did not mention in the methods if they compared black muntjac SIAFS with other species inter-chromosomal SIs. This should be done as black muntjac is the species with the lowest chromosome number and most of these interactions would be between chromosomes in the other species. Are the shared SIAFS between black muntjacs and Chinese muntjacs also shared fusions?

L308 p15: Where in the sequencing read (end or middle) are the truncated telomeric repeats? Could these be just an artifact of the sequencing technology (see comments above)?

L331 p16: The figure cited appears to be 4e instead of 2e.

L505 p24: Related to the first comment. Why did authors use a different minimum identity percentage (-minMatch) as threshold for different liftovers?

L512 p26: It is unclear if the same tissues were used for generating Hi-C data in the different species. This should be clarified. Why did the authors choose to use GC content and gene density to define A/B compartments? Would it not be better to use gene expression data?

Reviewer #4 (Remarks to the Author):

The muntjacs genus perhaps is the most spectacular example of rapid karyotype evolution. The authors produced the chromosome-scale reference genomes from three muntjac deer species and one outgroup, the Chinese water deer. These genomes are at high quality therefore allowing the authors to produce an in-depth view on the chromosome evolution process within muntjacs genus and the accompanying chromatic structures. They also proposed some mechanisms underlying the rapid karyotype evolution in this genus in association with repeat contents and some rapid evolutionary genes related with the genomic architecture stability.

Overall, I found this study was an exciting one with many interesting findings. I however have few

comments for the authors to consider for their revision.

1. It is unclear to me the rationality of selecting these species, particularly, why the Chinese water deer from another clade was used as outgroup and cattle as the reference for these very recent evolutionary events. Some explanations in the introduction would be useful for the readers to understand the design of this project.
2. The Indian muntjac was completely ignored in all the chromosome analyses. This is surprising as this species has been substantially studied before particularly with many chromosome painting experiments reported before. Even though it was not sequenced long reads, its genome assembly was done in chromosome level which is even better than the Gongshan muntjac that the authors reported here. Including the Indian muntjac genome would be more interesting than the Gongshan muntjac in telling the evolution of neo-XY in this genus and the genomic features near the fusion point. It is unfortunately that many questions regarding to the chromosome evolution in this genus have left behind without the Indian muntjac.
3. It is also lack of explanation for including the Gongshan muntjac. What conclusion was made from the comparison between Gongshan muntjac and black muntjac?
4. The authors claimed that the population decline of the species with reduced chromosome numbers in around 1million years ago might lead to the quick fixation of reduced karyotypes in population by genetic drift. This is a highly speculative claim as first, the demographic pattern produced by PSMC in such small time windows at 1million years ago is less reliable. And second, it is hardly to claim the N_e at $10^{-20} \times 10^4$ is a small one.
5. Line 174, It would be useful to explain what tandem fusion and Robertsonian fusion are in the main text.
6. Line 184-5, extended data Fig 2f only shows the synteny information but not the sequence identity, please provide the substitution rate in each node to confirm this claim.
7. Line 201-202, BM-specific TADs are more frequently located around the fusion sites of BMF, what is the control in this analysis?
8. Why BMF is used as representative of black muntjac in chromatic structure analyses?
9. Line 254-255, was this comparison done on whole genome level or only on neo-Y regions?
10. Line 266-267, what is the definition of differently express between neo-X and neo-Y alleles? The table does not include this information.
11. line 270, change 'conserved' to 'similar'
12. The figure legend needs to be improved by adding more detail explanation on all elements on the plots. Like Figure 1a, there is no explanation about the arrow in the plot. What is the meaning of '1p+4' in figure 1d. Figure 2b, what does the box plot tell? I can only say some examples here, but the figure legends for almost all figures (including the extended ones) need to be improved.

Response to reviewers:

Reviewer #1 (Remarks to the Author)

The manuscript "Molecular mechanisms and topological consequences of drastic chromosomal rearrangements of muntjac deer" employs a variety of methods to examine the evolution of genomic architecture in the muntjac deer. The authors have been meticulous in the analyses from assembly to the identification of chromosome fusions. The balance between the sections in the paper is satisfactory. I have two major comments that I would like to see addressed and a few minor comments. Overall I believe this manuscript is very strong and contributes with significant knowledge on the evolution of mammalian genomes.

Response: Thank you very much for your positive comments.

Major comments:

1. Reproducibility: Several of the analyses described in the methods are not reproducible. This makes it difficult to verify some analyses. In addition some in-house scripts have been used (e.g. line: 430). These should ideally be publicly available.

Response: Thank you for your suggestion. We have rewritten the methods part and added the detailed descriptions for the methods used for the analyses in the revised manuscript. In addition, we also uploaded all the pipelines and in-house scripts to the github database with the accessible link https://github.com/YinYuan-001/muntjac_code.

2. A/B compartment analyses: This paper contrasts the findings in Mudd et al. I would like this part to be extended in terms of analyses. In particular the 'Comparison of compartment A/B' section could be extended. E.g. On what basis did you choose to only look at PC1, What is the sensitivity and specificity of the designation of the compartments using gene density and GC content.

Response: Thank you for your comments. We have extended the descriptions about the comparison between our results and the findings in Mudd et al. in the revised manuscript as below:

“We downloaded their data and conducted careful comparison and found their analyses and conclusions had problems. Firstly, the quality of their genomes and the amount of Hi-C data are limited. The contig N50 lengths of muntjac genomes we assembled (3.79~37.86 Mb) are 15~150 folds higher than that of *M. reevesi* and *M. muntjac* genome (~200 kb) assembled by Mudd et al. (Mudd et al. 2020). We used about 100 folds more Hi-C data (264~328×) than Mudd et al. (~30×) to analysis the 3D chromatin architecture, making the resolution reach 20 kb, which is sufficient to analyze 3D chromatin architectures at different hierarchical levels. The amount of Hi-C data sequenced by Mudd et al only guarantee 1 Mb resolution which is not suitable for analyzing local compartment

(Dong et al. 2017; Du et al. 2020; Rowley et al. 2017; Wang, Wang, et al. 2019; Rao et al. 2014). Secondly, the method Mudd et al used to detect and compare the compartment of *M. reevesi* and *M. muntjac* is even more problematic. They mapped the Hi-C reads from *M. reevesi* to *M. muntjac* genome diverging ~3 Mya, which caused the reduction of half of the mapping rate (26.63%) compared with aligning *M. reevesi*'s Hi-C reads to its own genome (59.373%) (Supplementary Table 17). We mapped the Hi-C reads of *M. muntjac* and *M. reevesi* from Mudd et al. to their own reference genomes and used all of the interaction information on whole chromosome to identify the global compartments as regular practice in Hi-C compartment analysis (Lieberman-Aiden et al. 2009). A similar result to our comparison results between *M. crinifrons* and *H. inermis* or *M. reevesi* was obtained, namely, the compartment structure is conserved between *M. muntjac* and *M. reevesi* (Supplementary Fig. 4).” (Line 479)

We chose to look at only PC1 is based on the method by which A/B compartment was first identified by Lieberman-Aiden et al. (Lieberman-Aiden et al. 2009). They found a genome can be partitioned into two spatial compartments by PC1 such that interaction occurs within each compartment rather than across compartments. Most subsequent published studies on chromatin architecture followed this practice to identify compartment A/B (Barutcu et al. 2015; Wang et al. 2018; Wu et al. 2017; Luo et al. 2021). The regions with higher gene density and GC content were designated as compartment A while the rest regions were compartment B. In our study, we followed the same procedure. The compartment type is reported to be associated with chromatin type, where compartment A corresponds to euchromatin and compartment B corresponds to heterochromatin (Lieberman-Aiden et al. 2009). Gene density and GC content in euchromatin (compartment A) are higher than that in heterochromatin (compartment B). High gene density and GC content in compartment A could also be clearly seen in our results (**Response Fig. 1 and 2**). We have made these descriptions clearer in the Results and Method sections of the revised manuscript as below:

“At a 100 kb resolution, we identified the A or B (active or inactive) compartments of different muntjac species following the previous practice (Lieberman-Aiden et al. 2009; Wu et al. 2017). As expected, we found significantly higher gene density and GC content, higher gene expression level in the euchromatic A compartments than in the heterochromatic B compartments (Extended Data Fig. 3c, d) (Lieberman-Aiden et al. 2009).” (Line 236)

“Following many published studies (Lieberman-Aiden et al. 2009; Barutcu et al. 2015; Wang et al. 2018; Luo et al. 2021), the first principal component (PC1) was used to identify compartment A/B. Positive or negative values of the PC1 separate chromatin regions into two spatially segregated compartments and regions with higher gene density and GC content were assigned as compartment A, while the rest were compartment B.” (Line 683)

Response Figure 1. Gene density and GC content in compartment A and B.
 The difference of gene density or GC content in different compartment regions was checked using *t*-test.

Response Figure 2. Compartment A/B, gene expression level, GC content and gene density of female *M. crinifrons*, male *M. crinifrons*, *M. reevesi* and *Hydropotes inermis*. Compartment A/B: The blue part represents the compartment A, and the red part represents compartment B. **Gene expression level:** The gene

expression level was represented by the FPKM (fragments per kilobase of transcript per million). **GC content:** the GC content per 100kb. **Gene density:** the length ratio of gene per 100 kb. For male *M. crinifrons*, 1p and 1q represent short arm and long arm of chromosome 1, respectively.

Minor:

3. Supp. Table 1: BMF4 (F stands for female?) is reported as a male?

Response: The BMF represents the female *M. crinifrons* (black muntjac) while BMM represents the male *M. crinifrons*. The number after “BMF” or “BMM” represent different female or male *M. crinifrons* individuals. In order to avoid the ambiguity, we have changed to use Latin names for all species rather than simple abbreviations in the revised manuscripts and supplementary information.

4. Line 143: BMM (N50 contig length) is 3.7mb according to Supp. table 5.

Response: Thank you for catching this typo. We have rephrased the corresponding description in the revised manuscript as below:

“Then we used the Illumina short reads to polish these draft genomes and generated contig-level genome assemblies with contig N50 length ranging from 24.47 Mb to 37.86 Mb (Supplementary Table 5). For male *M. crinifrons*, we reassembled its genome with the PacBio data that we previously generated (Chen et al. 2019) and improved the contig N50 length from 1.46 Mb (Chen et al. 2019) to 3.79Mb.” (Line 152)

5. Figure 1a: Placement of 6 tandem fusions?

Since milu deer and white-lipped deer have 68 and 66 some of the tandem fusions could as well have been placed before the split of the muntjacs and these two samples?

In addition I suggest adding all fission and fusions to the figure just like in Extended data fig2 e.

Response: Thank you for your suggestion. In the Figure 1a, the red number on the side of arrow-shaped icons represent the times of tandem fusion or Robertsonian fusion events occurred during muntjac evolution. In addition, since *E. davidianus* (milu deer) and *C. albirostris* (white-lipped deer) have 68 and 66 chromosomes, we also detected the fusion events occurred in them and explored whether fusion events happened in common ancestor of these two species and muntjac, based on the previous chromosome painting results (Chi et al. 2005; Frohlich et al. 2017; Huang, Chi, Nie, et al. 2006). The results showed that a total of one and two Robertsonian fusion events occurred in *E. davidianus* and *C. albirostris*, respectively, and no fusion event was identified before the split of muntjac and *E. davidianus* and *C. albirostris*. We have added all of these fission and fusions in the Figure 1a of the revised manuscript as you suggested.

6. Line 185: Think it is an important result but circos plot, albeit pretty, does not convey the message very well to me.

Response: Yes, we agree that only circos plot doesn't convey much message. Therefore, to provide more evidence for the result that rapid karyotype evolution among muntjacs is not accompanied by rapid evolution of genomic sequences, we additionally calculated the substitution rate and sequence identity between female *M. crinifrons* (black muntjac, 2n=8) and *M. reevesi* (Chinese muntjac, 2n=46) as well as *H. inermis* (Chinese water deer, 2n=70) (**supplementary table 8**). For comparison, we also calculated the substitution rate and sequence identity between goat (2n=60) and cattle (2n=60) as well as sheep (2n=54) using the same method. Although the difference of chromosome number among female *M. crinifrons*, *M. reevesi* and *H. inermis* are so large, the substitution rates between these genomes are not higher than those among goat, cattle and sheep which have similar chromosome numbers (**supplementary table 8**). In addition, based on phylogenetic tree with calibrated divergence time (**Fig. 1a**), we also used the r8s to calculate the mutation rate of all species in the phylogenetic tree (**supplementary table 6**). The mutation rates of these species are similar to that calculated by Chen et al. (Chen et al. 2019) and don't have difference between species with and without large difference in chromosome number. Overall, all of these results demonstrated that the rapid karyotype evolution among muntjacs is not accompanied by rapid evolution of genomic sequences. We have added these comparison results in the Result part of revised manuscript as below: **“We compared the genome-wide substitution rates between species (*M. reevesi* (2n=46), *H. inermis* (2n=70) vs female *M. crinifrons* (2n=8)) with dramatic karyotype changes and Bovidae species (*B. taurus* (2n=60), *Ovis aries* (2n=54) vs *Capra hircus* (2n=60)) with similar karyotypes. The results show that at least 90% of female *M. crinifrons* genomic sequences could be mapped to *M. reevesi* and *H. inermis* with average sequence identity more than 90% (Extended Data Fig. 2f and supplementary Table 8), similar to that of the Bovidae species (supplementary Table 8). In addition, the substitution rates and mutation rate between different muntjac genomes are also similar to those between Bovidae species (supplementary Table 6 and 8).” (Line 213).**

7. Line 188 and ext. fig. 3a: Why is CWD so rich on trans read pairs compared to the muntjacs?

Response: Thank you for your question. Richness of trans reads pairs often happened in some species, which is also found in other published study (Aldiri et al. 2017) (Detailed information is in their Table S1). Moreover, we also calculated the “ligation motif present” which was defined as the percentage of restriction motif (Servant et al. 2015) using the juicer software. The results show that the “ligation motif present” of *H. inermis* reaches 88%, indicating that the Hi-C reads of *H. inermis* are not from random connection. We hypothesize that *H. inermis* have more trans read pairs than muntjacs probably because its chromosome number is much more than those of muntjacs, in which most trans reads become cis reads because of the chromosome fusions.

8. Line 365: The comparison to the Mudd et al could be expanded (see major comment).

Response: Thank you for your comments. We have added detailed comparison between our results and Mudd et al. in the Discussion part of the revised manuscript (Line 479), as detailedly listed in the response to your second major comment

9. Line 457: What is the genotype likelihood data used for?

Response: The genotype likelihood data generated by “samtools mpileup” tools with adjusted mapping quality more than 50 was fed to the “bcftools call” tool for calling SNPs. We have rephrased the corresponding description in the revised manuscript to make it clearer:

“The genotype likelihoods data were fed to bcftools (Li, 2011) for identifying SNPs with default parameters.” (Line 613)

10. Line 463: What estimates of mutation rate did the authors obtain using r8s?

Response: Thank you for your comment. We have added a supplementary table (**supplementary table 6**) which included the mutation rate of different muntjac species estimated by the r8s software in the revised manuscript.

11. Line 511: Please provide information on how you decided on the 0.85 and 0.7 minimum match?

Response: Thank you for your comments. Actually, we have tried different parameters ranged from 0.70 to 0.95 to obtain the homologous bin map for different muntjac species, and based on the mapping results, we chose the parameter –minMatch=0.85 for female vs male *M. crinifrons* (black muntjac) and female *M. crinifrons* vs *M. reevesi* (Chinese muntjac) and –minMatch=0.7 for female *M. crinifrons* vs *H. inermis* (Chinese water deer) to ensure that more bins (over 95% at 40 kb resolution) between genomes can be homologously mapped. We have added this information in method section of the revised manuscript as below:

“To ensure that more bins (over 95% at 40 kb resolution) in female *M. crinifrons* genome have homologous bins in other three genomes, the liftOver tool (Kent et al. 2002) with different parameters (female *M. crinifrons* vs male *M. crinifrons* and female *M. crinifrons* vs *M. reevesi*: -minMatch=0.85, female *M. crinifrons* vs *H. inermis*: -minMatch=0.7) was used to obtain the homologous bin pairs (Supplementary Fig. 5).” (Line 671).

Response Figure 3. Distribution of ratios of bins at 40 kb resolution in female *M. crinifrons* mapped with homologous bins in other three genomes. MCR, *M. crinifrons*; MRE, *M. reevesi*; HIN, *H. inermis*.

12. Line 515: Please specify the software and parameters used. This could generally be improved in the method section.

Response: Thank you for your suggestion. Following the published article (Crane et al. 2015), the *cworld-dekker* software (<https://github.com/dekkerlab/cworld-dekker/releases>) under default parameters was used to conduct compartment analysis. We have made this description clearer in the Method section as below:

“Principal component analysis was conducted using the *cworld-dekker* software with default parameters (<https://github.com/dekkerlab/cworld-dekker/releases>, v1.01) based on the ICE-normalized contact matrix at 100 kb resolution.”(Line 680).

13. Line 515: Why is only a single PC for the A/B compartment?

Response: Yes, as explained in detail in the response to your second major comment, PC1 is sufficient for partitioning the genome into two spatial compartments, and gene density and GC content can assign them into A or B type of compartment.

14. Line 619: Given that the sample size for males is two, how often would you expect that designated male-specific SNPs happen by chance?

Response: Yes, it is not very accurate that only using two individuals to estimate the male-specific SNP. To be more rigorous, we changed the “male-specific SNPs or indels” to “candidate male-specific SNPs or indels” in the revised manuscript.

15. Line 698: Did the authors correct for multiple testing both in PSG and REG?

Response: Thank you for the comment. Actually, we originally did the multiple testing correction and computed the false discovery rates (FDR) for PSGs and REGs. However, only a few PSGs and REGs with FDR less than 0.05 were identified (**Response Table 1**). Previous studies have demonstrated that the positive selection results would be robust when synonymous substitutions are far from saturation (Gharib and Robinson-Rechavi 2013). The mean dS (synonymous mutation rate) between muntjacs and other deer ranges from 0 to 0.3 (**Response Fig. 4**), which is far from synonymous substitution saturation. Therefore, in order to identify as many fast evolving genes as we can, we retained the PSGs and REGs with a raw p -value lower than 0.05 for comprehensively exploring the possible genetic mutations relevant to muntjac chromosome evolution. We have clearly stated this treatment in the method section of revised manuscript as below:

“The p -value (chi-square statistics) and false discovery rates (FDR) of each gene were calculated. There were only a few genes with FDR less than 0.05.

Considering that the positive selection results would be robust when synonymous substitutions are far from saturation (Gharib and Robinson-Rechavi 2013) and mean dS (synonymous mutation rate) between muntjacs and other deer ranges from 0 to 0.3 (Supplementary Fig. 7), which is far from synonymous substitution saturation, we finally selected the genes with p -value less than 0.05 as PSGs to comprehensively explore the possible genetic mutations relevant to muntjac chromosome evolution (Supplementary Table 18).” (Line 812)

Response Table 1. Number of PSGs and REGs under different filtering criterion. The p -values are from chi-square test and the FDR values are adjusted p -values using false discovery rates (FDR) method. “MCR_MGO_MMU” indicates that the PSGs and the REGs are identified in the common ancestor of *M. crinifrons* (MCR), *M. gongshanensis* (MGO) and *M. muntjac* (MMU).

Lineages	p -value <0.05		FDR <0.05	
	PSGs	REGs	PSGs	REGs
MCR_MGO_MMU	32	210	1	9
MCR	72	509	6	17
MGO	70	582	8	21
MMU	131	611	18	38

Response Figure 4. Synonymous substitution rate (dS) of genes. The value of dS was calculated using the free-ratio model of the Codeml module in PAML software package. The red stars represent the average value of dS .

Reviewer #2 (Remarks to the Author)

In this paper, the authors investigate chromosomal rearrangements in Muntjac deer, a clade with extensive karyotype variation. The authors generate chromosome-level genome assemblies to characterize how chromosome fusions have contributed to genome evolution among species. They then investigate potential demographic and molecular drivers of these fusions, and their effects on chromatin architecture. The authors also leverage these data to investigate the neo-sex chromosomes of Muntjac deer, revealing how chromatin architecture is evolving in sex-linked genomic regions. Overall, I think the findings will be of interest to the community. However, some of the text and figure panels are difficult to parse. Below I give my comments and recommendations for how the manuscript could be strengthened.

Response: Thank you for your positive comments. Following your suggestions, we have rephrased the manuscript especially the text related to the figure panels.

1. Introduction: More context could be given for the study, both in terms of karyotype diversity across mammals, and in terms of molecular/evolutionary drivers of karyotype variation. For example, how much variation exists among mammals in chromosome number? Or at lines 99-101, what are some hypothetical mechanisms that could underlie fusions in this group?

There could also be more context presented for sex chromosome evolution in Black Muntjac, for example, what sex chromosome system does this species have and how does this compare to other taxa in the group?

Response: Thank you for your great comments. Following your suggestions, we have extended the context related to karyotype diversity (Line 2), molecular drivers of

karyotype variation (Line 14) and sex chromosome system of *M. crinifrons* (Line 21) besides the original schematic diagram (Extended Data Fig. 7a) as follows:

“In mammals, the chromosome number ranges from $2n=6$ in the female Indian muntjac (*M. muntjak*) to $2n=102$ in the viscacha rat (*Tympanoctomys barrerae*) (Ferguson-Smith and Trifonov 2007; Pardo-Manuel de Villena and Sapienza 2001).” (Line 84)

“Previous studies on the sequence composition of some fusion sites have suggested that the chromosome fusions may be produced by sequence-specific recognition and illegitimate recombination between homologous DNA elements (or other specific motifs) on non-homologous ancestral chromosomes (Tsipouri et al. 2008; Hartmann and Scherthan 2004; Li et al. 2000b).” (Line 96)

“As one of the muntjacs with a very low chromosome number, *M. crinifrons* additionally possesses one sex chromosome system that does not exist in other muntjac species including its closest relative, *M. gongshanensis*. In *M. crinifrons*, the original eutherian X chromosome had experienced a centric fusion to one copy of chromosome 4, forming the “X+4” chromosome, and the short arm of chromosome 1 had undergone a male-specific translocation to another copy of chromosome 4, creating the “1p+4” chromosome (Zhou et al. 2008; Yang et al. 1997c; Huang, Chi, Wang, et al. 2006) (see Extended Data Fig. 8a below). Interestingly, two inversions involving large part of the “1p+4” were identified in male *M. crinifrons*, making the influenced regions (the ‘neo-Y’ regions) to evolve like a canonical mammalian Y chromosome, and its homologous counterparts on the chromosome X+4 and chromosome 1p to be neo-X regions.” (Line 103)

2. L197-201/Fig 2b: Can you use RNA-seq data to look at gene expression/co-expression around fusion sites before and after fusions have occurred?

Response: Thank you very much for your suggestion. Actually, we originally compared the gene expression level of 13,908 homologous genes (Best reciprocal hit BLAST) between *H. inermis* (Chinese water deer) and *M. crinifrons* (black muntjac) which represent the state before and after the fusion. After normalized gene expression level in different species, we found 5,928 genes showed expression differences (edgeR packages, p -value <0.01 and change fold ≥ 1.5), including 2,901 up-regulated and 3,027 down-regulated genes in *M. crinifrons*, respectively. We examined the distances of these differentially expressed genes to their nearest fusion sites (**Response Fig. 5a, b**), and calculated the ratio of differentially expressed genes within 1 Mb sliding window (**Response Fig. 5c**). The results showed that the differentially expressed genes are not enriched near fusion sites (**Response Fig. 5c**). However, this conclusion has to be interpreted with caution because we were unable to acquire the exact same tissue samples between the two species for comparison. As *M. crinifrons* is an endangered species, the only available RNA sample that we were able to acquire is a fibroblast cell line, but the samples of *H. inermis* were derived

from fresh blood. The difference of RNA samples may lead to incorrect inference. Therefore, we have not added these analysis results in the manuscript.

Response Figure 5. Possible differentially expressed genes in *M. crinifrons* and *H. inermis* based on limited RNA-seq data in our hand. a, Density distribution of distance between up-regulated or down-regulated genes with their nearest fusion sites. b, Density distribution of distance between homologous genes of female *M. crinifrons* and *H. inermis* from their nearest fusion sites. c, Ratio of number of up-regulated or down-regulated genes in 1 Mb window.

3. L208-211/Fib 2c: It would be helpful to label the figure so it matches with the percentages described in the main text (95.27%/44.37%).

Response: Following your suggestion, we have labeled the **Fig. 2c** in the revised manuscript.

4. L212-214/Fig 2d: I found this difficult to follow. I'm not sure what is meant by the 'third or farther ancestral chromosomes'.

Response: Sorry for the ambiguity. The 'third or farther ancestral chromosomes' here means the significant interaction across more than two fusion sites. In contrast, significant interactions across just one fusion site anchored their two ends on the ancestral chromosomes which adjacently fused in *M. crinifrons* (black muntjac). We have rephrased the description to:

"In addition, most of these significant interactions only span one fusion site and anchor their two ends on two adjacent fused ancestral chromosomes while only a few (16.82%) span two or more fusion sites (Fig. 2d)." (Line 286)

For a clearer presentation, we also split the original **Fig. 2d** into two panels (now **Fig. 2d** and **Fig. 2e**).

5. L217-219/Fig 2e: I also found this difficult to follow. Please edit this section and the figure to more clearly communicate the results.

Response: Sorry for the ambiguity. We have re-phrased the statement here as:

"Interestingly, the abundance of significant interactions across fusion sites is positively associated with the ages of fusion sites (Fig. 2e and Extended Data Fig. 2f). Similar patterns can also be directly seen in those interaction matrix that we constructed using Hi-C read pair for all 3D chromatin structure analyses (Fig. 2f and Extended Data Fig. 7). Particularly, the frequency of Hi-C read pair interactions spanning the oldest fusion sites has reached the same level as those in other genomic regions that have not undergone fusions (Fig. 2f, Extended Data Fig. 7)." (Line 289)

The legend of original **Fig. 2e** (now **Fig. 2f**) has been revised as below:

"Combined heatmaps of contact matrix around the fusion sites of female *M. crinifrons* (upper right) and their homologous regions in *M. reevesi* (lower left) at 20 kb resolution. Hollow shapes represent the locations of fusion sites. Different geometries represent different fusion site types. The "ancestral fusion sites" refers to the oldest fusion sites shared by five muntjac species. The "tandem fusion sites" represent the remaining tandem fusion sites of female *M. crinifrons* except for the ancestral fusion sites. The "Robertsonian fusion sites" refers to fusion sites raised by Robertsonian fusion and they are the youngest fusion sites." (Line 1229)

6. Fig 3a: It's difficult to see the coverage, SNPs, etc in the circular plot. It would be helpful to see the sex-linked regions in their own panel.

Response: Thank you for your suggestion. We replaced the scatter diagram in circos plot using the heatmap which can clearly highlight the higher density of candidate male-specific SNPs, InDels and SVs in neo-sex-linked regions than other genomic regions. Meanwhile, the coverage track has become clearer. We also added violin plots (now **Fig. 4b** due to the reorder between different results sections) to show the significant difference of candidate male-specific mutations in neo-sex-linked regions and other genomic regions.

7. Fig 3b: Are these inversions contributing to the formation of strata on the sex chromosomes?

Response: Yes, these inversions have contributed to the formation of strata of the neo-Y chromosome regions but not the original mammalian Y chromosome. Density of the candidate male-specific mutations in the first inverted region of neo-Y chromosome is indeed higher than that in the second (**Fig. 4a**), which could comprise the “strata” pattern. However, because the degradation degree of neo-Y inverted region, either in first or the second inverted region, is very low, we combine them together in the analysis.

8. Fig 3c/d: Is there any relationship between compartment switching and certain GOs or pathways?

Response: Thanks for this intriguing question. A total of 131 genes in neo-sex-linked regions with switched compartment type between female and male *M. crinifrons* (black muntjac) were identified, 57 of which were located in the regions with compartment switched from A compartment in female *M. crinifrons* to B compartment in male *M. crinifrons* while 74 of which were located in the regions with compartment switched from B compartment to A compartment. We conducted GO and pathway enrichment analysis on these genes using metascape (Zhou et al. 2019). The results showed that the former genes were enriched in some GOs or pathways that were related to metabolism and biosynthesis and the latter genes were enriched in some GOs or pathways that were related to transmembrane transport and development. We have added these results in the revised manuscripts:

“There are 131 genes in the neo-Y regions with switched compartment (Supplementary Data 7). The 57 genes in regions with compartment switched from A in female *M. crinifrons* to B in male *M. crinifrons* are enriched in GOs or pathways related to metabolism and biosynthesis (Supplementary Data 8), and the 74 genes in regions with an opposite direction of compartment switch are enriched in GOs or pathways related to transmembrane transport and development (Supplementary Data 8). This result indicates that these function-related genes located in neo-Y region are firstly affected by compartment switch.” (Line 431)

9. Fig 3c/d: How many genes are included in these analyses?

Response: For Fig. 3c (now **Fig. 4d**), we did not use genes in the A/B compartment analysis. The points in the original Fig. 3c (now **Fig. 4d**) represent the homologous bin pairs in female and male *M. crinifrons* (black muntjac). The total number of bin pairs in neo-Y regions is 3310, while that in other regions is 17640.

For Fig. 3d (now **Fig. 4e**), the data indicate the number of RNA-seq reads supporting different alleles at 2,331 sites on 169 genes in neo-sex regions with switched compartment. We have added these descriptions in the revised manuscript:

“In contrast to the highly similar A/B compartment pattern between female and male *M. crinifrons* in general (Fig. 2a), the neo-Y region has accumulated significantly more A/B compartment switches (8.9% of total 3,310 bins have undergone compartment switch, χ^2 test, $p < 0.01$) than other genomic regions (only 0.7% of 17,640 bins have undergone compartment switch) (Fig. 4d and Extended Data Fig. 7d, e).” (Line 426)

“Our results show that 2,331 sites on 169 genes that can be distinguished by candidate male-specific mutation generally have significantly higher expression level in compartment A than compartment B in the neo-Y region with compartment switch (Fig. 4e).” (Line 440)

10. L366-368: The same sentence is repeated at the end of the discussion (L393-394).
Response: Sorry for this redundancy. We have deleted this sentence in the revised manuscript.

11. General: The final section of the results (molecular causes of fusions) would go better before the section on sex chromosome evolution. I suggest re-ordering the results in this way.

Response: Thank you for this helpful suggestion. We have reordered the two sections in our revision.

12. Discussion: The results show that chromosome fusions have a greater impact on TADs than on compartments, while recombination suppression among the sex chromosomes seems to drive more changes in compartment than in TADs. Could the authors comment further on these differences and their implications?

Response: Thank you for this suggestion. Suggested by the reviewer#3, we further examined our *M. crinifrons*-specific TAD identification procedure. We found that the considered missing bins when comparing TADs of female *M. crinifrons* and male *M. crinifrons*, *M. reevesi* and *H. inermis* and the inclusion of male *M. crinifrons*-specific and female *M. crinifrons*-specific TADs would lead to an overestimate of *M. crinifrons*-specific TAD. When excluding missing bin and defining *M. crinifrons*-specific TADs as female *M. crinifrons* TADs shared by male *M. crinifrons* but not by *M. reevesi* and *H. inermis*, we found that the *M. crinifrons*-specific TADs did not

concentrate near the fusion site. That is to say, like compartment, chromosome fusion has no greater impact on TADs. And we corrected this conclusion during this revision.

However, the conclusion still holds that recombination suppression among the sex chromosomes have driven more changes in compartment than in TADs. This may be caused by the formation mechanism of TADs, the biological effect of compartment and the different changes caused by chromosome fusion and recombination inhibition. TAD is formed by loop extrusion facilitated by the boundary proteins, including CTCF and architectural proteins cohesion (Fudenberg et al. 2016). CTCF can identify specific DNA sequence motif and limit the boundary of TAD (Fudenberg et al. 2016). Therefore, as long as CTCF and cohesion proteins, as well as the sequence motif, are not affected by chromosome fusion or inversion, TADs do not seem to be affected. TADs in other genomic regions are normal, indicating that CTCF and cohesion proteins function normally, and the large-scale chromosomal rearrangement, such as fusion and inversion are unlikely to affect the two proteins too much. The TAD boundary sequence motif is short and also unlikely damaged biasedly in recombination suppression regions or regions near fusion sites. The compartment type is associated with chromatin type, where compartment A corresponds to euchromatin and compartment B corresponds to heterochromatin (Lieberman-Aiden et al. 2009). Genes in genomic regions of compartment A tend to express actively, while that in compartment B tend to be silent (Wu et al. 2017). Compartment switch would thus be accompanied by changes of gene expression (Wu et al. 2017). Sequence degradation and reduced gene expression caused by recombination inhibition and changed selection pressure may lead to compartment switch happened first in the neo-Y regions. However, the neo-X sequences or homologous sequences of other species are more likely to maintain the compartment type because they are not affected by the recombination suppression. We have added these in the discussion section of revised manuscript (Line 519). We also checked other analysis pipelines to make sure about our other results. We didn't find any more flaw besides the TAD analysis around fusion sites.

13. There are a number of grammatical errors throughout the text that make some parts difficult to read. I recommend revising these for clarity; a few are highlighted below:

Response: During the revision, we have carefully revised the entire manuscript and corrected the grammar and spelling errors.

14. L209: should be 'almost all of them'

Response : Corrected as suggested.

15. L210: should be 'this is in contrast to the rest of the significant interactions'

Response : Corrected as suggested.

16. L223: should be 'may be of biological significance'

Response : Corrected as suggested.

17. L292: should be 'we searched for evidence'

Response : Corrected as suggested.

Reviewer #3 (Remarks to the Author):

Review of Yin et al.

Molecular mechanisms and topological consequences of drastic chromosomal rearrangements of muntjac deer

The authors have addressed one of the most interesting questions in mammalian chromosome evolution, i.e., how did the chromosomes of some but not all species of muntjacs become reduced to a small number of giant chromosomes in a relatively short time span of 1-2 million years. Although it has been known from FISH and other comparative studies that the giant chromosomes were formed largely by fusions and Robertsonian translocations, the mechanisms by which this occurs and the consequences of the rearrangements on the three dimensional structure in the nucleus have not been determined. The authors conducted comparative and evolutionary whole-genome analysis using de novo chromosome-scale assemblies and long read data to identify the possible mechanisms by which the fusions have occurred in the muntjac lineages that exhibit reduced diploid numbers. Deep coverage Hi-C data was used to explore how such rearrangements affected the definition of A/B chromatin compartments and TADs, as well as long range interactions between non-homologous chromosomes. Additionally, an interesting evolutionary analysis of the neo-X chromosome and the neo-Y was conducted.

Response: Thank you very much for acknowledging the importance of this study.

General Comments

1. The authors are commended for preparing a well-written manuscript, although it needs thorough editing for grammar and spelling. Another issue is the excessive use of non-standard abbreviations that make the manuscript very difficult to read. Examples include species names, LRSI, and SIAF.

Response: Thanks for your comment. During the revision, we have carefully revised the entire manuscript and corrected the grammar and spelling errors. In addition, we have abandoned non-standard abbreviations in the revised manuscript and used short Latin species names to make reading much clearer.

2. Genome assemblies and phylogenetic analysis are presented with an overlay of the demographic history of deer in the Far East. Based on genomic analysis the authors have concluded that Indian Muntjac, Black muntjac and Gongshan muntjac underwent a dramatic reduction in population size 1-2 Mya near the time of their

divergence, and correlated these findings with the Xixiabangma glaciation event. The authors proposed that genetic drift and reproductive isolation might have contributed to the fixation of the dramatic chromosomal rearrangements found in these three species of muntjacs, but not Chinese muntjacs, which diverged from a common ancestor about 4.7 Mya. This conclusion seems reasonably well supported by the data.

Response: Thank you very much for your positive comments.

3. One of the main conclusions of the paper is grossly overstated, e.g., “These results resolve the long-standing mystery underpinning the 79 recurrent chromosome fusions and reveal how chromosome rearrangements ...” The results do not “resolve” the mystery of precisely how the reduction in genome size occurred (see also line 281 section title “molecular cause of chromosome fusions in muntjac species”). While the authors preferred this interpretation, cause and effect was not demonstrated (see further discussion below). The alternative hypothesis that other genomic mutations could have led to the chromosome reductions was not ruled out by the data, which also showed accelerated evolution and positive selection in genes that could be part of the compensatory mechanism for very large chromosome size. While it is recognized that cause and effect are difficult (but not impossible) to determine, this does not give the authors license to conclude that the mystery has been solved. For example, experimental systems with artificial chromosomes containing the complex repeat could be used to demonstrate chromosome instability (e.g., promotion of fusions) in vitro, as has been done for other similar systems.

Response: Thank you for your suggestion. We agree that it is difficult to recognize the cause and consequence. Therefore, we have toned down all related statements throughout the manuscript to tentatively propose that the complex repeat structure is likely to be the cause of the recurrent fusion. In addition, except for fusion, genome-wide rearrangement rates in muntjac species with extensive chromosome fusions were not higher than other ruminant species without recurrent fusions. This result suggest that accelerated evolution of genome stability genes is less likely to be the driving force to result in the recurrent fusion, because if so, we would also see more other rearrangements in genomes. We agree future experiments with artificial chromosomes could be the solidest evidence to this conclusion. But given the tremendous difficulty to construct and mimic mammalian artificial chromosomes at this stage of our team, this experiment has to be left for future study.

4. The issue of compensatory mechanisms among the positively selected genes is an interesting one. Are there any data to support that the muntjac mutations are loss/gain-of-function mutations in any other species? Have knock-out, knock-ins or CRISPR mutations in these genes in mice been shown to affect chromosome stability during mitosis or meiosis?

Response: Thank you for your comments. The REGs and PSGs having functions in processes of “cell cycle”, “DNA damage response” or “telomere maintenance” is from the results of the GOs or pathways enrichment analysis. Following your suggestion, we further searched the REGs and PSGs at the Mouse Genome

Informatics (MGI) database. In MGI, mutations in 72 mice genes of these rapidly evolving or positively selected genes have been verified to cause “abnormal meiosis or mitosis”, “abnormal DNA repair” and other phenotypes related to chromosome stability (**Response Table 2**). We have added this information and the **Response Table 2** as a new supplementary table in the revised manuscript:

“Among these lineages, orthologs of a total of 72 REGs and PSGs have been functionally tested in mice, and they seem to play important roles in processes related to cell cycle, DNA repair or chromosome stability (Supplementary Data 5).” (Line 385)

Response Table 2. Genes and phenotype related to chromosome stability in Mouse Genome Informatics (MGI) database. Mammalian Phenotype (MP) represent some phenotypic term clearly defined in MGI database, with unique corresponding MP identity (id). MCR_MGO_MMU means the common ancestor of *M. crinifrons* (MCR), *M. gongshanensis* (MGO) and *M. muntjac* (MMU)

Linages	Gene symbol	Mammalian Phenotype
MCR_MGO_MMU	ASPM	abnormal cell cycle
MCR_MGO_MMU	AURKA	increased mitotic index; chromosomal instability; abnormal mitotic spindle morphology
MCR_MGO_MMU	CDK12	chromosomal instability
MCR_MGO_MMU	FANCB	abnormal double-strand DNA break repair
MCR_MGO_MMU	MBD4	abnormal cell cycle checkpoint function
MCR_MGO_MMU	RCOR2	abnormal cell cycle
MCR_MGO_MMU	SIRT1	abnormal chromosome morphology; abnormal cell cycle checkpoint function; increased mitotic index; abnormal DNA repair; arrest of male meiosis
MCR_MGO_MMU	SPTBN1	abnormal cell cycle; abnormal cell cycle checkpoint function; decreased mitotic index
MCR	B3GAT3	abnormal mitotic cytokinesis
MCR	CDKN1A	abnormal cell cycle; abnormal chromosome number; abnormal cell cycle checkpoint function
MCR	CEP290	abnormal double-strand DNA break repair
MCR	CHD1	abnormal cell cycle
MCR	COPS8	abnormal cell cycle
MCR	CTNND1	abnormal mitotic spindle morphology
MCR	EXO1	abnormal male meiosis; chromosomal instability; abnormal mismatch repair; abnormal double-strand DNA break repair
MCR	LIG3	elevated level of mitotic sister chromatid exchange; abnormal DNA repair

MCR	LIN9	abnormal mitotic spindle assembly checkpoint
MCR	MEIOB	abnormal meiosis; abnormal double-strand DNA break repair
MCR	MUS81	chromosome breakage; abnormal DNA repair
MCR	NIN	abnormal mitotic spindle morphology
MCR	PAXIP1	abnormal DNA repair
MCR	TRIP13	abnormal female meiosis; abnormal DNA repair; arrest of male meiosis; abnormal chromosomal synapsis; abnormal double-strand DNA break repair
MCR	UNG	abnormal base-excision repair
MCR	WDFY3	abnormal cell cycle
MCR	WEE1	abnormal cell cycle checkpoint function
MGO	ATM	abnormal cell cycle; abnormal chromosome morphology; chromosome breakage; spontaneous chromosome breakage; abnormal cell cycle checkpoint function; abnormal DNA repair; chromosomal instability
MGO	BRPF1	abnormal cell cycle checkpoint function
MGO	BTRC	abnormal mitotic spindle morphology
MGO	BUB1B	abnormal chromosome number; chromosome breakage; abnormal cell cycle checkpoint function; abnormal mitotic spindle morphology; abnormal mitotic spindle assembly checkpoint
MGO	CENPE	chromosomal instability
MGO	DCLRE1C	induced chromosome breakage; chromosomal instability
MGO	ERCC2	abnormal DNA repair
MGO	ERCC6L	abnormal cell cycle checkpoint function; chromosomal instability
MGO	EYAI	abnormal mitotic spindle morphology
MGO	FANL	abnormal DNA repair; abnormal cell cycle
MGO	FIGN	abnormal cell cycle
MGO	FUS	abnormal chromosome morphology; chromosome breakage; abnormal male meiosis
MGO	HUS1	abnormal chromosome morphology; chromosome breakage; induced chromosome breakage; abnormal cell cycle checkpoint function
MGO	LIG1	chromosomal instability
MGO	MCM9	abnormal cell cycle; spontaneous chromosome breakage; arrest of male meiosis;
MGO	MEIOB	abnormal meiosis; abnormal double-strand DNA break repair

MGO	MEIOC	arrest of male meiosis; abnormal female meiosis I arrest; abnormal double-strand DNA break repair;
MGO	NEIL2	spontaneous chromosome breakage
MGO	PHACTR4	abnormal cell cycle
MGO	PRKDC	abnormal DNA repair
MGO	PRR19	abnormal meiosis; abnormal double-strand DNA break repair; abnormal X-Y chromosome synapsis during male meiosis
MGO	RAG1	spontaneous chromosome breakage
MGO	RBM14	abnormal cell cycle
MGO	RPAI	chromosome breakage
MGO	SIN3B	abnormal cell cycle checkpoint function
MGO	TDG	abnormal mismatch repair
MGO	ZCCHC8	abnormal female meiosis
MMU	ATM	abnormal cell cycle; abnormal chromosome morphology; chromosome breakage; spontaneous chromosome breakage; abnormal cell cycle checkpoint function; abnormal DNA repair; chromosomal instability
MMU	CHD4	abnormal cell cycle
MMU	CTCI	abnormal DNA repair; chromosomal instability
MMU	DPHI	abnormal cell cycle
MMU	FEN1	spontaneous chromosome breakage; abnormal DNA repair; chromosomal instability; abnormal double-strand DNA break repair
MMU	GAS2L3	abnormal mitotic cytokinesis
MMU	JDP2	abnormal cell cycle
MMU	JUND	abnormal cell cycle
MMU	MIF	decreased mitotic index
MMU	NUSAP1	increased mitotic index; abnormal mitotic spindle morphology
MMU	POLE3	abnormal cell cycle
MMU	POLK	abnormal DNA repair
MMU	PRDX1	abnormal cell cycle
MMU	RAE1	abnormal cell cycle checkpoint function
MMU	RANBP1	abnormal cell cycle
MMU	RBBP8	abnormal cell cycle
MMU	RBM14	abnormal cell cycle
MMU	RIF1	abnormal DNA repair; abnormal double-strand DNA break repair
MMU	RPS6	abnormal cell cycle
MMU	TELO2	abnormal cell cycle; abnormal DNA repair

MMU	VCPIP1	abnormal chromosome morphology; abnormal DNA repair; chromosomal instability
MMU	WAPL	abnormal cell cycle
MMU	WDR62	abnormal cell cycle; abnormal mitotic spindle morphology; abnormal mitotic spindle assembly checkpoint
MMU	WEE1	abnormal cell cycle checkpoint function

5. The conclusion that truncated telomeric repeats are related to the fusion of muntjac chromosomes may not be supported by the data presented in the manuscript. Firstly, the genomes analyzed were not assembled across the fusion sites except for one fusion site in black muntjacs. To compensate for this, the authors had to use long ONT reads that “possibly” spanned the fusion sites (line 293). This approach is very different than directly analyzing a complete assembly where the length of the specific repeats can be accurately measured. Complete assembly across the fusion sites, admittedly being difficult, is really the only way to support the authors’ conclusions about the length differences between species. The authors identified what they believe to be truncated telomeric sequences to support their claim. Can the authors be sure that these truncated reads are not due to problems with ONT reads? Long ONT reads are susceptible to termination due to secondary structures (hairpins, etc.), that are common when the single strands contain repetitive elements. Can the authors rule this out?

Response: Thank you for your comments. We agree with you that complete assembly across the fusion sites is a more convincing data to prove the role of the telomeric sequence-related repeat in the fusion of chromosomes. However, as you pointed out, at this stage it is still extremely difficult to read through large repetitive fragment using current sequencing technologies. We thus had to turn to the alternative approach to directly analyze long ONT reads that may contain markers close or within the fusion sites. According previous studies, the centromeric regions of ancestral chromosomes and the fusion sites of muntjac deer contain three types of Cervidae-specific satellite sequence and telomeric sequence (Liu et al. 2008; Li and Lin 2011; Lin et al. 2004; Hartmann and Scherthan 2004). Therefore, the long ONT reads which contained these satellite and telomeric sequences may have higher possibility to be from the fusion sites or telomere/centromere regions. Sequence features of these long reads and their abundance could partially reflect the footprint of chromosome fusion. We are sorry for the previous inaccurate description by saying “possibly spanning fusion sites”.

We found that the very short telomeric sequence (~38bp or ~25bp) we identified is usually in the middle of a ONT long read, not at the truncated points of the long reads. We analyzed the distance between these short/truncated telomeric sequences with the nearest end of the ONT long reads and calculated the percentage of the distance in total length of each ONT long read (**Response Fig. 6**). The results show that the short telomeric sequence does not concentrate in the end of ONT long reads in all the four

investigated muntjac species, but usually distribute in the middle of reads. Moreover, we also analyzed the PacBio reads of male *M. crinifrons* (black muntjac) from Chen et al. (Chen et al. 2019) by the same method, and still found that there are also similar short telomeric sequences together with the satellite DNA and palindrome structure in them, which are also usually located in the middle of the reads. Therefore, the structure of the short telomeric sequences is less likely to be false sequencing results in the long ONT reads. The previous statement of “truncated telomeric sequence” might have caused your misunderstanding about the short telomeric sequence, sorry for that. We have added the results of the new analysis shown in the **Response Fig. 6** in the revised manuscript:

“Further examination revealed that the truncated telomeric sequence is primarily located in the middle, not at the ends of the Nanopore reads (Supplementary Fig. 2), which also indicated that these truncated telomeric sequences are not caused by pre-termination of the Nanopore reads, but are more likely derived from the acrocentric regions of ancestral chromosomes and have originally been located in the region between satI and satIV (Supplementary Fig. 3).” (Line 359)

In addition, as described above, the genomic rearrangement rate comparison also shows the accelerated genome stability genes didn't cause high rearrangement rate in muntjac. Taken all, we have rephrased related sentences to tone down the statement to tentatively propose that the featured repeat may possibly be the cause of the recurrent chromosome fusions in muntjacs, and as you pointed, further experiments such as artificial experiments are needed to test this possibility.

Response Figure 6. Density of reads with different distance between short telomeric repeats from reads' end. The abscissa shows the percentage of distance between the short telomeric repeat and its nearest read end to the total length of the read. MGO, *M. gongshanensis*; MCR, *M. crinifrons*; EDA, *E. davidianus*; MRE, *M. reevesi*.

6. A second major conclusion of the paper is that A/B compartments are minimally

affected by rearrangements, but that TADs are affected around the rearrangement sites. This is probably the most interesting and important finding of the paper. However, there appear to be methodological issues that require further explanation to support these conclusions.

Response: Thank you for pointing that out. We took your suggestion and re-examined our methods of analyzing the impact of fusions of compartments and TADs. The new results showed that in fact both compartments and TADs are minimally affected by fusion. In the previous analysis, we included the male and female *M. crinions* specific TADs when identifying *M. crinions* specific TADs, the results showed that the impact on the TADs seemed to be caused by the TAD differences between sexes (we estimated there are about 30% of the TADs that are different between sexes), rather than between before and after the chromosome fusions. Moreover, as suggested by your comment #12, we also found that the missing bins might slightly overestimate *M. crinifrons* specific TADs. After excluding the impact of male and female samples by defining *M. crinifrons* specific TADs as female *M. crinifrons* TADs shared by male *M. crinifrons* but not shared by *M. reevesi* and *H. inermis* and excluding missing bins, we found that the chromosome fusion has no greater impact on TADs either, as the case in compartment, but significant interactions have been intensively reshaped as we show in the manuscript. We have updated all related statements in our revised manuscript.

7. As mentioned above, the genomes used were not fully assembled around the fusion sites. Given this issue, how can the authors be sure that the liftover coordinates on the black muntjac genome are not causing a problem with the definition of TAD boundaries? Are there any gaps or Ns added to these regions in any of the assemblies that might change the boundaries?

Response: Thank you for this comment. There are indeed gaps in the fusion site regions. The fusion site regions of female *M. crinifrons* (black muntjac) and their homologous centromeric and telomeric regions in related species have a large quantity of repetitive sequences which lead to the gaps. Even if gap regions have been assembled completely and the Hi-C reads alignment falling in these regions will be mostly filtered out by HiC-Pro because of multiple alignments caused by repetitive sequences (Servant et al. 2015). Therefore, it is possible that TAD boundary will not be identified at these long repetitive regions represented by gaps. These unrecognized TAD boundaries do lead to mis-recognition of a few TAD intervals, but the mis-recognition is limited. Other genomic regions were assembled well, which ensured that most TAD boundaries and TAD intervals were unaffected. In addition, we had respectively identified TAD boundaries in all genomes before liftover, which would not cause the calculation error of TAD boundaries during coordinate transformation. At the same time, in order to avoid the influence of gap on a single bin, we adopt the way of comparing the whole TAD interval across genomes rather than the TAD boundaries, which allow us minimize the influence of gaps.

8. Relatedly, is the 70% of TAD length sufficient to call TAD boundaries around breakpoint and/or fusion sites? In the paper cited (Dixon et al., 2012) the comparison between human and mouse TADs is done using the exact human and mouse coordinates to determine the boundaries within syntenic regions. The authors did not use custom parameters for their analysis. The default parameter for the UCSC liftover tool for identity percentage is 0.95. That is the likely reason why the percentage of shared TADs between human and mouse was low. Using TADs overlap of 70% does not appear to be justified given the recent divergence time and would not appear to give precise coordinates for the comparison of fusion points in the different species.

Response: Thank you for this important comment, which intrigued us to realize that it is untenable to compare the results of our muntjac deer with those of humans and mice. We have corrected the statement about comparison between our results and that of Dixon et al. in our revised manuscript. We explain our choice of the 70% length cutoff as follows:

We have carefully scrutinized the literatures about studying the conservation of TAD between samples or species. We found that different studies adopt different strategies to assess the conservation of TADs between different genomes. Sometimes people were comparing TADs themselves, but sometimes were comparing the boundaries of TADs. It is clear now that the stability of TAD boundaries between cell lines or species is often higher than that of TAD itself (McArthur and Capra 2020), which could also be inferred from many other works as we summarized in the **Response Table 3**. From these previous works, it is also clear that different strategies and parameters can be used to compare TAD boundaries or TADs across species or cell lines. For instance, Luo et al. compared the TAD boundaries between human, mouse and macaque (Luo et al. 2021). They considered the upstream and downstream bins ($\pm 80\text{kb}$) of each boundary when identifying the conserved boundaries between species, instead of using the exact coordinate and 0.95 liftover parameter as Dixon et al. did (Dixon et al. 2012). When identifying conserved TADs, Wu et al (2017) used 70% overlap cutoff, and Liu et al. (2017) used 75%. In our study we also compared TADs by corresponding each bin of female *M. crinifrons* (black muntjac) to bins of other three genomes using 0.7~0.85 liftover parameter (explanation on parameter selection is in response to your twelfth comments), rather than TAD boundaries. Then the conserved TADs within or between deer species are defined by following the practice of Wu et al. (Wu et al. 2017) who defined the conserved TADs as the two TADs from different cell lines of human containing 70% overlap regions. Using the same parameter, we found that the percentage of stable TADs (72.5%~72.7%) in different *M. crinifrons* individuals is almost the same as that between different human cell lines (70.9%~75.6%, their Fig. 4f) (Wu et al. 2017), indicating the robustness of this analyzing method. Beyond this, we further found that the conservation of TADs decreases obviously along divergence time while boundary changes very little when the divergence time varies from 15 to 90 million years (**Response Table 3**). These data suggest that TAD stability is not comparable to stability of TAD boundaries. Therefore, in this study when we explore the conservation pattern between species we compared TADs themselves rather than DNA boundaries, and used 70% overlap as

the cutoff as Wu et al. to identify conserved TADs. Using higher cutoff, say 95%, will miss some homologous TADs between distant species, such as between *M. crinifrons* and *H. inermis*. Since TADs are usually long, the utilization of 70% overlap actually has very low false positive results.

Response Table 3. Comparison of TAD boundaries and TADs within or across species. The method or parameters of comparing TAD boundaries or TADs in different studies are different in different studies. Divergence time were from the reference or TimeTree database.

Species1	Species2	Divergence time (Mya)	TAD boundary	TAD	Reference
Human	mouse	90	53.8~75.9%	-	(Dixon et al. 2012)
Human	macaque	28.81	78.7%~81.5%	-	(Luo et al. 2021)
Human	mouse	90	62.3%~73.0%	-	(Luo et al. 2021)
D. melanogaster	D. triauraria	15	72%	25%	(Torosin et al. 2020)
Human	chimpanzees	6.7	-	43%	(Eres et al. 2019)
M. crinifrons	H. inermis	11.3	-	43.3%~49.2%	Our study
M. crinifrons	M. reevesi	3.05	-	63.7%~65.4%	Our study
M. crinifrons (female and male individual)		-	-	72.5%~72.7%	Our study
Human (GM12878, RPMI-8226, U266)		-	-	70.9%~75.6%	(Wu et al. 2017)
D. melanogaster (replicate 1 and 2)		-	74%	-	(Torosin et al. 2020)
D. triauraria (replicate 1 and 2)		-	70%	-	(Torosin et al. 2020)
Human (hESC, IMR90)		-	65.5%~71.8%	-	(Dixon et al. 2012)

Specific comments

9. L138 p7: “The assembled 4, 5, 23...” sentence not clear. Please rephrase.

Response: Sorry for this unclear sentence. We rephrased this sentence to

“Using Hi-C data, we further anchored 98.82%, 91.49%, 98.62% and 97.57% of the contigs from female and male *M. crinifrons*, *M. reevesi* and *H. inermis* (Wang, Zhang, et al. 2019) into 4, 5, 23, and 35 haploid chromosomes, respectively (Extended Data Fig. 1b and Supplementary Table 5), which are consistent with their reported karyotypes (Yang et al. 1997a; Lin and Li 2006; Yang et al. 1997c).” (Line 157)

10. L183 p9: Extended data Fig. 2f only show the percentage of the genomes that were aligned between the species. To support the claim that the rapid karyotype evolution in muntjacs is not accompanied by rapid sequence evolution, the authors should calculate the substitution rates for the species and compare with other existing data or at least report the average sequence identity across the alignment.

Response: Thank you for your great suggestion. Following your suggestions, we additionally calculated the substitution rate and sequence identity between female *M. crinifrons* (black muntjac, 2n=8) and *M. reevesi* (Chinese muntjac, 2n=46) as well as *H. inermis* (Chinese water deer, 2n=70) (**supplementary table 8**). For comparison, we also calculated the substitution rate and sequence identity between goat (2n=60) and cattle (2n=60) as well as sheep (2n=54) using the same method. Although the difference of chromosome numbers among female *M. crinifrons*, *M. reevesi* and *H. inermis* is large, the substitution rates between these genomes are not higher than those among goat, cattle and sheep, which have a similar chromosome number (**supplementary table 8**). In addition, based on phylogenetic tree with calibrated divergence time (**Fig. 1a**), we also used the r8s to calculate the mutation rate of all species in the phylogenetic tree (**supplementary table 6**). The mutation rates of these species are similar to that calculated by Chen et al. (Chen et al. 2019) and don't have obvious difference between species with large difference in chromosome number and those with small chromosome variation. Overall, all of these results demonstrated that the rapid karyotype evolution among muntjacs is not accompanied by rapid evolution of genomic sequences. We have added these comparison results in the Result part of revised manuscript:

“We compared the genome-wide substitution rates between species (*M. reevesi* (2n=46), *H. inermis* (2n=70) vs female *M. crinifrons* (2n=8)) with dramatic karyotype changes and Bovidae species (*B. taurus* (2n=60), *Ovis aries* (2n=54) vs *Capra hircus* (2n=60)) with similar karyotypes. The results show that at least 90% of female *M. crinifrons* genomic sequences could be mapped to *M. reevesi* and *H. inermis* with average sequence identity more than 90% (Extended Data Fig. 2f and supplementary Table 8), similar to that of the Bovidae species (supplementary Table 8). In addition, the substitution rates and mutation rate between different muntjac genomes are also similar to those between Bovidae species (supplementary Table 6 and 8). These results demonstrated that the rapid karyotype evolution among muntjacs is not accompanied by rapid evolution of genomic sequences.” (Line 213).

Supplementary Table 1. The statistics of genome alignment results. The total base

means total sequence length assigned on chromosomes. The alignment ratio was the percentage of alignment length in total base. Sequence identity is the percentage of exact matched base in alignment length. Substitution rate is the number of substituted sites divided by two folds of divergence time.

	Female MCR as the reference			Goat as the reference	
	Male MCR	MRE	HIN	Cattle	Sheep
Total base	2,432,301,072	2,459,394,662	2,468,664,404	2,582,134,882	2,582,134,882
Mapped base	2,392,894,791	2,391,481,492	2,279,835,188	2,111,717,083	2,423,998,847
Mapped ratio	98.4%	97.2%	92.3%	81.78%	93.88%
Identical base	2,368,023,317	2,326,978,218	2,112,335,753	1,479,479,959	2,257,529,284
Sequence Identity	98.9%	97.3%	92.7%	70.06%	93.13%
Substitution rate	-	4.4E-09	3.2E-09	10.6E-09	5.7E-09
Divergence time (Mya)	-	3.05	11.33	14.1	6.0

MCR, *M. crinifrons*; MRE, *M. reevesi*; HIN, *H. inermis*.

Supplementary Table 2. Mutation rate and generation time. The mutation rate was calculated using r8s based on the phylogenetic tree with calibrated divergence time. Generation time of the species used in PSMC analysis are listed here. The generation time of *H. inermis* and *C. albirostris* are from Chen et al. (Chen et al. 2019) and that of the four muntjac species are from Di Marco et al. (Di Marco et al. 2013).

Species	Mutation rate	Generation time
Cattle	1.7406e-09	-
Reindeer	2.5032e-09	-
H. inermis	3.2651e-09	5
E. davidianus	2.2076e-09	-
C. albirostris	2.0384e-09	5
M. reevesi	2.3796e-09	2.5
M. gongshanensis	2.4537e-09	2.5
Female M. crinifrons	2.4796e-09	2.5
M. muntjac	2.4365e-09	2.5

11. L190 p10: What is the percentage of recovered bins for each species after the liftover?

Response: Over 90% bins at different resolutions in different genomes could be mapped with over 90% bins in the reference female *M. crinifrons* (black muntjac) genome using liftover. This result is now displayed in the **Extended Data Fig. 3b**.

12. L199 p10. Related to the previous comment, did the authors underestimate the number of fusion related TADs in non-fused chromosomes because of possible missing bins in other species due to the liftover?

Response: Thank you very much for this very important comment which helped us to clarify a problem, which influenced the conclusion about effects of chromosome fusion on TADs near fusion sites. The inclusion of male and female *M. crinifrons*

specific TADs overestimated the identified *M. crinions* specific TADs. The inclusion of missing bins missing bins when calculating the percentage of overlapped bins in TADs between female *M. crinifrons* (black muntjac) and other species also slightly underestimated the number of conserved TADs between other species and female *M. crinifrons*, so the specific TADs of *M. crinifrons* was slightly overestimated. We have corrected the defining method of *M. crinions* specific TADs (see the above response to your comment#6) and the comparison method of TADs, and now the reciprocally mapped bin pairs between genomes are considered. The missing bins in any genomes have now been excluded and the following three analyses show that the excluded missing bins does not change the TAD comparison results. This is because first, our liftover parameters reduce the proportion of total missing bins at 40 kb resolution to be less than 5% (**Extended Data Fig. 3b and Response Fig. 3**). Second, most TADs, no matter species-specific or conserved, actually have very few (0~2) missing bins (**Response Fig. 7**). Third, TADs with more missing bins are not enriched near the fusion sites (**Response Fig. 8**). As described in the response to your comment #6, when we went back to check this point, we have corrected the results by excluding the influence of male and female *M. crinifrons* samples and missing bins, the results show that TADs do not enrich close fusion sites, indicating that chromosome fusion has no greater impact on TADs. We have clarified all these points in the revised manuscript:

“To ensure that more bins (over 95% at 40 kb resolution) in female *M. crinifrons* genome have homologous bins in other three genomes, the liftOver tool (Kent et al. 2002) with different parameters (female *M. crinifrons* vs male *M. crinifrons* and female *M. crinifrons* vs *M. reevesi*: -minMatch=0.85, female *M. crinifrons* vs *H. inermis*: -minMatch=0.7) was used to obtain the homologous bin pairs (Supplementary Fig. 5).” (Line 671)

“Then based on the mapped bins of male *M. crinifrons*, *H. inermis* and *M. reevesi* with female *M. crinifrons*, we used the bedtools (Quinlan and Hall 2010) to obtain the overlapped regions between TADs of different genomes. Following the previous practice (Wu et al. 2017), the conserved or shared TADs are defined as two TADs in different genomes whose overlapped regions cover more than 70% of their respective lengths. We excluded missing bins when calculating the proportion of overlap interval. Because our liftover parameters reduce the proportion of total missing bins at the 40 kb resolution to be less than 5% (Supplementary Fig. 5) and most TADs actually have only 0~2 missing bin (Supplementary Fig. 6), the TAD comparison between genomes would not be affected. The *M. crinifrons*-specific TADs are defined as the TADs of female *M. crinifrons* which are shared with male *M. crinifrons* but are not shared with *H. inermis* and *M. reevesi*. The distance of bins in *M. crinifrons*-specific TADs from their nearest fusion sites were calculated and then counted using the “geom_density” function in the ggplot2 package (Villanueva and Chen 2019).” (Line 703)

Response Figure 3. Ratio of bins at 40 kb resolution in female *M. crinifrons* mapped with homologous bins in other three genomes.
MCR, *M. crinifrons*; MRE, *M. reevesi*; HIN, *H. inermis*.

Response Figure 7. Number of missing bins in different type of TADs. “female MCR_overlapped” means female *M. crinifrons* TADs that are homologous with TADs in male *M. crinifrons*, *M. reevesi* (Chinese muntjac) or *H. inermis* (Chinses water deer). “female MCR_specific” means female *M. crinifrons* TADs that are not homologous with TADs in male *M. crinifrons*, *M. reevesi* or *H. inermis*. Similarly, “male MCR_overlapped”, “MRE_overlapped” and “HIN_overlapped” respectively represent TADs of male *M. crinifrons*, *M. reevesi*, and *H. inermis* that are homologous

with female *M. crinifrons*. “male MCR_specific”, “MRE_specific” and “HIN_specific” represent male *M. crinifrons*, *M. reevesi*, and *H. inermis* TADs that are not homologous with female *M. crinifrons*.

Response Figure 8. Number of missing bins in TADs with different distance from female *M. crinifrons* fusion sites.

“female MCR overlapped with male MCR/MRE/HIN” means female *M. crinifrons* TADs that are homologous with TADs of male *M. crinifrons*, *M. reevesi* or *H. inermis*. “female MCR specific” means female *M. crinifrons* TADs that are not homologous with TADs of male *M. crinifrons*, *M. reevesi* or *H. inermis*. The fitting is done by using the `geom_smooth` function with method parameter “lm” in `ggplot2` Package. The gray transparent area along a lines indicates a confidence interval of 0.95.

13. L195 p10: It might be expected that a higher fraction of TADs would be shared between the muntjacs as the divergence times are 3 Mya maximum, when compared to human mouse that diverged 90 Mya. Can this difference be due to the methodological issues described above?

Response: Thank you for your comments. Following your suggestions, we have carefully scrutinized literature about the conservation of TAD between samples or species, while sometimes people were comparing TADs themselves, but sometimes were comparing the boundaries of TADs. Different strategies and parameters can be used to compare TAD boundaries or TADs across species or cell lines. It is clear that the stability of TAD boundaries between cell lines or species is often higher than that of TAD itself (McArthur and Capra 2020), which could also be inferred from many other works as we summarized in the **Response Table 3**. From these previous works, we found that the percentage of stable TADs (72.5%~72.7%) in different *M. crinifrons* individuals is almost the same as that between different human cell lines (70.9%~75.6%, their Fig. 4f) (Wu et al. 2017). Beyond this, we further found that the conservation of TADs decreases obviously along divergence time while boundary

changes very little when the divergence time varies from 15 to 90 million years (**Response Table 3**). These data suggest that TADs stability is not comparable to stability of TAD boundaries. We have rephrased the statement about comparison between our results and that of Dixon et al. in our revised manuscript.

14. L202 p10: Can the length of LRSI observed be due to chromosome length? Were more LRSI in black muntjac observed because they have the longest chromosomes? Also, are inter-chromosomal LRSI in Chinese muntjac (and other species with higher chromosome number) more frequent between chromosomes that are fused in black muntjac? To help resolve these issues, it would be helpful to determine if the fused chromosomes are physically closer in the nucleus.

Response: Thank you for your comment. We found that only 26.88% (15148/56345) long range significant interaction (LRSI) span fusion sites in female *M. crinifrons* (black muntjac). In other words, most of the LRSI in female *M. crinifrons* were established within the ancestral chromosome. Therefore, the plenty of long range significant interaction is not mainly due to its super-long fused chromosome. In addition, we simulated the 3D genome structure to show the physical position of chromosomes in the nucleus. We painted the homologous ancestral chromosomes in the nucleus of male and female *M. crinifrons*, *M. reevesi* and *H. inermis* using the same color (**Extended Data Fig. 5e**). The results show that the hypothesis that ancestral chromosomes physically closer in *M. reevesi* and *H. inermis* fused in *M. crinifrons* does not exist. We have added results about this issue in revised manuscript:

“In detail, 73.12% (15148/56345) of these long-range significant interactions (>5 Mb) are established within ancestral chromosome segments, indicating that these long-range significant interactions in *M. crinifrons* are not due to the calculation error caused by its super long chromosomes. Furthermore, most of these long-range significant interactions (88.38%) have no homologous significant interactions in *M. reevesi* (Extended Data Fig. 6c). These results suggested that these long-range significant interactions may be related to the more compacted chromosomes observed in the reconstructed 3D genome structure of *M. crinifrons* (Extended Data Fig. 6f).” (Line 263)

“However, the reconstructed 3D genome structure revealed that ancestral chromosome segments fused in *M. crinifrons* are not physically closer in the reconstructed 3D genome structures of *M. reevesi* and *H. inermis* (Extended Data Fig. 6e), indicating that the fusion events were not directly caused by spatial proximity of two ancestral chromosomes.” (Line 275)

15. L209 p11: Do authors think that these newly established interactions (LRSI) are formed de novo and are completely absent in the non-fused chromosomes?

Response: We compared the significant interactions of female *M. crinifrons* (black muntjac) with those of *M. reevesi* (Chinese muntjac). We found that 88.38% of the

long range significant interaction in female *M. crinifrons* don't have homologous significant interactions in *M. reevesi* (**Extended Data Fig. 5c**), which suggest they may be newly established in female *M. crinifrons*. In contrast, there are only 37.31% other significant interactions without homologous significant interactions in *M. reevesi* (**Extended Data Fig. 5c**). The number of significant interactions of *H. inermis* (199,010) is much fewer than that of female *M. crinifrons* (451,276) and *M. reevesi* (591,322). If *H. inermis* is compared with female *M. crinifrons*, the number of significant interactions of newly established in female *M. crinifrons* will be greatly overestimated, therefore we did not compare significant interactions of female *M. crinifrons* with *H. inermis*. Compared with only one outgroup species, we are not sure whether the newly established interactions (LRSI) in female *M. crinifrons* are indeed formed de novo and are completely absent in the non-fused chromosomes, and therefore we did not present these results and only discussed more data are needed to test if these *M. crinifrons* specific significant interactions are newly established or absent in non-fused chromosomes.

16. L211 p11: The authors did not mention in the methods if they compared black muntjac SIAFS with other species inter-chromosomal SIs. This should be done as black muntjac is the species with the lowest chromosome number and most of these interactions would be between chromosomes in the other species. Are the shared SIAFS between black muntjacs and Chinese muntjacs also shared fusions?

Response: We compared the significant interactions across fusion sites (SIAFS), as well the significant interactions (SIs) not across fusion sites in *M. crinifrons* (black muntjac) with all SIs of the *M. reevesi* (Chinese muntjac), including inter- and intra-chromosomal SIs. In *M. reevesi*, there are 931 SIs are homologous with SIAFS of female *M. crinifrons*, and 338,455 SIs are homologous with SIs not across fusion sites in female *M. crinifrons* (**Fig. 2c**). The 931 SIs include 73 inter-chromosomal SIs and 858 intra-chromosomal SIs. Almost all of the 858 SIs cross fusion sites in *M. reevesi* (**Response Fig. 9**), and most of them cross the six oldest fusion sites shared by five muntjac species (**Fig. 1a and Extended Data Fig. 2e**). Most of the 338,455 SIs in *M. reevesi* are also intra-chromosomal, and didn't cross fusion sites (**Response Fig. 9**). Because we focus on the significant interaction across fusion sites in *M. crinifrons* in the main text and *M. reevesi* is only used as a control, we do not add these results about *M. reevesi* in the main text.

significant interactions (SIs) of M. reevesi	931 SIs homologous with SIs across fusion sites in female M. crinifrons			
	intra-chromosome			inter-chromosome
	crossing fusion sites		not crossing fusion sites	
	shared fusion sites	specific fusion sites		
	856	1	1	73
	338,455 SIs homologous with SIs not across fusion sites in M. crinifrons			
	intra-chromosome			inter-chromosome
	crossing fusion sites		not crossing fusion sites	
	shared fusion sites	specific fusion sites		
	2 (crossing more than one fusion site)		338,067	386

Response Figure 9. Significant interactions (SIs) of *M. reevesi* homologous with SIs in female *M. crinifrons*.

Different colors indicate different levels of classification of significant interactions.

17. L308 p15: Where in the sequencing read (end or middle) are the truncated telomeric repeats? Could these be just an artifact of the sequencing technology (see comments above)?

Response: As described above, the “truncated telomeric sequence” is a very short telomeric sequence (~38bp or ~25bp) in the middle of the ONT long reads, not the direct telomeric sequences in the truncated point of the long reads. To prove that the short telomeric sequence is less possibly due to the problem of ONT reads, we obtained the distance between the telomeric sequence with the nearest end of the ONT long reads and calculated the percentage of the distance in total length of each ONT long reads (see **Response Fig. 6** above). The results show that the telomeric sequences do not biased in the end of ONT long reads in all four investigated species, but distribute in the middle of reads. We also further analyzed the PacBio reads of male *M. crinifrons* (black muntjac) from Chen et al. (Chen et al. 2019) by the same method, and still found that this featured short telomeric sequence are usually located in the middle of PacBio reads (**Response Fig. 6**). We added these results in the revised manuscript to show the structure of the short telomeric sequences is less likely to be false sequencing results in the long ONT reads:

“Further examination revealed that the truncated telomeric sequence is primarily located in the middle, not at the ends of the Nanopore reads (Supplementary Fig. 2), which also indicated that these truncated telomeric sequences are not caused by pre-termination of the Nanopore reads, but are more likely derived from the acrocentric regions of ancestral chromosomes and have originally been located in the region between satI and satIV (Supplementary Fig. 3).” (Line 359)

18. L331 p16: The figure cited appears to be 4e instead of 2e.

Response: Sorry for the typo. We have corrected as suggested.

19. L505 p24: Related to the first comment. Why did authors use a different minimum identity percentage (-minMatch) as threshold for different liftovers?

Response: Thank you for your comments. We had tried different parameters ranged from 0.70 to 0.95 to call the homologous bin pairs between female *M. crinifrons* (black muntjac) and other three genomes (**Response Fig. 3**). In order to ensure that more bins (over 95% at 40 kb resolution) between genomes can be homologously mapped, we chosen the parameter -minMatch=0.85 for female vs male *M. crinifrons* and female *M. crinifrons* vs *M. reevesi* (Chinses muntjac) and -minMacth=0.7 for female *M. crinifrons* vs *H. inermis* (Chinese water deer). We have added these pieces of information in the method section in revised manuscript:

“To ensure that more bins (over 95% at 40 kb resolution) in female *M. crinifrons* genome have homologous bins in other three genomes, the liftOver tool (Kent et

al. 2002) with different parameters (female *M. crinifrons* vs male *M. crinifrons* and female *M. crinifrons* vs *M. reevesi*: $-\text{minMatch}=0.85$, female *M. crinifrons* vs *H. inermis*: $-\text{minMatch}=0.7$) was used to obtain the homologous bin pairs (Supplementary Fig. 5).” (Line 671)

Response Figure 3. Ratios of bins at 40 kb resolution in female *M. crinifrons* mapped with homologous bin in other three genomes.

MCR, *M. crinifrons*; MRE, *M. reevesi*; HIN, *H. inermis*.

20. L512 p26: It is unclear if the same tissues were used for generating Hi-C data in the different species. This should be clarified. Why did the authors choose to use GC content and gene density to define A/B compartments? Would it not be better to use gene expression data?

Response: Thank you for your comment. All the tissues used for generating Hi-C data in different species are blood. We have classified this information in the revised manuscript.

Previous studies have revealed that the compartment type is associated with chromatin type, where compartment A correspond to euchromatin and compartment B correspond to heterochromatin (Lieberman-Aiden et al. 2009). Gene density and GC content in euchromatin (compartment A) are higher than that in heterochromatin (compartment B). Therefore, many studies also used the gene density and GC content to help distinguish compartment A/B (Barutcu et al. 2015; Wang et al. 2018; Wu et al. 2017; Luo et al. 2021), and some compartment analysis tools, such as FAN-C (<https://fan-c.readthedocs.io/en/latest/api/analyse/compartments.html>) recommend using gene density and GC content. In this study, we first used PC1 in the principal analysis (PC) to partitioned genomes into two types and then used the gene density and GC content to assign compartment A/B as most previous studies. Our results showed that gene density and GC content in compartment A are significantly higher than those in compartment B (**Response Fig. 1**), indicating that high gene density and

GC content are very sensitive to determine compartment A. Of course, compartment A/B can also be distinguished by expression data. We also display the expression level of female and male *M. crinifrons* (black muntjac) and *H. inermis* (Chinese water deer), together with compartment A/B, gene density and GC content (**Response Figure 2**). Expression data of female and male *M. crinifrons* are from cell line samples and that of *H. inermis* are from fresh blood sample. *M. reevesi* (Chinese muntjac) don't have available expression data. We have made these description clearer in the revised manuscript:

“At a 100 kb resolution, we identified the A or B (active or inactive) compartments of different muntjac species following the previous practice (Lieberman-Aiden et al. 2009; Wu et al. 2017). As expected, we found significantly higher gene density and GC content, higher gene expression level in the euchromatic A compartments than in the heterochromatic B compartments (Extended Data Fig. 3c, d) (Lieberman-Aiden et al. 2009).” (Line 236)

“Following many published studies (Lieberman-Aiden et al. 2009; Barutcu et al. 2015; Wang et al. 2018; Luo et al. 2021), the first principal component (PC1) was used to identify compartment A/B. Positive or negative values of the PC1 separate chromatin regions into two spatially segregated compartments and regions with higher gene density and GC content were assigned as compartment A, while the rest were compartment B.” (Line 683)

Response Figure 1. Gene density and GC content in compartment A and B.

The difference of gene density or GC content in different compartment regions was checked using T test.

Response Figure 2. Compartment A/B, gene expression level, GC content and gene density of female *M. crinifrons*, male *M. crinifrons*, *M. reevesi* and *Hydropotes inermis*. Compartment A/B: The blue part represent the compartment A, and the red part represent compartment B. **Gene expression level:** The gene expression level was represented by the FPKM (fragments per kilobase of transcript per million). **GC content:** the GC content per 100kb. **Gene density:** the length ratio of gene per 100 kb. For male *M. crinifrons*, 1p and 1q represent short arm and long arm of chromosome 1, respectively.

Reviewer #4 (Remarks to the Author):

The muntjacs genus perhaps is the most spectacular example of rapid karyotype evolution. The authors produced the chromosome-scale reference genomes from three muntjac deer species and one outgroup, the Chinese water deer. These genomes are at high quality therefore allowing the authors to produce an in-depth view on the chromosome evolution process within muntjacs genus and the accompanying chromatic structures. They also proposed some mechanisms underlying the rapid karyotype evolution in this genus in association with repeat contents and some rapid evolutionary genes related with the genomic architecture stability.

Overall, I found this study was an exciting one with many interesting findings. I however have few comments for the authors to consider for their revision.

Response: Thank you very much for your positive comments.

1. It is unclear to me the rationality of selecting these species, particularly, why the Chinese water deer from another clade was used as outgroup and cattle as the reference for these very recent evolutionary events. Some explanations in the introduction would be useful for the readers to understand the design of this project.

Response: Thank you for your helpful suggestion. Previous studies had indicated that the ancestral karyotype of all Cervidae species is $2n=70$ which is still remained in some Cervidae species, such as *H. inermis* (Chinese water deer) and brown-brocket deer (Yang et al. 1997a). Though species belonging to the Cervinae subfamily are more related to muntjac deer, they also experienced independent chromosome rearrangements during evolutionary process (Huang et al. 2006). Therefore, given that we did not obtain samples from brown-brocket deer, *H. inermis* is now the best outgroup candidate that could help us clarify the process of chromosome fusion from $2n=70$ in ancestor to $2n=8/9$ in muntjac deer.

In general, a reference species with high-quality genome assembly and gene annotation is pivotal for the phylogenetic analysis and gene evolution analysis. As one of the most important domestic species, cattle has very high-quality chromosome-level genome and gene annotation results. Therefore, we selected cattle as the reference species in this study. We have explained more about the selecting strategy of species in Introduction and Method materials in the revised manuscript:

“They were proposed to have an ancestral karyotype ($2n=70$) similar to that of *Hydropotes inermis* (Huang, Chi, Nie, et al. 2006), and recurrent chromosome fusions have led to the karyotypes of extant species varying from $2n=46$ of *M. reevesi* (Wurster and Benirschke 1967) to $2n=8/9$ of *M. crinifrons* or *M. gongshanensis* (Shi 1983; Shi and Ma 1988), and to $2n=6/7$ of *M. muntjac* (Wurster and Benirschke 1970).” (Line 92)

“Here, we produced high-quality chromosome-level genomes and large quantity of Hi-C data for multiple muntjacs and *H. inermis* representing ancestral karyotype (Fig. 1a and Extended Data Fig. 1a), which enable us to reconstruct the detailed process of chromosome fusions from the muntjac ancestor to the extant species, and to investigate the molecular basis of dramatic chromosome fusion events during muntjac species evolution and explore the impact of chromosome fusions on 3D chromatin architectures.” (Line 135)

“Firstly, genome of *B. taurus* (ARS-UCD1.2) was selected as reference, because it has very high-quality chromosome-level genome and gene annotation results as one of the most important domestic species.” (Line 575)

2. The Indian muntjac was completely ignored in all the chromosome analyses. This is surprising as this species has been substantially studied before particularly with many chromosome painting experiments reported before. Even though it was not sequenced long reads, its genome assembly was done in chromosome level which is even better than the Gongshan muntjac that the authors reported here. Including the

Indian muntjac genome would be more interesting than the Gongshan muntjac in telling the evolution of neo-XY in this genus and the genomic features near the fusion point. It is unfortunately that many questions regarding to the chromosome evolution in this genus have left behind without the Indian muntjac.

Response: Yes, we agree with you that the *M. muntjac* (Indian muntjac) has been substantially studied before because it has the least mammalian chromosome number. However, both of the *M. muntjac* and the *M. crinifrons* (black muntjac) had undergone drastic chromosome fusions, and their chromosome numbers only differ by one. In the design of this study, the *M. crinifrons* was chosen due to it has a neo-sex chromosome system which is absent in other muntjac species, including *M. muntjac* and *M. gongshanensis*. As the nearest species of *M. crinifrons*, data of *M. gongshanensis* is needed for us to more accurately reveal age of the neo-sex chromosomes of *M. crinifrons* and explore the degeneration level of neo-Y using *M. gongshanensis* as a controlled outgroup. The corresponding explanation in the revised manuscript is as follows:

“As one of the muntjacs with a very low chromosome number, *M. crinifrons* additionally possesses one sex chromosome system that does not exist in other muntjac species including its closest relative, *M. gongshanensis*.” (Line 103)

“We estimated the divergence time among *M. muntjac*, *M. gongshanensis* and *M. crinifrons* to be about 1~2 million years ago (Mya). Particularly the 1.44 Mya divergence time between *M. crinifrons* and *M. gongshanensis* (Fig. 1a and Extended Data Fig.2b) sets the upper limit for the age of neo-sex chromosomes of *M. crinifrons*.” (Line 171)

“Identification and annotation of candidate male-specific mutations. We separately detected the candidate male-specific SNPs and indels using the Illumina reads and structural variations (SVs) using the PacBio reads. Firstly, to detect the candidate male-specific SNPs and indels, we aligned the Illumina reads from four *M. crinifrons* individuals (female MCR2 and MCR3, male MCR2 and MCR3) and two *M. gongshanensis* individuals (MGO1 and MGO2) to the female *M. crinifrons* genome using bwa software (Li and Durbin 2009).” (Line 858)

Furthermore, the amount of published Hi-C data of *M. muntjac* is not enough to support the analysis about compartment A/B, TAD and significant interactions. Due to the lack of available long reads, the *M. muntjac* could not be included in the analysis of fusion mechanism. In some analysis needed more muntjac species with low chromosome number, such as phylogeny and divergence time, population history and gene evolution, we indeed used the genome data of *M. muntjac*. The corresponding explanation in the revised manuscript is as follows:

“To provide a phylogenetic framework for subsequent evolutionary analysis, by including the published *M. muntjac*’s draft genome sequences (Chen et al. 2019),

we reconstructed the maximum likelihood (ML) tree for muntjac deer based on the fourfold degenerate sites (4dTV) and mitochondrial genomes (Fig. 1a and Extended Data Fig.2a).” (Line 168)

“To test the second hypothesis, we identified the rapidly evolving genes (REGs) and positively selected genes (PSGs) in the *M. crinifrons*, *M. gongshanensis* and *M. muntjac* with large fused chromosomes, as well in their common ancestor node (Supplementary Data 3).” (Line 379)

We wish we have explained the reason why we had chosen *M. crinifrons*, and wish you would understand the treatment. Now we don't have high quality sample of *M. muntjac* in hand to generate long reads. In the future, long reads from *M. muntjac*, no matter by other group or our team, can further testify the results observed in this study.

3. It is also lack of explanation for including the Gongshan muntjac. What conclusion was made from the comparison between Gongshan muntjac and black muntjac?

Response: In the revised manuscript we further strengthened the descriptions about the inclusion of *M. gongshanensis*. Among these muntjac species, only *M. crinifrons* own the neo-sex chromosome system and *M. gongshanensis* is its closest species. Adding *M. gongshanensis* into the phylogeny and divergence time analysis is helpful for revealing the age of neo-sex chromosome system of *M. crinifrons* and more accurately reveal the fine patterns of neo-sex chromosome evolution in *M. crinifrons*. For example, for the neo-sex chromosome evolution analysis, it is very helpful to detect the male-specific mutations occurred in the inverted regions on the neo-Y chromosome of *M. crinifrons* when we added the *M. gongshanensis* as the outgroup. The corresponding statement in the revised manuscript is as follows:

“As one of the muntjacs with a very low chromosome number, *M. crinifrons* additionally possesses one sex chromosome system that does not exist in other muntjac species including its closest relative, *M. gongshanensis*.” (Line 103)

“We estimated the divergence time among *M. muntjac*, *M. gongshanensis* and *M. crinifrons* to be about 1~2 million years ago (Mya). Particularly the 1.44 Mya divergence time between *M. crinifrons* and *M. gongshanensis* (Fig. 1a and Extended Data Fig.2b) sets the upper limit for the age of neo-sex chromosomes of *M. crinifrons*.” (Line 171)

“Identification and annotation of candidate male-specific mutations. We separately detected the candidate male-specific SNPs and indels using the Illumina reads and structural variations (SVs) using the PacBio reads. Firstly, to detect the candidate male-specific SNPs and indels, we aligned the Illumina reads from four *M. crinifrons* individuals (female MCR2 and MCR3, male MCR2 and MCR3) and two *M. gongshanensis* individuals (MGO1 and MGO2)

to the female *M. crinifrons* genome using bwa software (Li and Durbin 2009).”
(Line 858)

4. The authors claimed that the population decline of the species with reduced chromosome numbers in around 1 million years ago might lead to the quick fixation of reduced karyotypes in population by genetic drift. This is a highly speculative claim as first, the demographic pattern produced by PSMC in such small time windows at 1 million years ago is less reliable. And second, it is hardly to claim the N_e at $10-20 \times 10^4$ is a small one.

Response: Thank you for your comment. We agree with you that this is a speculative claim. We have toned down the statement to show this is a possible explanation (“**However, this conclusion needs more evidence, such as more population data, due to the reduced reliability of demographic inference of PSMC method at 1 Mya.**”), although, as the reviewer#3 pointed out, this result is important and make sense to understanding the previous possible speciation events in muntjacs. The N_e at $10-15 \times 10^4$ of the three muntjac species with largely fused chromosome is not indeed a small one, but much smaller than that of *M. reevesi* at the same period. We have toned down the corresponding statement in the revised manuscript:

5. Line 174, It would be useful to explain what tandem fusion and Robertsonian fusion are in the main text.

Response: Thank you for your suggestion. We have added a sentence to explain tandem and Robertsonian fusions as below:

“A fusion event was defined as tandem fusion if it connected the apical centromeres of one ancestral chromosomes and the distal telomere of another ancestral chromosome, and a fusion event was defined as Robertsonian fusion if it connected the apical centromeres of two ancestral chromosomes.” (Line 201)

6. Line 184-5, extended data Fig 2f only shows the synteny information but not the sequence identity, please provide the substitution rate in each node to confirm this claim.

Response: Thank you for your helpful comment, which is also raised by the reviewer#1 and reviewer#3. Following your suggestions, we additionally calculated the substitution rate and sequence identity between chromosomal genomes between female *M. crinifrons* (black muntjac, $2n=8$) and *M. reevesi* (Chinese muntjac, $2n=46$) as well as *H. inermis* (Chinese water deer, $2n=70$) (**supplementary table 8**). For comparison, we also calculated the substitution rate and sequence identity between goat ($2n=60$) and cattle ($2n=60$) as well as sheep ($2n=54$) using the same method. Although the difference of chromosome number among female *M. crinifrons*, *M. reevesi* and *H. inermis* are so large, the substitution rates between these genomes are not higher than that among goat, cattle and sheep which own similar chromosome number (**supplementary table 8**). In addition, based on phylogenetic tree with calibrated divergence time (**Fig. 1a**), we also used the r8s to calculate the mutation rate of all species in the phylogenetic tree (**supplementary table 6**). The mutation

rates of these species are similar to that calculated by Chen et al. (Chen et al. 2019) and don't have difference between species with large difference in chromosome number. Overall, all of these results demonstrated that the rapid karyotype evolution among muntjacs is not accompanied by rapid evolution of genomic sequences. We have added these comparison results in the Result part of revised manuscript:

“We compared the genome-wide substitution rates between species (*M. reevesi* (2n=46), *H. inermis* (2n=70) vs female *M. crinifrons* (2n=8)) with dramatic karyotype changes and Bovidae species (*B. taurus* (2n=60), *Ovis aries* (2n=54) vs *Capra hircus* (2n=60)) with similar karyotypes. The results show that at least 90% of female *M. crinifrons* genomic sequences could be mapped to *M. reevesi* and *H. inermis* with average sequence identity more than 90% (Extended Data Fig. 2f and supplementary Table 8), similar to that of the Bovidae species (supplementary Table 8). In addition, the substitution rates and mutation rate between different muntjac genomes are also similar to those between Bovidae species (supplementary Table 6 and 8). These results demonstrated that the rapid karyotype evolution among muntjacs is not accompanied by rapid evolution of genomic sequences.” (Line 213).

Supplementary Table 3. The statistics of genome alignment results. The total base mean the smallest total chromosome length between genome pairs. The alignment ratio was the percentage of alignment length in total base. Sequence identity is the percentage of exact matched base in alignment length. Substitution rate is the number of substituted sites divided by two folds of divergence time.

	Female MCR as the reference			Goat as the reference	
	Male MCR	MRE	HIN	Cattle	Sheep
Total base	2,432,301,072	2,459,394,662	2,468,664,404	2,582,134,882	2,582,134,882
Mapped base	2,392,894,791	2,391,481,492	2,279,835,188	2,111,717,083	2,423,998,847
Mapped ratio	98.4%	97.2%	92.3%	81.78%	93.88%
Identical base	2,368,023,317	2,326,978,218	2,112,335,753	1,479,479,959	2,257,529,284
Sequence Identity	98.9%	97.3%	92.7%	70.06%	93.13%
Substitution rate	-	4.4E-09	3.2E-09	10.6E-09	5.7E-09
Divergence time (Mya)	-	3.05	11.33	14.1	6.0

MCR, *M. crinifrons*; MRE, *M. reevesi*, HIN, *H. inermis*.

Supplementary Table 4. Mutation rate and generation time. The mutation rate was calculated using r8s based on the phylogenetic tree with calibrated divergence time. Only the generation time of the species used in our PSMC analysis are listed here. The generation time of *H. inermis* and *C. albirostris* are from Chen et al. (Chen et al. 2019) and that of the four muntjac species are from Di Marco et al. (Di Marco et al. 2013).

Species	Mutation rate	Generation time
Cattle	1.7406e-09	-
Reindeer	2.5032e-09	-

H. inermis	3.2651e-09	5
E. davidianus	2.2076e-09	-
C. albirostris	2.0384e-09	5
M. reevesi	2.3796e-09	2.5
M. gongshanensis	2.4537e-09	2.5
Female M. crinifrons	2.4796e-09	2.5
M. muntjac	2.4365e-09	2.5

7. Line201-202, BM-specific TADs are more frequently located around the fusion sites of BMF, what is the control in this analysis?

Response: Thank you for your question. The control in this analysis was the frequency of *M. crinifrons* -specific TADs more than 3.5 Mb away from fusion sites.

However, intrigued by the comment #12 of reviewer#3, when we went back to check missing bins and distribution of TADs along the chromosomes, we realized that we should not have mixed female and male *M. crinifrons* specific TADs when we tried to look at the impact of chromosome fusions on TAD distribution because the TAD difference between female and male *M. crinifrons* are most possibly caused by sex difference rather than by fusions. Previously we defined the specific TADs of *M. crinifrons* as TADs specific in both or either of female and male *M. crinifrons*, which would overestimate the specific TADs in *M. crinifrons*, because about 30% of TADs are different between female and male, and such TADs may have resulted from sex difference rather than chromosome fusions. To be conservative and more accurate, we identify *M. crinifrons* specific TADs as female *M. crinifrons* TADs that are not shared by *M. reevesi* and *H. inermis*, but shared with male *M. crinifrons*. And we also excluded all missing bins when comparing TADs between species because they may also slightly overestimate female *M. crinifrons* specific TADs although the effect is very limited on the TAD comparison. After these corrections, now the specific TADs of *M. crinifrons* do not group near the fusion sites. Therefore, it seems that chromosome fusion has no greater impact on TADs either, as the case in compartment, but significant interactions have been intensively reshaped as we show in the manuscript. We have updated all related statements in our revised manuscript.

8. Why BMF is used as representative of black muntjac in chromatic structure analyses?

Response: We used the female *M. crinifrons* (black muntjac) to represent *M. crinifrons* due to the following two reasons. Firstly, the chromosome composition of female *M. crinifrons* is simpler than male *M. crinifrons* which has a complex sex chromosome system, making chromatic structure analyses more complicated. In addition to the neo-X chromosome formed by the fusion of X chromosome and chromosome 4, male *M. crinifrons* also has a male-specific translocation from short arm of chromosome 1 to chromosome 4, forming neo-Y chromosome. The neo-Y chromosome own a heterozygous inverted region where the differentiation between neo-X and neo-Y chromosomes in male *M. crinifrons* is incomplete. Therefore, it is

convenient for us to identify chromosome fusion events or 3D genome architecture changes when we mapped other muntjac genomes to the female *M. crinifrons*. Secondly, the quality of genome assembly of female *M. crinifrons* is higher than that of male *M. crinifrons*. We have explained it in the Method section in our revised manuscript:

“It is worth noted that the analysis of compartment A/B, TADs and neo-sex chromosome include female and male *M. crinifrons* genomes, while in all other analysis, just female *M. crinifrons* are used to represent *M. crinifrons*, due to that the female *M. crinifrons* has higher assembly quality and simpler chromosome composition.” (Line 559)

9. Line 254-255, was this comparison done on whole genome level or only on neo-Y regions?

Response: Thank you for your question. We did this comparison not only for the neo-Y regions but also for other genomic regions that are as the genomic background control. The densities of SNPs and insertions/deletions (indels) between male and female *M. crinifrons* samples are almost the same in other genomic regions. Because of the recombination suppression between neo-X and the neo-Y regions in male *M. crinifrons*, male individual would accumulate more mutations in the neo-Y regions, the density of SNPs or indels in neo-Y regions are higher in male *M. crinifrons* samples than female samples. We have made the description clearer in the revised manuscript:

“Although limited sequence degeneration was identified in the neo-Y regions, we found higher densities of SNPs and insertions/deletions (indels) of male *M. crinifrons* individuals than those of female in the homologous neo-sex regions but not in the rest of the genome (Fig. 4a, track D), indicating the early divergence between neo-Y and neo-X.” (Line 408)

10. Line 266-267, what is the definition of differently express between neo-X and neo-Y alleles? The table does not include this information.

Response: We defined the differentially expressed neo-X or neo-Y alleles in the method section. Namely, using the candidate male-specific SNP as allele markers, we distinguished RNA-seq reads of male *M. crinifrons* from neo-X and neo-Y allele using an in-house perl script. For a specific gene in the neo-Y regions, if it contains at least two male-specific SNPs and the neo-X and neo-Y RNA-sequencing reads numbers are significantly different (paired samples *t*-test, *p*-value <0.05), this gene was defined as having differentially expressed alleles between neo-X and neo-Y. The last column of the **Supplementary Data 6** shows the value of *p*-value.

11. line 270, change ‘conserved’ to ‘similar’

Response: Corrected as suggested.

12. The figure legend needs to be improved by adding more detail explanation on all elements on the plots. Like Figure 1a, there is no explanation about the arrow in the

plot. What is the meaning of '1p+4' in figure 1d. Figure 2b, what does the box plot tell? I can only say some examples here, but the figure legends for almost all figures (including the extended ones) need to be improved.

Response: Thank you very much for your suggestion. We have added the detailed explanation on all elements on the plots and improved the legend of all our figures in the revised manuscript.

References cited in the response:

- Aldiri, I., B. Xu, L. Wang, X. Chen, D. Hiler, L. Griffiths, M. Valentine, A. Shirinifard, S. Thiagarajan, A. Sablauer, M. E. Barabas, J. Zhang, D. Johnson, S. Frase, X. Zhou, J. Easton, J. Zhang, E. R. Mardis, R. K. Wilson, J. R. Downing, M. A. Dyer, and Project St. Jude Children's Research Hospital-Washington University Pediatric Cancer Genome. 2017. 'The Dynamic Epigenetic Landscape of the Retina During Development, Reprogramming, and Tumorigenesis', *Neuron*, 94: 550-68 e10.
- Barutcu, A. R., B. R. Lajoie, R. P. McCord, C. E. Tye, D. Hong, T. L. Messier, G. Browne, A. J. van Wijnen, J. B. Lian, J. L. Stein, J. Dekker, A. N. Imbalzano, and G. S. Stein. 2015. 'Chromatin interaction analysis reveals changes in small chromosome and telomere clustering between epithelial and breast cancer cells', *Genome Biol*, 16: 214.
- Chen, L., Q. Qiu, Y. Jiang, K. Wang, Z. Lin, Z. Li, F. Bibi, Y. Yang, J. Wang, W. Nie, W. Su, G. Liu, Q. Li, W. Fu, X. Pan, C. Liu, J. Yang, C. Zhang, Y. Yin, Y. Wang, Y. Zhao, C. Zhang, Z. Wang, Y. Qin, W. Liu, B. Wang, Y. Ren, R. Zhang, Y. Zeng, R. R. da Fonseca, B. Wei, R. Li, W. Wan, R. Zhao, W. Zhu, Y. Wang, S. Duan, Y. Gao, Y. E. Zhang, C. Chen, C. Hvilsom, C. W. Epps, L. G. Chemnick, Y. Dong, S. Mirarab, H. R. Siegismund, O. A. Ryder, M. T. P. Gilbert, H. A. Lewin, G. Zhang, R. Heller, and W. Wang. 2019. 'Large-scale ruminant genome sequencing provides insights into their evolution and distinct traits', *Science*, 364.
- Chi, J., B. Fu, W. Nie, J. Wang, A. S. Graphodatsky, and F. Yang. 2005. 'New insights into the karyotypic relationships of Chinese muntjac (*Muntiacus reevesi*), forest musk deer (*Moschus berezovskii*) and gayal (*Bos frontalis*)', *Cytogenet Genome Res*, 108: 310-6.
- Crane, E., Q. Bian, R. P. McCord, B. R. Lajoie, B. S. Wheeler, E. J. Ralston, S. Uzawa, J. Dekker, and B. J. Meyer. 2015. 'Condensin-driven remodelling of X chromosome topology during dosage compensation', *Nature*, 523: 240-4.
- Di Marco, Moreno, Michela Pacifici, Luca Santini, Daniele Baisero, Lucilla Francucci, Gabriele Grottolo Marasini, Piero Visconti, and Carlo Rondinini. 2013. 'Generation length for mammals', *Nature Conservation*, 5: 89-94.
- Dixon, J. R., S. Selvaraj, F. Yue, A. Kim, Y. Li, Y. Shen, M. Hu, J. S. Liu, and B. Ren. 2012. 'Topological domains in mammalian genomes identified by analysis of chromatin interactions', *Nature*, 485: 376-80.
- Dong, P., X. Tu, P. Y. Chu, P. Lu, N. Zhu, D. Grierson, B. Du, P. Li, and S. Zhong. 2017. '3D Chromatin Architecture of Large Plant Genomes Determined by Local A/B Compartments', *Mol Plant*, 10: 1497-509.
- Du, Z., H. Zheng, Y. K. Kawamura, K. Zhang, J. Gassler, S. Powell, Q. Xu, Z. Lin, K. Xu, Q. Zhou, E.

- A. Ozonov, N. Veron, B. Huang, L. Li, G. Yu, L. Liu, W. K. Au Yeung, P. Wang, L. Chang, Q. Wang, A. He, Y. Sun, J. Na, Q. Sun, H. Sasaki, K. Tachibana, Ahfm Peters, and W. Xie. 2020. 'Polycomb Group Proteins Regulate Chromatin Architecture in Mouse Oocytes and Early Embryos', *Mol Cell*, 77: 825-39 e7.
- Eres, I. E., K. Luo, C. J. Hsiao, L. E. Blake, and Y. Gilad. 2019. 'Reorganization of 3D genome structure may contribute to gene regulatory evolution in primates', *PLoS Genet*, 15: e1008278.
- Ferguson-Smith, M. A., and V. Trifonov. 2007. 'Mammalian karyotype evolution', *Nat Rev Genet*, 8: 950-62.
- Frohlich, J., S. Kubickova, P. Musilova, H. Cernohorska, H. Muskova, R. Vodicka, and J. Rubes. 2017. 'Karyotype relationships among selected deer species and cattle revealed by bovine FISH probes', *PLoS One*, 12: e0187559.
- Fudenberg, Geoffrey, Maxim Imakaev, Carolyn Lu, Anton Goloborodko, Nezar Abdennur, and Leonid A. Mirny. 2016. 'Formation of Chromosomal Domains by Loop Extrusion', *Cell Reports*, 15: 2038-49.
- Gharib, W. H., and M. Robinson-Rechavi. 2013. 'The branch-site test of positive selection is surprisingly robust but lacks power under synonymous substitution saturation and variation in GC', *Mol Biol Evol*, 30: 1675-86.
- Hartmann, N., and H. Scherthan. 2004. 'Characterization of ancestral chromosome fusion points in the Indian muntjac deer', *Chromosoma*, 112: 213-20.
- Huang, L., J. Chi, W. Nie, J. Wang, and F. Yang. 2006. 'Phylogenomics of several deer species revealed by comparative chromosome painting with Chinese muntjac paints', *Genetica*, 127: 25-33.
- Huang, L., J. Chi, J. Wang, W. Nie, W. Su, and F. Yang. 2006. 'High-density comparative BAC mapping in the black muntjac (*Muntiacus crinifrons*): molecular cytogenetic dissection of the origin of MCR 1p+4 in the X1X2Y1Y2Y3 sex chromosome system', *Genomics*, 87: 608-15.
- Kent, W. J., C. W. Sugnet, T. S. Furey, K. M. Roskin, T. H. Pringle, A. M. Zahler, and D. Haussler. 2002. 'The human genome browser at UCSC', *Genome Research*, 12: 996-1006.
- Li, H. 2011. 'A statistical framework for SNP calling, mutation discovery, association mapping and population genetical parameter estimation from sequencing data', *Bioinformatics*, 27: 2987-93.
- Li, H., and R. Durbin. 2009. 'Fast and accurate short read alignment with Burrows-Wheeler transform', *Bioinformatics*, 25: 1754-60.
- Li, Y. C., C. Lee, D. Sanoudou, T. H. Hseu, S. Y. Li, and C. C. Lin. 2000b. 'Interstitial colocalization of two cervid satellite DNAs involved in the genesis of the Indian muntjac karyotype', *Chromosome Res*, 8: 363-73.
- Li, Yueh-Chun, and Chyi-Chyang Lin. 2011. 'Cervid satellite DNA and karyotypic evolution of Indian muntjac', *Genes & Genomics*, 34: 7-11.
- Lieberman-Aiden, E., N. L. van Berkum, L. Williams, M. Imakaev, T. Ragoczy, A. Telling, I. Amit, B. R. Lajoie, P. J. Sabo, M. O. Dorschner, R. Sandstrom, B. Bernstein, M. A. Bender, M. Groudine, A. Gnirke, J. Stamatoyannopoulos, L. A. Mirny, E. S. Lander, and J. Dekker. 2009. 'Comprehensive mapping of long-range interactions reveals folding principles of the human genome', *Science*, 326: 289-93.
- Lin, C. C., P. Y. Chiang, L. J. Hsieh, S. J. Liao, M. C. Chao, and Y. C. Li. 2004. 'Cloning, characterization and physical mapping of three cervid satellite DNA families in the genome of the Formosan muntjac (*Muntiacus reevesi micrurus*)', *Cytogenet Genome Res*, 105: 100-6.
- Lin, C. C., and Y. C. Li. 2006. 'Chromosomal distribution and organization of three cervid satellite DNAs

- in Chinese water deer (*Hydropotes inermis*), *Cytogenet Genome Res*, 114: 147-54.
- Liu, Y, WH Nie, L Huang, JH Wang, and SU Wei-Ting. 2008. 'Cloning, characterization, and FISH mapping of four satellite DNAs from black muntjac (*M. crinifrons*) and Fea's muntjac (*M. feae*)', *Zoological Research*, 29: 225-35.
- Luo, Xin, Yuting Liu, Dachang Dang, Ting Hu, Yingping Hou, Xiaoyu Meng, Fengyun Zhang, Tingting Li, Can Wang, Min Li, Haixu Wu, Qiushuo Shen, Yan Hu, Xuerui Zeng, Xiechao He, Lanzhen Yan, Shihua Zhang, Cheng Li, and Bing Su. 2021. '3D Genome of macaque fetal brain reveals evolutionary innovations during primate corticogenesis', *Cell*.
- McArthur, Evonne, and John A. Capra. 2020. 'Topologically associating domain (TAD) boundaries stable across diverse cell types are evolutionarily constrained and enriched for heritability'.
- Mudd, A. B., J. V. Bredeson, R. Baum, D. Hockemeyer, and D. S. Rokhsar. 2020. 'Analysis of muntjac deer genome and chromatin architecture reveals rapid karyotype evolution', *Commun Biol*, 3: 480.
- Pardo-Manuel de Villena, F., and C. Sapienza. 2001. 'Female meiosis drives karyotypic evolution in mammals', *Genetics*, 159: 1179-89.
- Quinlan, A. R., and I. M. Hall. 2010. 'BEDTools: a flexible suite of utilities for comparing genomic features', *Bioinformatics*, 26: 841-2.
- Rao, S. S., M. H. Huntley, N. C. Durand, E. K. Stamenova, I. D. Bochkov, J. T. Robinson, A. L. Sanborn, I. Machol, A. D. Omer, E. S. Lander, and E. L. Aiden. 2014. 'A 3D map of the human genome at kilobase resolution reveals principles of chromatin looping', *Cell*, 159: 1665-80.
- Rowley, M. J., M. H. Nichols, X. Lyu, M. Ando-Kuri, I. S. M. Rivera, K. Hermetz, P. Wang, Y. Ruan, and V. G. Corces. 2017. 'Evolutionarily Conserved Principles Predict 3D Chromatin Organization', *Mol Cell*, 67: 837-52 e7.
- Servant, N., N. Varoquaux, B. R. Lajoie, E. Viara, C. J. Chen, J. P. Vert, E. Heard, J. Dekker, and E. Barillot. 2015. 'HiC-Pro: an optimized and flexible pipeline for Hi-C data processing', *Genome Biol*, 16: 259.
- Shi, L. M. 1983. 'Sex-linked chromosome polymorphism in black muntjac, *Muntiacus crinifrons*', *In: Swamina-Tham M S, ed. New Dehli: Proceedings of the Fifth International Congress of Genetics*, 153.
- Shi, L. M., and C. X. Ma. 1988. 'A new karyotype of muntjac (*Muntiacus* sp.) from Gongshan county in China', *Zool. Res.*, 9: 343-47.
- Torosin, N. S., A. Anand, T. R. Golla, W. Cao, and C. E. Ellison. 2020. '3D genome evolution and reorganization in the *Drosophila melanogaster* species group', *PLoS Genet*, 16: e1009229.
- Tsipouri, V., M. G. Schueler, S. Hu, Nisc Comparative Sequencing Program, A. Dutra, E. Pak, H. Riethman, and E. D. Green. 2008. 'Comparative sequence analyses reveal sites of ancestral chromosomal fusions in the Indian muntjac genome', *Genome Biol*, 9: R155.
- Villanueva, Randle Aaron M., and Zhuo Job Chen. 2019. 'ggplot2: Elegant Graphics for Data Analysis (2nd ed.)', *Measurement: Interdisciplinary Research and Perspectives*, 17: 160-67.
- Wang, M., P. Wang, M. Lin, Z. Ye, G. Li, L. Tu, C. Shen, J. Li, Q. Yang, and X. Zhang. 2018. 'Evolutionary dynamics of 3D genome architecture following polyploidization in cotton', *Nat Plants*, 4: 90-97.
- Wang, Y., H. Wang, Y. Zhang, Z. Du, W. Si, S. Fan, D. Qin, M. Wang, Y. Duan, L. Li, Y. Jiao, Y. Li, Q. Wang, Q. Shi, X. Wu, and W. Xie. 2019. 'Reprogramming of Meiotic Chromatin Architecture during Spermatogenesis', *Mol Cell*, 73: 547-61 e6.

- Wang, Y., C. Zhang, N. Wang, Z. Li, R. Heller, R. Liu, Y. Zhao, J. Han, X. Pan, Z. Zheng, X. Dai, C. Chen, M. Dou, S. Peng, X. Chen, J. Liu, M. Li, K. Wang, C. Liu, Z. Lin, L. Chen, F. Hao, W. Zhu, C. Song, C. Zhao, C. Zheng, J. Wang, S. Hu, C. Li, H. Yang, L. Jiang, G. Li, M. Liu, T. S. Sonstegard, G. Zhang, Y. Jiang, W. Wang, and Q. Qiu. 2019. 'Genetic basis of ruminant headgear and rapid antler regeneration', *Science*, 364.
- Wu, P., T. Li, R. Li, L. Jia, P. Zhu, Y. Liu, Q. Chen, D. Tang, Y. Yu, and C. Li. 2017. '3D genome of multiple myeloma reveals spatial genome disorganization associated with copy number variations', *Nat Commun*, 8: 1937.
- Wurster, Doris H., and Kurt Benirschke. 1967. 'Chromosome studies in some deer, the springbok, and the pronghorn, with notes on placentation in deer', *Cytologia*, 32: 273-85.
- Wurster D.H., Benirschke K. 1970. 'Indian muntjac, *Muntiacus muntjak*: a deer with a low diploid chromosome number', *Science*, 168: 1364-66.
- Yang, F., P. C. O'Brien, J. Wienberg, and M. A. Ferguson-Smith. 1997c. 'Evolution of the black muntjac (*Muntiacus crinifrons*) karyotype revealed by comparative chromosome painting', *Cytogenet Cell Genet*, 76: 159-63.
- Yang, F., P. C. O'Brien, J. Wienberg, H. Neitzel, C. C. Lin, and M. A. Ferguson-Smith. 1997a. 'Chromosomal evolution of the Chinese muntjac (*Muntiacus reevesi*)', *Chromosoma*, 106: 37-43.
- Zhou, Q., J. Wang, L. Huang, W. Nie, J. Wang, Y. Liu, X. Zhao, F. Yang, and W. Wang. 2008. 'Neo-sex chromosomes in the black muntjac recapitulate incipient evolution of mammalian sex chromosomes', *Genome Biol*, 9: R98.
- Zhou, Yingyao, Bin Zhou, Lars Paché, Max Chang, Alireza Hadj Khodabakhshi, Olga Tanaseichuk, Christopher Benner, and Sumit K. Chanda. 2019. 'Metascape provides a biologist-oriented resource for the analysis of systems-level datasets', *Nat Commun*, 10.

REVIEWERS' COMMENTS

Reviewer #1 (Remarks to the Author):

Thank you for all the thorough responses to the comments. I have no further comments and find the paper in the current state ready for acceptance.

Reviewer #2 (Remarks to the Author):

The authors have done a good job responding to my previous comments, and I think the manuscript has improved substantially. I have a couple of minor comments remaining, but after these have been addressed I think the manuscript will be ready for publication.

L86: Please rephrase, rodent is not a species. I would argue that rodents, gibbon, and muntjacs are not that closely related.

Fig 2F: I think the old species abbreviations are still being used here (BMF, CM), please change according to the new species labels.

Reviewer #3 (Remarks to the Author):

The authors have made a great effort to respond to my concerns, as well as those of the other reviewers. After careful editing for grammar, spelling and most importantly, clarity, I recommend this article be accepted for publication.

Reviewer #4 (Remarks to the Author):

The authors have addressed all my concerns in the revision.

Response to reviewers:

Reviewer #2 (Remarks to the Author)

1. L86: Please rephrase, rodent is not a species. I would argue that rodents, gibbon, and muntjacs are not that closely related.

Response: Thank you for your suggestion. We have changed the description to avoid ambiguity:

Even in related species, such as within rodents, gibbons and muntjacs, the chromosome number can be dramatically different.

2. Fig 2F: I think the old species abbreviations are still being used here (BMF, CM), please change according to the new species labels.

Response: Sorry for overlooking this problem in the figure. We have checked and corrected the species abbreviations throughout the manuscript again.